# L.S.D. of sample covariances of superposition of matrices with separable covariance structure

Javed Hazarika*
Indian Statistical Institute, Kolkata

Debashis Paul
Indian Statistical Institute, Kolkata
*and*
University of California, Davis

### Abstract

We study the asymptotic behavior of the spectra of matrices of the form $S_n = \frac{1}{n}XX^*$, where $X = \sum_{r=1}^{K} X_r$ and $X_r = A_r^{\frac{1}{2}} Z_r B_r^{\frac{1}{2}}$, $K \in \mathbb{N}$. Here, $\{Z_r : r \in [K]\}$ are $p \times n$ matrices containing zero mean, unit variance innovation entries. $\{A_r : r \in [K]\}$ and $\{B_r : r \in [K]\}$ are sequences of p.s.d. matrices of order $p$ and $n$, respectively, which are simultaneously diagonalizable within themselves. We establish the existence of a limiting spectral distribution (L.S.D.) for $S_n$ assuming that the joint spectral distributions of $\{A_r : r \in [K]\}$ and $\{B_r : r \in [K]\}$ converge to some $K$-dimensional distributions as $p, n \to \infty$ such that $p/n \to c \in (0, \infty)$. The L.S.D. of $S_n$ is characterized by a system of equations with unique solutions within the class of Stieltjes transforms of measures on $\mathbb{R}_+$. Our findings generalize existing results associated with the L.S.D. of sample covariance matrices when the data matrices have a separable covariance structure.

**Keywords:** Limiting spectral distribution; Random matrix theory; Stieltjes transform; Separable Covariance.

**Mathematics Subject Classification (2020):** 62H10; 60B20

## 1 Introduction

There is now an extensive body of literature focusing on the limiting behavior of the spectra of sample covariance matrices corresponding to data matrices with a separable covariance structure under high dimensional asymptotic regimes. The earliest comprehensive work, in particular establishing the limiting spectral distribution (L.S.D.) of sample covariances, was by Zhang (2006), which has been followed by Paul and Silverstein (2009), Couillet and Hachem (2014), who covered various aspects of the limiting spectrum. More recently, CLTs for linear spectral statistics of sample covariances have been established under a separable covariance model Bai et al. (2019).

Separable covariance models represent an interesting generalization of multivariate data with independent realizations. In particular, such models have been used in spatio-temporal data analysis (Genton and Kleiber, 2015; Li et al., 2008), where the row and column covariances

---

*Corresponding author: javarika@gmail.com

typically represent the spatial and temporal dimensions, respectively. The separable covariance structure in such settings implies that the pattern of temporal variations remains the same across all spatial scales and frequencies, or locations. However, this restriction may be unrealistic for various practical spatio-temporal processes, as a result of which a considerable effort has gone into formulating non-separable models for multivariate random fields.

Let $X \in \mathbb{C}^{p \times n}$ represent a random matrix with a separable covariance structure, i.e. $X = A^{\frac{1}{2}} Z B^{\frac{1}{2}}$ for some positive-semidefinite matrices $A \in \mathbb{C}^{p \times p}, B \in \mathbb{C}^{n \times n}$ and an innovation matrix $Z \in \mathbb{C}^{p \times n}$ with independent entries having zero mean and unit variance. The data matrix $X$ may be viewed as a slice of a spatio-temporal process by thinking the rows of the data matrix $X$ as representing spatial coordinates and the columns indicating (regularly spaced) time points. Here, $A, B$ represent the population variance in the spatial and temporal components, respectively. The variance of the vectorized form of the data matrix, i.e. $vec(X)$ is given by $\mathrm{Var}(vec(X)) = B \otimes A$, where $\otimes$ denotes the Kronecker product. The eigenvalues of $\mathrm{Var}(vec(X))$ are $\{a_i b_j; i \in [p], j \in [n]\}$ where $\{a_i\}_{i=1}^p, \{b_j\}_{j=1}^n$ are the eigenvalues of $A$ and $B$, respectively. This shows that the logarithms of eigenvalues of the population covariance of the data are sums of logarithms of eigenvalues of the corresponding spatial and temporal components.

Let us now consider a generalized model $X = \sum_{r=1}^K X_r$, with each $X_r \in \mathbb{C}^{p \times n}$ having a separable covariance structure. In particular, we assume the following representation: $X_r = A_r^{\frac{1}{2}} Z_r B_r^{\frac{1}{2}}$, where $A_r$ and $B_r$ are $p \times p$ and $n \times n$ positive-semidefinite matrices, and $Z_r$ are matrices with independent entries with zero mean and unit variance, for $r = 1, \ldots, K$. Due to independence across coordinates, the variance of the vectorized data matrix is of the form

$$\mathrm{Var}(vec(X)) = \sum_{r=1}^K B_r \otimes A_r.$$

Denote the eigenvalues of $A_r$ and $B_r$ as $\{a_i^{(r)}\}_{i=1}^p$ and $\{b_j^{(r)}\}_{j=1}^n$ , respectively. We further assume simultaneous diagonalizability among the $A_r$ and $B_r$ matrices separately, which means that $A_r A_s = A_s A_r$ and $B_r B_s = B_s B_r$, for all $1 \leq r, s \leq K$. Under this assumption, the eigenvalues of $\mathrm{Var}(vec(X))$ are $\{\sum_{r=1}^K a_i^{(r)} \sigma_j^{(r)}; i \in [p], j \in [n]\}$. This shows that a superposition of data with separable covariance structures leads to a spectrum that is a sum of tensor products of the spectra of spatial and temporal components. With a moderately large $K$, this model therefore represents much more complex spatio-temporal dependencies than that captured by a separable covariance model.

The proposed model represents a generalization of the separable covariance model commonly used in spatio-temporal statistical modeling such as Paul and Silverstein (2009), Constantinou et al. (2017), Bagchi and Dette (2020), Lu and Zimmerman (2005), Mitchell et al. (2006), Genton (2007), Genton and Kleiber (2015). Under the separable covariance model (i.e., when $K = 1$), the matrices $A_1$ and $B_1$ capture the spatial and temporal variability in the data, and the Kronecker product structure for the variance of $vec(X)$ implies that the spatial and temporal variability are independent of each other. In the case of Gaussian data, there is no loss of generality in assuming that the matrices $A_1$ and $B_1$ are diagonal with non-negative diagonal entries. In this case, it is obvious that the pattern of temporal variability is the same across spatial coordinates, up to multiplicative scale factors.

The extension to the data model under study, for the case $K > 1$, thus breaks away from the rather restrictive assumption of separability. Indeed, depending on the choice of the matrices $\{A_r\}$ and $\{B_r\}$, the model allows for significant space-time interactions. For example, if the eigenvalues of $A_r$ concentrate on disjoint intervals for distinct $r$, then the corresponding data exhibits different patterns of temporal variability at different spatial scales. This is a much more realistic scenario for spatio-temporal data than what a separable model can cover. Therefore, the analysis carried out here has potential applications in capturing phenomena in the analysis of such data, if those phenomena can be quantified through certain eigenvalue statistics. Furthermore, assuming that the proposed model provides a meaningful description of some spatio-temporal data, a natural question is the determination of the parameter $K$ (number of separable components). Finally, if the temporal dependence is assumed to be stationary, with then one may also be interested in estimating the corresponding spectral density function, and/or the distribution of eigenvalues of the spatial covariances. The work presented here provides a mathematical framework for addressing such questions.

We study the asymptotic properties of the spectra of the sample covariance matrix $n^{-1}XX^*$ and its dual $p^{-1}X^*X$ under the stated model, as the ratio $p/n$ converges to a finite, positive constant. Under this asymptotic regime, we establish the existence of an L.S.D. of the sample covariance matrix. We further show that the Stieltjes transform of the L.S.D. of the covariance matrices of such matrices can be expressed as a system of integral equations with a unique solution. This work therefore can be viewed as a stepping stone towards developing statistical inference tools for a class of spatio-temporal processes that constitute a generalization of the separable covariance model.

We would like to point out that, with this exact same data structure, Wang and Xiang (2026) have analyzed the spectral behavior of renormalized sample covariance matrix $C_n$ defined as

$$C_n := \sqrt{\frac{p}{n}} \Big( \sum_{r,s=1}^{K} p^{-1} B_r^{\frac{1}{2}} Z_r^* A_r^{\frac{1}{2}} A_s^{\frac{1}{2}} Z_s B_s^{\frac{1}{2}} - \delta_{[r=s]} p^{-1} \operatorname{tr}(A_r) B_r \Big).$$

Under the ultra-high dimensional regime, i.e., $p/n \to \infty$, they have established that the L.S.D. of $C_n$ admits a Wigner-type behavior similar to that in Wang and Paul (2014). As a further remark, we note that some work related to estimation of the separable components, in the context of functional data analysis, has been done by Masak et al. (2023). A corresponding extension of the estimation procedure, with appropriate structural assumptions, in the context of high-dimensional spatio-temporal data, may benefit from the work presented here. This is beyond the scope of this paper.

The rest of the paper is organized as follows. Section 2 presents a few established results which will be utilized by us. Section 3 shows the model setup and preliminaries. Section 4 contains our main contribution which is summarized in Theorem 4.1. The next section highlights a few properties through Theorem 5.1, Corollary 5.1 and Theorem 5.2. Section 6 contains a few special cases of the main result. In Theorem 7.1 of Section 7, we show that the conclusions of our main result continue to hold under a few conditions which are a relaxation of the ones presented in Theorem 4.1. Section 8 contains some numerical simulations to validate our result.

## 2 Review of some relevant results

In this section we will state (without proof) a few established results which will be utilized throughout the paper. We state below Theorem 1 of Geronimo and Hill (2003) which characterizes the weak limit of a sequence of probability distribution functions in terms of their Stieltjes transforms.

**Theorem 2.1.** *Suppose that $(P_n)$ are real Borel probability measures with Stieltjes transforms $(S_n)$ respectively. If $\lim_{n\to\infty} S_n(z) = S(z)$ for all $z$ with $\Im(z) > 0$, then there exists a Borel probability measure $P$ with Stieltjes transform $S_P = S$ if and only if*

$$\lim_{y\to\infty} \mathring{\imath} y S(\mathring{\imath} y) = -1$$

*in which case $P_n \to P$ in distribution.*

Lemma B.18 of Bai and Silverstein (2009) establishes the following inequality between the Levy and uniform distance between two probability distribution functions.

$$L(F, G) \le ||F - G||_\infty. \tag{1}$$

The **Grommer-Hamburger Theorem** from Goh and Wimp (1997) is an important result which states a condition for the limit of Stieltjes transforms to be a Stieltjes transform.

**Theorem 2.2.** *Let $\{\mu_n\}_{n=1}^\infty$ be a sequence of measures in $\mathbb{R}$ for which the total variation is uniformly bounded.*

1. *If $\mu_n \xrightarrow{d} \mu$, then $S(\mu_n; z) \to S(\mu; z)$ uniformly on compact subsets of $\mathbb{C}\backslash\mathbb{R}$.*

2. *If $S(\mu_n; z) \to S(z)$ uniformly on compact subsets of $\mathbb{C}\backslash\mathbb{R}$, then $S(z)$ is the Stieltjes transform of a measure on $\mathbb{R}$ and $\mu_n \xrightarrow{d} \mu$.*

The **Vitali-Porter Theorem** from Section 2.4 of Schiff (1993) is an essential tool that is useful particularly in situations where we need to establish convergence of a sequence of analytic functions (on $\mathbb{C}_+$) but are able to do it only when the imaginary component is large.

**Theorem 2.3.** *Let $\{f_n\}_{n=1}^\infty$ be a locally uniformly bounded sequence of analytic functions in a domain $\Omega$ such that $\lim_{n\to\infty} f_n(z)$ exists for each $z$ belonging to a set $E \subset \Omega$ which has an accumulation point in $\Omega$. Then $\{f_n\}_{n=1}^\infty$ converges uniformly on compact subsets of $\Omega$ to an analytic function.*

The **Earle Hamilton Theorem** from Harris (2003) related to unique fixed points is of special interest to us since our main result involves existence of a unique solution to a system of integral equations.

**Theorem 2.4.** *Let $\mathcal{D}$ be a nonempty domain in a complex Banach space $X$ and let $h : \mathcal{D} \to \mathcal{D}$ be a bounded holomorphic function. If $h(\mathcal{D})$ lies strictly inside $\mathcal{D}$, then $h$ has a unique fixed point in $\mathcal{D}$.*

# 3 Model and preliminaries

For $n \in \mathbb{N}$, we use the symbol $[n]$ to denote the set $\{1, \ldots, n\}$. $A_{\cdot j}$ refers to the $j^{th}$ column of the matrix $A$. Let $K \in \mathbb{N}$ be fixed. For each $r \in [K]$, suppose $\{Z_n^{(r)}\}_{n=1}^{\infty}$ are sequences of complex random matrices, each having dimension $p \times n$ such that $p/n \to c \in (0, \infty)$. The entries have zero mean, unit variance and *satisfy some higher order moment conditions* to be stated explicitly later. Let $\{A_n^{(r)} \in \mathbb{C}^{p \times p}; r \in [K]\}_{n=1}^{\infty}$ and $\{B_n^{(r)} \in \mathbb{C}^{n \times n}; r \in [K]\}_{n=1}^{\infty}$ be sequences of random positive semi-definite matrices that commute among themselves and satisfy some mild limiting conditions. We are interested in the limiting behavior (as $p, n \to \infty$) of the ESDs of matrices of the type

$$S_n = \frac{1}{n} X_n X_n^*; \text{ where } X_n = \sum_{r=1}^{K} X_n^{(r)}, \tag{2}$$
$$\text{and, } X_n^{(r)} = U_n^{(r)} Z_n^{(r)} V_n^{(r)}; U_n^{(r)} = (A_n^{(r)})^{\frac{1}{2}}; V_n^{(r)} = (B_n^{(r)})^{\frac{1}{2}}.$$

We use the method of Stieltjes Transforms to arrive at the non-random L.S.D. of such matrices under certain conditions. We define the following central objects of our analysis. Henceforth, the $Z_n^{(r)}$ matrices will be referred to as innovation matrices and $A_n^{(r)}, B_n^{(r)}$ as scaling matrices.

**Definition 3.1.** *For pairwise commuting p.s.d. matrices $M_1, \ldots, M_K \in \mathbb{C}^{p \times p}$, let $P$ be a unitary matrix such that $M_r = P D_r P^*$ where $D_r = \text{diag}(\lambda_1^{(r)}, \ldots, \lambda_p^{(r)})$. For $j \in [p]$, let $\boldsymbol{\lambda}_j := \{\lambda_j^{(1)}, \ldots, \lambda_j^{(K)}\}_{j=1}^{p}$, i.e. $\boldsymbol{\lambda}_j$ is the $K$-tuple consisting of the $j^{th}$ eigenvalue (see Remark 3.1) from each of the $K$ coordinates. Let $\boldsymbol{M} := (M_1, \ldots, M_K)$. The Joint Empirical Spectral Distribution (JESD) of $\boldsymbol{M}$ is the probability measure on $\mathbb{R}_+^K$ that assigns equal mass to $\boldsymbol{\lambda}_j; j \in [p]$, i.e.*

$$\text{JESD}(\boldsymbol{M}) = \frac{1}{p} \sum_{j=1}^{p} \delta_{\boldsymbol{\lambda}_j}. \tag{3}$$

*Let $\boldsymbol{A}_n := (A_n^{(1)}, \ldots, A_n^{(K)})$ and $\boldsymbol{B}_n := (B_n^{(1)}, \ldots, B_n^{(K)})$. Since these matrices commute among themselves, we define their JESDs below, similar to (3):*

$$H_n := \text{JESD}(\boldsymbol{A}_n); \quad G_n := \text{JESD}(\boldsymbol{B}_n). \tag{4}$$

**Remark 3.1.** *Note that the choice of the unitary matrix $P$ in the spectral decomposition of the $K$ matrices is not unique. However, once we fix a $P$, the order of the $p$ eigenvalues within each of the $K$ coordinate gets fixed. But we observe that $\text{JESD}(\boldsymbol{M})$ is independent of the choice of $P$ and is therefore well-defined.*

**Definition 3.2.** *The dual of the Sample Covariance Matrix is defined as*

$$\tilde{S}_n := \frac{1}{n} X_n^* X_n. \tag{5}$$

**Definition 3.3.** $S_{nj} := \frac{1}{n} \sum_{r \neq j} X_{\cdot r} X_{\cdot r}^*$ *for $j \in [n]$.*

Additionally, for $z \in \mathbb{C}_+ := \{u + \mathrm{i}v : v > 0, u \in \mathbb{R}\}$, we define the following.

**Definition 3.4.** $Q(z) := (S_n - zI_p)^{-1}$ is the resolvent of $S_n$.

**Definition 3.5.** $Q_{-j}(z) := (S_{nj} - zI_p)^{-1}$ is the resolvent of $S_{nj}$ for $j \in [n]$.

**Definition 3.6.** $\tilde{Q}(z) := (\tilde{S}_n - zI_n)^{-1}$ is the resolvent of $\tilde{S}_n$.

**Definition 3.7.** $s_n(z) := \frac{1}{p} \operatorname{trace}(Q(z))$ is the Stieltjes Transform of $F^{S_n}$.

**Definition 3.8.** $\boldsymbol{h}_n(z) := (h_{1n}(z), \ldots, h_{Kn}(z))^T$; $h_{rn}(z) := \frac{1}{p} \operatorname{trace}\{A_n^{(r)} Q(z)\}, r \in [K]$.

**Definition 3.9.** $\boldsymbol{g}_n(z) := (g_{1n}(z), \ldots, g_{Kn}(z))^T$; $g_{rn}(z) := \frac{1}{p} \operatorname{trace}\{B_n^{(r)} \tilde{Q}(z)\}, r \in [K]$.

**Remark 3.2.** *Note that, $h_{rn}(\cdot)$ is the Stieltjes Transform of a measure. To see this, let $S_n = P_n \Lambda_n P_n^*$ be a spectral decomposition of $S_n$ and let $D_n^{(r)} = P_n^* A_n^{(r)} P_n$. Denote the diagonal elements of $D_n^{(r)}$ by $d_{jj}^{(r)}$ and denote the (real) eigenvalues of $S_n$ by $\lambda_j$ for $j \in [p]$. Then, observe that*

$$
h_{rn}(z) = \frac{1}{p} \operatorname{trace}(A_n^{(r)} Q(z)) = \frac{1}{p} \operatorname{trace}(A_n^{(r)} P_n (\Lambda_n - zI_p)^{-1} P_n^*) \tag{6}
$$

$$
= \frac{1}{p} \operatorname{trace}(D_n^{(r)} (\Lambda_n - zI_p)^{-1}) = \frac{1}{p} \sum_{j=1}^{p} \frac{d_{jj}^{(r)}}{\lambda_j - z}.
$$

*The RHS is the Stieltjes Transform of the measure that allocates a mass of $d_{jj}^{(r)}/p$ to the point $\lambda_j$. Similar results hold also for $g_{rn}(\cdot)$.*

## 4 L.S.D. under commutativity of scaling matrices

For a probability measure $\mu$ on $\mathbb{R}_+^K$, $z \in \mathbb{C}_+$ and $\mathbf{p} \in \mathbb{C}_+^K$, we define the following complex valued vector function:

$$
\mathbf{O}(z, \mathbf{p}, \mu) := \int \frac{\boldsymbol{\lambda} \mu(d\boldsymbol{\lambda})}{-z(1 + \boldsymbol{\lambda}^T \mathbf{p})}. \tag{7}
$$

Here, $\boldsymbol{\lambda}$ represents an arbitrary point in $\mathbb{R}_+^K$. $O_r(z, \mathbf{p}, \mu)$ is the $r^{th}$ coordinate of $\mathbf{O}(z, \mathbf{p}, \mu)$.

**Theorem 4.1.** *Suppose the following conditions hold.*

**T1** $c_n := p/n \to c \in (0, \infty)$.

**T2** *The innovation matrices $Z_n^{(r)}$ are independent across $r \in [K]$. Within each $r$, the entries of $Z_n^{(r)}$ are independent, have zero mean, unit variance and for some $\eta_0 > 0$, they satisfy the condition below:*

$$
\sup_{i,j,r,n} \mathbb{E}|z_{ij}^{(r)}|^{2+\eta_0} \leq M_{2+\eta_0} < \infty.
$$

**T3** $A_n^{(r)} A_n^{(s)} = A_n^{(s)} A_n^{(r)}$ and $B_n^{(r)} B_n^{(s)} = B_n^{(s)} B_n^{(r)}$ for $r, s \in [K]$.

**T4** $\{A_n^{(r)}, B_n^{(r)} : r \in [K]\}_{n=1}^{\infty}$ are sequences of p.s.d. matrices such that their JESDs (defined in 4) satisfy $H_n \xrightarrow{d} H$ a.s. and $G_n \xrightarrow{d} G$ a.s. where $H, G$ are non-random probability distributions on $\mathbb{R}_+^K$ such that $H \neq \delta_{\mathbf{0}}$, $G \neq \delta_{\mathbf{0}}$ and none of the marginals of $H$ or $G$ is degenerate at $0$.

**T5** There exists a constant $C_0 > 0$ such that

$$\limsup_{n \to \infty} \max_{1 \leq r \leq K} \max \left\{ \frac{1}{p} \operatorname{trace}(A_n^{(r)}), \frac{1}{n} \operatorname{trace}(B_n^{(r)}) \right\} < C_0.$$

Then, $F^{S_n} \xrightarrow{d} F$ a.s. where $s_F$, the Stieltjes transform of $F$ at $z \in \mathbb{C}_+$ is given by

$$s_F(z) = \int_{\mathbb{R}_+^K} \frac{dH(\boldsymbol{\lambda})}{-z(1 + \boldsymbol{\lambda}^T \boldsymbol{O}(z, c\boldsymbol{h}(z), G))}, \tag{8}$$

where, $\boldsymbol{h}(z) = (h_1(z), \dots, h_K(z))^T \in \mathbb{C}_+^K$ is the unique solution within the class of Stieltjes transforms of measures and satisfies

$$\boldsymbol{h}(z) = \int_{\mathbb{R}_+^K} \frac{\boldsymbol{\lambda} dH(\boldsymbol{\lambda})}{-z(1 + \boldsymbol{\lambda}^T \boldsymbol{O}(z, c\boldsymbol{h}(z), G))}. \tag{9}$$

**Remark 4.1.** *In this paper, we deal only with the limiting spectral distribution of the class of matrices described in equation (2) of the manuscript. Results of the type "No eigenvalues outside the support" (variant of the Bai-Yin law) are beyond the scope of this paper. Establishing such results (e.g. Paul and Silverstein (2009)) requires a very careful analysis of the Stieltjes transform of the LSD near the real axis. In particular, higher order moment conditions are needed to uniformly control the fluctuations of the terms involved in the corresponding approximations across a growing number of complex arguments close to the real axis. Whereas for establishing LSD, we only need pointwise convergence results.*

**Corollary 4.1.** *Under **T1-T5** of Theorem 4.1, we have an alternate characterization for the Stieltjes Transform of the L.S.D. $F$ of $S_n$ given by*

$$s_F(z) = \frac{1}{c} \int \frac{dG(\boldsymbol{\theta})}{-z(1 + c\boldsymbol{\theta}^T \boldsymbol{O}(z, \boldsymbol{g}(z), H))} + \left( \frac{1}{c} - 1 \right) \frac{1}{z}, \tag{10}$$

*where $\boldsymbol{g}(z) = (g_1(z), \dots, g_K(z))^T \in \mathbb{C}_+^K$ is the unique solution within the class of Stieltjes transforms of measures and satisfies*

$$\boldsymbol{g}(z) = \int \frac{\boldsymbol{\theta} dG(\boldsymbol{\theta})}{-z(1 + c\boldsymbol{\theta}^T \boldsymbol{O}(z, \boldsymbol{g}(z), H))}. \tag{11}$$

*Moreover $\boldsymbol{h}$ (from 9) and $\boldsymbol{g}$ satisfy the below:*

$$\boldsymbol{g}(z) = \boldsymbol{O}(z, c\boldsymbol{h}(z), G); \quad \boldsymbol{h}(z) = \boldsymbol{O}(z, \boldsymbol{g}(z), H). \tag{12}$$

*Yet another characterization of the Stieltjes Transform of the L.S.D. is given by*

$$s_F(z) = \frac{1}{z}\left(\frac{1}{c} - 1\right) - \frac{1}{cz}\int \frac{dG(\boldsymbol{\theta})}{1 + c\boldsymbol{\theta}^T \boldsymbol{h}(z)} = -z^{-1} - \boldsymbol{h}^T(z)\boldsymbol{g}(z). \tag{13}$$

*Proof.* (12) is established through the proof of Theorem 4.1. Assuming (8), (9) hold, interchanging the roles of $(X, X^*)$, $(p, n)$ and utilizing the relationship between the ESDs of $XX^*/n$ and $X^*X/n$, (11) and (10) are immediate. Finally, (13) follows from (9), (12) and a few algebraic manipulations as follows:

$$z\sum_{r=1}^{K} h_r O_r = \int \frac{z - z - z\boldsymbol{\lambda}^T \mathbf{O}}{-z - z\boldsymbol{\lambda}^T \mathbf{O}} dH(\boldsymbol{\lambda}) \tag{14}$$

$$\implies \sum_{r=1}^{K} h_r(z) \int \frac{\theta_r dG(\boldsymbol{\theta})}{1 + c\boldsymbol{\theta}^T \mathbf{h}(z)} = zs_F(z) + 1, \text{ using (8)}$$

$$\implies s_F(z) = \frac{1}{z}\left(\frac{1}{c} - 1\right) - \frac{1}{cz}\int \frac{dG(\boldsymbol{\theta})}{1 + c\boldsymbol{\theta}^T \mathbf{h}(z)} = -z^{-1} - \mathbf{h}^T(z)\mathbf{g}(z).$$

$$\square$$

**Remark 4.2.** *The assumptions on the scaling matrices $A_n^{(r)}, B_n^{(r)}; r \in [K]$ hold in an almost sure sense. Moreover, $H_n, G_n$ (defined in 3) converge weakly to non-random limits $H$ and $G$ almost surely. We show that $F^{S_n}$ converges weakly to a non-random limit $F$ that depends on the scaling matrices only through their non-random limits $H$ and $G$. Therefore, we treat $\{A_n^{(r)}, B_n^{(r)} : r \in [K]\}_{n=1}^{\infty}$ as a non-random sequence.*

## 4.1 Sketch of the proof

First, we show that for all $z \in \mathbb{C}_+$, the system of equations spanned by (9), (11) and (12) can have at most one solution. This is done in Theorem 4.2. After this, we impose a set of assumptions on $A_n^{(r)}, B_n^{(r)}, Z_n^{(r)}$ similar to Zhang (2006). This acts as a stepping stone to prove the result under general conditions of Theorem 4.1. The assumptions are as follows.

### 4.1.1 Assumptions

**A1** There exists a constant $\tau > 0$ such that $\sup_{n \in \mathbb{N}} \max_{r \in [K]} \{||A_n^{(r)}||_{op}, ||B_n^{(r)}||_{op}\} \leq \tau$, and

**A2** For $r \in [K]$, $\mathbb{E}z_{ij}^{(k)} = 0, \mathbb{E}|z_{ij}^{(k)}|^2 = 1, |z_{ij}^{(k)}| \leq n^b$, for some $\frac{1}{2+\eta_0} < b < \frac{1}{2}$ and $\eta_0$ is as per **T2**.

Under **A1-A2**, the proof is done in the following steps.

1 In Theorem 4.3, we show that for any $r \in [K]$, $\{h_{rn}(z), g_{rn}(z)\}_{n=1}^{\infty}$ have convergent subsequences. Assuming both sets of scaling matrices are diagonal, we construct deterministic equivalents for the resolvent of the sample covariance matrix and its dual. Using these, we show that any subsequential limit of $\{h_{rn}(z), g_{rn}(z)\}_{n=1}^{\infty}$ satisfies (9), (11) and (12), thus establishing existence and uniqueness.

2 This unique solution $(\mathbf{h}^\infty(z), \mathbf{g}^\infty(z))$ when plugged into (8) gives a function $(s^\infty(z))$ which satisfies the necessary and sufficient condition for a Stieltjes Transform of a probability measure. Let $F$ be the probability distribution characterized by $s^\infty(z)$. Theorem 4.5 shows that $s_n(z) \xrightarrow{a.s.} s^\infty(z)$ and thus $F^{S_n} \xrightarrow{d} F$ a.s.

3 When the scaling matrices are not diagonal, we construct an analog of $S_n$ with i.i.d. standard Gaussian innovations which bypasses the issue due to rotational invariance property. We use Lindeberg's principle to show that the difference between expected Stieltjes transforms arising from Gaussian and non-Gaussian innovations converges to zero.

The treatment of Theorem 4.1 under general conditions will be continued in Theorem 4.7 which builds upon the results derived under **A1-A2**.

**Lemma 4.1.** *Let $c, H, G$ be as in Theorem 4.1. Let $z \in \mathbb{C}_+$ and $\mathbf{h}, \mathbf{g} \in \mathbb{C}_+^K$. Then for any $r \in [K]$, we have $\Im(zO_r(z, c\mathbf{h}, G)) \geq 0$ and $\Im(zO_r(z, \mathbf{g}, H)) \geq 0$.*

*Proof.* We observe that

$$\Im(zO_r(z, c\mathbf{h}, G)) = \int \frac{\theta_r \Im(\boldsymbol{\theta}^T \mathbf{h})}{|1 + c\boldsymbol{\theta}^T \mathbf{h}|^2} dG(\boldsymbol{\theta}) = \sum_{s=1}^K \Im(\mathbf{h}_s) \left( \int \frac{\theta_r \theta_s}{|1 + \boldsymbol{\theta}^T \mathbf{h}|^2} dG(\boldsymbol{\theta}) \right) \geq 0.$$

The other part follows similarly. $\qquad\square$

**Lemma 4.2.** *Let $z = u + \mathbb{i}v; v > 0$. Suppose $\mathbf{h}, \mathbf{g}$ satisfies (9) and (11). Then for $r \in [K]$, $|h_r(z)| \leq C_0/v$ and $|g_r(z)| \leq C_0/v$.*

*Proof.* Let $r \in [K]$ and $z \in \mathbb{C}_+$. Then, using Lemma 4.1 and **T5** of Theorem 4.1, we get

$$|h_r(z)| \leq \int \frac{\lambda_r dH(\boldsymbol{\lambda})}{|-z - z\boldsymbol{\lambda}^T \mathbf{O}(z, c\mathbf{h}(z), G)|} \leq \int \frac{\lambda_r dH(\boldsymbol{\lambda})}{|\Im(z + z\boldsymbol{\lambda}^T \mathbf{O}(z, c\mathbf{h}(z), G))|} \leq \int \frac{\lambda_r dH(\boldsymbol{\lambda})}{v} \leq \frac{C_0}{v}.$$

Similarly, we have $|g_r(z)| \leq C_0/v$. $\qquad\square$

To establish the uniqueness of solutions of (9) and (11), for a fixed $z \in \mathbb{C}_+$, we define the functionals, $\mathbf{P}_z$ and $\mathbf{Q}_z$ depending on $z$ as follows:

$$\mathbf{P}_z(\mathbf{h}) := \int \frac{\boldsymbol{\lambda} dH(\boldsymbol{\lambda})}{-z(1 + \boldsymbol{\lambda}^T \mathbf{O}(z, c\mathbf{h}, G))}; \quad \mathbf{Q}_z(\mathbf{g}) := \int \frac{\boldsymbol{\theta} dG(\boldsymbol{\theta})}{-z(1 + c\boldsymbol{\theta}^T \mathbf{O}(z, \mathbf{g}, H))}. \tag{15}$$

**Theorem 4.2.** *(Uniqueness): Under the conditions imposed on $H, G$ in Theorem 4.1, for each $z \in \mathbb{C}_+$, there exist unique $\mathbf{h}, \mathbf{g} \in \mathbb{C}_+^K$ such that $\mathbf{P}_z(\mathbf{h}) = \mathbf{h}$ and $\mathbf{Q}_z(\mathbf{g}) = \mathbf{g}$.*

*Proof.* Note that $\mathbf{P}_z(\mathbb{C}_+^K) \subset \mathbb{C}_+^K$ and $\mathbf{Q}_z(\mathbb{C}_+^K) \subset \mathbb{C}_+^K$ follow from Lemma 4.1. For fixed $z$, $\mathbf{P}_z(\mathbf{h}), \mathbf{Q}_z(\mathbf{g})$ are bounded holomorphic functions from Lemma 4.1, Lemma 4.2 and Lemma E.4. Thus $\mathbf{P}_z(\mathbb{C}_+^K)$ and $\mathbf{Q}_z(\mathbb{C}_+^K)$ are proper subsets of $\mathbb{C}_+^K$. Therefore, $\mathbf{P}_z, \mathbf{Q}_z$ satisfy all the conditions of Theorem 2.4 thus proving the result. $\qquad\square$

**Theorem 4.3.** *(Compact convergence): Fix $z \in \mathbb{C}_+$ and $r \in [K]$. Then any subsequence of $\mathcal{H}_r = \{h_{rn}(z) : n \in \mathbb{N}\}$ and $\mathcal{G}_r = \{g_{rn}(z) : n \in \mathbb{N}\}$ have further subsequences that converge uniformly in any compact subset of $\mathbb{C}_+$. Moreover, these limits are Stieltjes Transforms.*

*Proof.* By *Montel's Theorem* (Theorem 3.3 of Stein and Shakarchi (2003)), it suffices to show that $\{h_{rn}(\cdot); r \in [K]\}$ is locally uniformly bounded in every compact subset of $\mathbb{C}_+$. Let $M \subset \mathbb{C}_+$ be an arbitrary compact set. Then, define $v_0 := \inf\{\Im(z) : z \in M\}$. It is clear that $v_0 > 0$ by the compactness of $M$. By basic matrix inequalities, we have

$$|h_{rn}(z)| = \left|\frac{1}{p}\operatorname{trace}(A_n^{(r)}Q(z))\right| \le \frac{1}{p}|\operatorname{trace}(A_n^{(r)})| \times ||Q(z)||_{op} \le \frac{C_0}{\Im(z)} \le \frac{C_0}{v_0}. \tag{16}$$

So, any subsequence of $\{\mathbf{h}_n(\cdot)\}_{n=1}^{\infty}$ and $\{\mathbf{g}_n(\cdot)\}_{n=1}^{\infty}$ have further subsequences which converge uniformly in any arbitrary compact subset $M \subset \mathbb{C}_+$. From Remark 3.2 and Theorem 2.2, the subsequential limits themselves must also be Stieltjes Transforms. Note that said theorem is applicable as the total variation norms of the underlying measures are uniformly bounded due to **T5** of Theorem 4.1. The proof for $\mathcal{G}_r$ follows similarly. $\qquad\square$

**Theorem 4.4.** *(Deterministic equivalent): Under **A1-A2** and assuming $B_n^{(r)}$ are diagonal, for $z \in \mathbb{C}_+$, a deterministic equivalent for $Q(z)$ is given by*

$$\bar{Q}(z) = \left(-zI_p + \sum_{r=1}^{K} A_n^{(r)}\left(\frac{1}{n}\operatorname{trace}\{B_n^{(r)}[I_n + c_n\sum_{s=1}^{K}\mathbb{E}h_{sn}(z)B_n^{(s)}]^{-1}\}\right)\right)^{-1}. \tag{17}$$

**Definition 4.1.** *For $r \in [K]$, we define the following quantities that serve as deterministic approximations to $h_{rn}(z)$.*

$$\tilde{h}_{rn}(z) := \frac{1}{p}\operatorname{trace}\{A_n^{(r)}\bar{Q}(z)\} = \int \frac{\lambda_r dH_n(\boldsymbol{\lambda})}{-z(1 + \boldsymbol{\lambda}^T\boldsymbol{O}(z, c_n\mathbb{E}\boldsymbol{h}_n, G_n))}, \tag{18}$$

$$\bar{\bar{Q}}(z) := \left(-zI_p + \sum_{r=1}^{K}A_n^{(r)}\left(\frac{1}{n}\operatorname{trace}\{B_n^{(r)}[I_n + c_n\sum_{s=1}^{K}\tilde{h}_{sn}(z)B_n^{(s)}]^{-1}\}\right)\right)^{-1}, \tag{19}$$

$$\tilde{\tilde{h}}_{rn}(z) := \frac{1}{p}\operatorname{trace}\{A_n^{(r)}\bar{\bar{Q}}(z)\} = \int \frac{\lambda_r dH_n(\boldsymbol{\lambda})}{-z(1 + \boldsymbol{\lambda}^T\boldsymbol{O}(z, c_n\tilde{\boldsymbol{h}}_n, G_n))}. \tag{20}$$

**Remark 4.3.** *In the above definitions, the simplification of the tracial terms to the integral expressions is possible because of **T3** of Theorem 4.1.*

**Theorem 4.5.** *(Existence of solution under A1-A2): Under **A1-A2** and assuming $A_n^{(r)}, B_n^{(r)}$ are diagonal, for $z \in \mathbb{C}_+$, $r \in [K]$ we have*

*1 $h_{rn}(z) \xrightarrow{a.s.} h_r^{\infty}(z)$ and $g_{rn}(z) \xrightarrow{a.s.} g_r^{\infty}(z)$ where $h_r^{\infty}(\cdot), g_r^{\infty}(\cdot)$ are Stieltjes Transforms,*

*2 $\{h_r^{\infty}(z)\}_{r=1}^K, \{g_r^{\infty}(z)\}_{r=1}^K$ uniquely satisfy (9), (11) and (12),*

*3 $s_n(z) \xrightarrow{a.s.} s^{\infty}(z)$ where $s^{\infty}$ is as defined in (8), and*

*4 $s^{\infty}(\cdot)$ satisfies $\lim_{y \to +\infty} \mathrm{i}y s^{\infty}(\mathrm{i}y) = -1$.*

## 4.2 Extending to non-diagonal scaling matrices

Having established Theorem 4.1 assuming the scaling matrices $\{A_n^{(r)}, B_n^{(r)}; r \in [K]\}_{n=1}^{\infty}$ are diagonal, we now consider the broader case when these matrices satisfy **T3** of Theorem 4.1. For each $n \in \mathbb{N}$, there exists unitary matrices $P_n \in \mathbb{C}^{p \times p}$ and $Q_n \in \mathbb{C}^{n \times n}$ such that $A_n^{(r)} =$

$P_n\Pi_n^{(r)}P_n^*$; $B_n^{(r)} = Q_n\Delta_n^{(r)}Q_n^*$ where $\Pi_n^{(r)}, \Delta_n^{(r)}$ are real diagonal matrices of orders $p$ and $n$ respectively. For completeness, we also define $\Omega_n^{(r)} := (\Pi_n^{(r)})^{\frac{1}{2}}$ and $\Sigma_n^{(r)} := (\Delta_n^{(r)})^{\frac{1}{2}}$ so that $U_n^{(r)} = P_n^*\Omega_n^{(r)}P_n^*$ and $V_n^{(r)} = Q_n\Sigma_n^{(r)}Q_n^*$.

We construct an analog (say $T_n$) of $S_n$ where the innovations are i.i.d. Gaussian, which makes them distribution-invariant under rotation. Let $W_n^{(r)} \in \mathbb{C}^{p\times n}$ be a sequence of matrices whose entries are i.i.d. complex standard Gaussian random variables. Replacing $Z_n^{(r)}$ with $W_n^{(r)}$ throughout in (2), let $Y_n := \sum_{r=1}^K U_n^{(r)}W_n^{(r)}V_n^{(r)}$ and noting that $P_n^*W_n^{(r)}Q_n \overset{d}{=} W_n^{(r)}$, we have

$$T_n := \frac{1}{n}Y_nY_n^* = \frac{1}{n}\sum_{r,s=1}^K P_n\Omega_n^{(r)}(P_n^*W_n^{(r)}Q_n)\Sigma_n^{(r)}\Sigma_n^{(s)}(Q_n^*(W_n^{(s)})^*P_n)\Omega_n^{(s)}P_n^* \qquad (21)$$

$$\overset{d}{=} P_n\Big(\frac{1}{n}\sum_{r,s=1}^K \Omega_n^{(r)}W_n^{(r)}\Sigma_n^{(r)}\Sigma_n^{(s)}(W_n^{(s)})^*\Omega_n^{(s)}\Big)P_n^*.$$

For this step, we observe that we can without loss of generality, assume that the entries of $W_n^{(r)}$ are truncated at $n^b$ where $b$ is defined in **A2**. To see this, we define $\hat{T}_n$ as the analogous covariance matrix constructed with $\hat{W}_n^{(r)}$ instead of $W_n^{(r)}$. Here, $\hat{W}_n^{(r)}$ is obtained from $W_n^{(r)}$ by truncating the entries at $n^b$. So, we have

$$\hat{T}_n := P_n\Big(\frac{1}{n}\sum_{r,s=1}^K \Omega_n^{(r)}\hat{W}_n^{(r)}\Sigma_n^{(r)}\Sigma_n^{(s)}(\hat{W}_n^{(s)})^*\Omega_n^{(s)}\Big)P_n^*. \qquad (22)$$

Note that $F^{T_n} = F^{P_n^*T_nP_n}$. From Section D.2, we have

$$||F^{P_n^*T_nP_n} - F^{P_n^*\hat{T}_nP_n}|| \overset{a.s.}{\longrightarrow} 0.$$

Therefore, using the fact that $F^{P_n^*\hat{T}_nP_n} = F^{\hat{T}_n}$ gives us $||F^{T_n} - F^{\hat{T}_n}|| \overset{a.s.}{\longrightarrow} 0$. Here, we remark that we could not have shown this directly, i.e. simply replacing $W_n^{(r)}$ with $\hat{W}_n^{(r)}$ in (21) would not have worked since $P_n^*\hat{W}_n^{(r)}Q_n \overset{d}{=} \hat{W}_n^{(r)}$ does not hold as $\hat{W}_n^{(r)}$ is not spherically symmetric.

In Theorem 4.6 below, we use Lindeberg's principle to show that $\hat{T}_n$ and $\hat{S}_n$ approach the same weak limit (in expectation) under **A1-A2**.

**Definition 4.2.** *Let $N = 2Kpn$. For $r \in [K]$, $\mathcal{R}_r, \mathcal{I}_r$ are operators from $\mathbb{R}^N \to \mathbb{R}^{p\times n}$ that construct $2K$ many real matrices from a real vector $\boldsymbol{m}$. $\mathcal{X}_r(\cdot)$ is an operator from $\mathbb{R}^N \to \mathbb{C}^{p\times n}$ that merges pairs of real matrices to form a complex matrix as defined below. In light of Remark 4.2, $U_n^{(r)}, V_n^{(r)}$ will be assumed to be fixed.*

$$\mathcal{R}_r(\boldsymbol{m}) := [m_{(2r-2)pn+1}, \ldots, m_{(2r-1)pn}]_{p\times n}; \quad \mathcal{I}_r(\boldsymbol{m}) := [m_{(2r-1)pn+1}, \ldots, m_{2rpn}]_{p\times n}.$$

$$\mathcal{X}_r(\boldsymbol{m}) := U_n^{(r)}\Big(\mathcal{R}_r(\boldsymbol{m}) + \mathtt{i}\mathcal{I}_r(\boldsymbol{m})\Big)V_n^{(r)}; \quad \mathcal{X}(\boldsymbol{m}) := \sum_{r=1}^K \mathcal{X}_r(\boldsymbol{m}); \quad \mathcal{S}(\boldsymbol{m}) := \frac{1}{n}\mathcal{X}(\boldsymbol{m})\mathcal{X}(\boldsymbol{m})^*.$$

**Theorem 4.6.** *With $\hat{S}_n, \hat{T}_n$ as defined in (24), let $s_n(\cdot), t_n(\cdot)$ represent the respective Stieltjes transforms of their ESDs. Then for $z \in \mathbb{C}_+$, we have*

$$|\mathbb{E}\hat{t}_n(z) - \mathbb{E}\hat{s}_n(z)| \to 0.$$

The proof can be found in Section C.1.

## 4.3 Existence of Solution under general conditions

Finally we generalize the result to the case when $H$ and $G$ have potentially unbounded support on $\mathbb{R}_+^K$ and the innovations satisfy **T2** of Theorem 4.1. Consider the random vector $\mathbf{z} = (z_1, \ldots, z_K) \sim H$. For $\tau > 0$, define $H^\tau(t)$ to be the distribution function of $\mathbf{z}^\tau = (z_1^\tau, \ldots, z_K^\tau)$ where $z_r^\tau = z_r \mathbb{1}_{\{z_r \leq \tau\}} + \tau \mathbb{1}_{\{z_r > \tau\}}$ for $r \in [K]$. By Theorem 4.5, for every $\tau > 0$, there exists unique $(\mathbf{h}^\tau, \mathbf{g}^\tau)$ that completely characterize the L.S.D. in terms of integral equations involving $H^\tau$ and $G^\tau$ as described in Theorem 4.1. Does $(\mathbf{h}^\tau, \mathbf{g}^\tau)$ achieve a meaningful limit as $\tau \to \infty$? The next theorem answers this question.

**Theorem 4.7.** *(Existence of solution – general): Under the conditions of Theorem 4.1, for $z \in \mathbb{C}_+$, $r \in [K]$, we have*

1 $h_r^\tau(z) \xrightarrow{a.s.} h_r^\infty(z)$ *and* $g_r^\tau(z) \xrightarrow{a.s.} g_r^\infty(z)$ *where* $h_r^\infty(\cdot), g_r^\infty(\cdot)$ *are Stieltjes Transforms,*

2 $\{h_r^\infty(z)\}_{r=1}^K, \{g_r^\infty(z)\}_{r=1}^K$ *uniquely satisfy (9), (11) and (12),*

3 $s_n(z) \xrightarrow{a.s.} s^\infty(z)$ *where* $s^\infty$ *is as defined in (8), and*

4 $s^\infty(\cdot)$ *satisfies* $\lim_{y \to +\infty} \mathrm{i} y s^\infty(\mathrm{i} y) = -1$.

*Proof.* In this section, we will prove (2)-(4) of Section 4.1 under the general conditions of Theorem 4.1. We create a sequence of matrices that are asymptotically *similar* in distribution to $\{S_n\}_{n=1}^\infty$ for large $n$ but satisfying properties that are amenable to our analysis. The below steps give an outline of the proof with the essential details split into individual modules.

Step1 For a p.s.d. matrix $S$ and a fixed $\tau > 0$, let $S(\tau)$ represent the matrix obtained by replacing all eigenvalues of $S$ greater than $\tau$ with $\tau$ in its spectral decomposition. Noting the definition of JESD (3), fixing $\tau > 0$, and letting $n \to \infty$, we must have

$$H_n^\tau := F^{\mathbf{A}_n(\tau)} \xrightarrow{d} H^\tau \quad G_n^\tau := F^{\mathbf{B}_n(\tau)} \xrightarrow{d} G^\tau.$$

Note that $F^{\mathbf{A}_n} = F^{\mathbf{\Pi}_n}$ and $F^{\mathbf{B}_n} = F^{\mathbf{\Delta}_n}$ because of **T3**. Therefore, we also have,

$$F^{\mathbf{\Pi}_n(\tau)} \xrightarrow{d} H^\tau \quad F^{\mathbf{\Delta}_n(\tau)} \xrightarrow{d} G^\tau.$$

However, we will choose $\tau > 0$ such that $(\tau, \ldots, \tau)$ is a continuity point of $H$.

Step2 Recall the definitions of $S_n$ (2) and $T_n$ (21). Next, we construct sequences $S_n^\tau, T_n^\tau$ by spectral truncation at $\tau > 0$.

$$S_n^\tau := \frac{1}{n} X_n^\tau (X_n^\tau)^*, \text{ where } X_n^\tau = \sum_{r=1}^K U_n^{(r)}(\tau) Z_n^{(r)} V_n^{(r)}(\tau), \text{ and} \tag{23}$$

$$T_n^\tau := \frac{1}{n} Y_n^\tau (Y_n^\tau)^*, \text{ where } Y_n^\tau = \sum_{r=1}^K \Omega_n^{(r)}(\tau) W_n^{(r)} \Sigma_n^{(r)}(\tau).$$

Step3 **Truncation:** With $a$ as in **A2**, define $\hat{Z}_n^{(r)} := ((\hat{z}_{ij}^{(r)}))$ with $\hat{z}_{ij}^{(r)} = z_{ij}^{(r)} \mathbb{1}_{\{|z_{ij}^{(r)}| \le n^a\}}$. Similarly, define $\hat{W}_n^{(r)} := ((\hat{w}_{ij}^{(r)}))$ with $\hat{w}_{ij}^{(r)} = w_{ij}^{(r)} \mathbb{1}_{\{|w_{ij}^{(r)}| \le n^a\}}$. Now, let

$$\hat{S}_n := \frac{1}{n}\hat{X}_n\hat{X}_n^*, \text{ where } \hat{X}_n = \sum_{r=1}^{K} U_n^{(r)}(\tau)\hat{Z}_n^{(r)}V_n^{(r)}(\tau), \text{ and} \tag{24}$$

$$\hat{T}_n := \frac{1}{n}\hat{Y}_n\hat{Y}_n^*, \text{ where } \hat{Y}_n = \sum_{r=1}^{K} \Omega_n^{(r)}(\tau)\hat{W}_n^{(r)}\Sigma_n^{(r)}(\tau).$$

Step4 **Centering:** Let $\tilde{X}_n = \hat{X}_n - \mathbb{E}\hat{X}_n$ and $\tilde{Y}_n := \hat{Y}_n - \mathbb{E}\hat{Y}_n$.

Step5 **Rescaling:** $\ddot{W}_n^{(r)} = (\ddot{w}_{ij}^{(r)}) = (\tilde{w}_{ij}^{(r)}/\rho_n)$ where $\rho_n^2 := \mathbb{E}|\tilde{w}_{11}^{(1)}|^2$. Let

$$\ddot{T}_n := \frac{1}{n}\ddot{Y}_n\ddot{Y}_n^*, \text{ where } \ddot{Y}_n = \sum_{r=1}^{K} U_n^{(r)}(\tau)\ddot{W}_n^{(r)}V_n^{(r)}(\tau). \tag{25}$$

Let $s_n(\cdot), s_n^\tau(\cdot), \hat{s}_n(\cdot), \hat{t}_n(\cdot), \tilde{t}_n(\cdot), \ddot{t}_n(\cdot)$ be the Stieltjes transforms of $F^{S_n}, F^{S_n^\tau}, F^{\hat{S}_n}, F^{\hat{T}_n}, F^{\tilde{T}_n}, F^{\ddot{T}_n}$ respectively.

Step6 Noting that $\ddot{T}_n$ satisfies **A1-A2**, by Theorem 4.5, we have $F^{\ddot{T}_n} \xrightarrow{a.s.} F^\tau$ for some probability distribution $F^\tau$ which is characterized by the triplet $(\mathbf{h}^\tau, \mathbf{g}^\tau, s^\tau)$ satisfying (8), (9) and (11) with $H^\tau$ instead of $H$. In particular, we have $|\ddot{t}_n(z) - s^\tau(z)| \xrightarrow{a.s.} 0$ by the same theorem. Note that $H$ is the weak limit of the JESD of $\{A_n^{(r)}; r \in [K]\}_{n=1}^\infty$ which are not necessarily diagonal matrices as was assumed in Theorem 4.5.

Step7 Next, we study the limiting behavior of $(\mathbf{h}^\tau, \mathbf{g}^\tau)$. The idea is to show that for any arbitrary subsequence of $\{\mathbf{h}^\tau\}$, there exists subsequential limits that are analytic and satisfy (9). By Theorem 4.2, all these subsequential limits must be the same, which we denote by $\mathbf{h}^\infty$. Therefore, $\mathbf{h}^\tau \to \mathbf{h}^\infty$ and by similar arguments, $\mathbf{g}^\tau \to \mathbf{g}^\infty$ as $\tau \to \infty$. Moreover, $(\mathbf{h}^\infty, \mathbf{g}^\infty)$ satisfy (12).

Step8 We derive $s^\infty$ from $\mathbf{h}^\infty$ using (8) and show that $s^\infty$ satisfies the conditions of Theorem 1 of Geronimo and Hill (2003). So, there exists some distribution $F^\infty$ corresponding to $s^\infty$. Moreover, we show that $s^\tau \to s^\infty$. **Step7** and **Step8** are shown in Appendix D concluding the proof of (1), (2) and (4).

Step9 For the proof of (3), we have the following inequality:

$$|s_n(z) - s^\infty(z)| \le |s_n(z) - s_n^\tau(z)| + |s_n^\tau(z) - \hat{s}_n(z)| + |\hat{s}_n(z) - \mathbb{E}\hat{s}_n(z)|$$
$$+ |\mathbb{E}\hat{s}_n(z) - \mathbb{E}\hat{t}_n(z)| + |\mathbb{E}\hat{t}_n(z) - \hat{t}_n(z)| + |\hat{t}_n(z) - \tilde{t}_n(z)|$$
$$+ |\tilde{t}_n(z) - \ddot{t}_n(z)| + |\ddot{t}_n(z) - s^\tau(z)| + |s^\tau(z) - s^\infty(z)|.$$

We will show that each term on the RHS goes to 0 as $n, \tau \to \infty$.

- $|\hat{s}_n(z) - \mathbb{E}\hat{s}_n(z)| \xrightarrow{a.s.} 0$, $|\hat{t}_n(z) - \mathbb{E}\hat{t}_n(z)| \xrightarrow{a.s.} 0$ follow from Lemma B.2.
- From Theorem 4.6, we have $|\mathbb{E}\hat{s}_n(z) - \mathbb{E}\hat{t}_n(z)| \to 0$.

From Appendix D.2 and using Lemma B.18 of Bai and Silverstein (2009), we have

- $L(F^{S_n}, F^{S_n^\tau}) \le ||F^{S_n} - F^{S_n^\tau}|| \xrightarrow{a.s.} 0,$

- $L(F^{S_n^\tau}, F^{\hat{S}_n}) \le ||F^{S_n^\tau} - F^{\hat{S}_n}|| \xrightarrow{a.s.} 0,$

- $L(F^{\hat{T}_n}, F^{\tilde{T}_n}) \le ||F^{\hat{T}_n} - F^{\tilde{T}_n}|| \xrightarrow{a.s.} 0,$ and

- $L(F^{\tilde{T}_n}, F^{\ddot{T}_n}) \xrightarrow{a.s.} 0.$

Applying Lemma A.2 on the above gives $|s_n(z) - s_n^\tau(z)| \xrightarrow{a.s.} 0$, $|s_n^\tau(z) - \hat{s}_n(z)| \xrightarrow{a.s.} 0$, $|\hat{t}_n(z) - \tilde{t}_n(z)| \xrightarrow{a.s.} 0$ and, $|\tilde{t}_n(z) - \ddot{t}_n(z)| \xrightarrow{a.s.} 0$ respectively. From **Step6**, we have $|\ddot{t}_n(z) - s^\tau(z)| \xrightarrow{a.s.} 0.$

Step10 Hence, $s_n(z) \xrightarrow{a.s.} s^\infty(z)$ which is a Stieltjes transform. Therefore, by Theorem 1 of Geronimo and Hill (2003), $F^{S_n} \xrightarrow{a.s.} F^\infty$ where $F^\infty$ is characterized by $(\mathbf{h}^\infty, \mathbf{g}^\infty, s^\infty)$ which satisfy (8), (9), (11) and (12). This concludes the proof.

$\square$

## 5 A few properties of the L.S.D.

**Theorem 5.1. (Point mass at 0):** *Let $F$ be the L.S.D. of $S_n$. Suppose $H$ and $G$ do not have any point mass at $\boldsymbol{0} \in \mathbb{R}_+^K$, the point mass of the L.S.D. ($F$) at $\boldsymbol{0}$ is given by*

$$F(\{\boldsymbol{0}\}) = \max\left\{0, 1 - \frac{1}{c}\right\}.$$

The proof has been done in Section F.2.

**Corollary 5.1. (Point Mass at 0)** *In addition to Theorem 5.1, if $H(\{\boldsymbol{0}\}) = 1 - \alpha$ and $G(\{\boldsymbol{0}\}) = 1 - \beta$ where $0 < \alpha, \beta \le 1$, the point mass of the L.S.D. ($F$) at $\boldsymbol{0}$ is given by*

$$F(\{\boldsymbol{0}\}) = \max\left\{1 - \alpha, 1 - \frac{\beta}{c}\right\}.$$

**Remark 5.1.** *See Figure 1 for an illustration of this result.*

*Proof.* We express $H$ and $G$ in terms of probability distributions $H_1, G_1$ over $\mathbb{R}_+^K$ that are free of any point mass at $\mathbf{0} \in \mathbb{R}^K$ as follows:

$$H = (1 - \alpha)\delta_{\mathbf{0}} + \alpha H_1; \quad G = (1 - \beta)\delta_{\mathbf{0}} + \beta G_1.$$

For a fixed $z \in \mathbb{C}_+$, let $(\mathbf{h}, \mathbf{g})$ be the unique solution of (12) corresponding to aspect ratio $c$ and population spectral distributions $H$ and $G$. Noting that $0 < \alpha, \beta \le 1$, we define $c_1 = c\alpha/\beta$, $\mathbf{h}_1 = \mathbf{h}\beta/\alpha$ and $\mathbf{g}_1 = \mathbf{g}$. Since $c\mathbf{h} = c_1\mathbf{h}_1$, we observe that

$$\mathbf{h}_1 = \int \frac{\boldsymbol{\lambda} dH_1(\boldsymbol{\lambda})}{-z/\beta + \boldsymbol{\lambda}^T \int \frac{\boldsymbol{\theta} dG_1(\boldsymbol{\theta})}{1 + c_1 \boldsymbol{\theta}^T \mathbf{h}_1}}; \quad \mathbf{g}_1 = \int \frac{\boldsymbol{\theta} dG_1(\boldsymbol{\theta})}{-z/\beta + c_1 \boldsymbol{\theta}^T \int \frac{\boldsymbol{\lambda} dH_1(\boldsymbol{\lambda})}{1 + \boldsymbol{\lambda}^T \mathbf{g}_1}}.$$

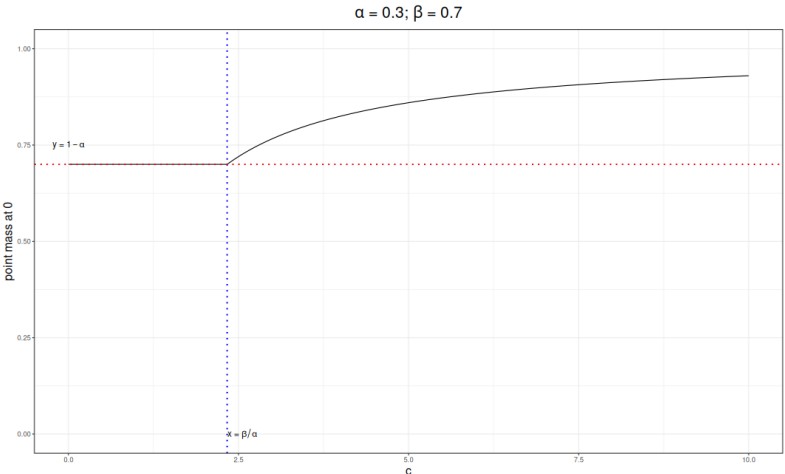

Figure 1: Point Mass of the L.S.D. at 0 as a function of $c$

Therefore we have the below equivalence relationships

$$\mathbf{h}(z, c, H, G) = \mathbf{h}(z/\beta, c_1, H_1, G_1) = \mathbf{h}_1; \quad \mathbf{g}(z, c, H, G) = \mathbf{g}(z/\beta, c_1, H_1, G_1) = \mathbf{g}_1. \tag{26}$$

Thus $(\mathbf{h}_1, \mathbf{g}_1)$ can be interpreted as the unique solution of (12) at $z/\beta$ corresponding to the modified aspect ratio $c_1 = c\alpha/\beta$ and population spectral distributions $H_1$ and $G_1$.

When $c_1 \geq 1$, By Lemma F.1, for some $r_0 \in [K]$, we have $\lim_{\epsilon\downarrow 0} h_{r_0}(\mathtt{i}\epsilon, c_1, H_1, G_1) = \infty$. Therefore, by (26), we have $\lim_{\epsilon\downarrow 0} h_{r_0}(\mathtt{i}\epsilon, c, H, G) = \lim_{\epsilon\downarrow 0} h_{r_0}(\mathtt{i}\epsilon/\beta, c_1, H_1, G_1) = \infty$. Note that for any $r \in [K]$, we have $\Re(h_r(\mathtt{i}\epsilon)) \geq 0$. For $\boldsymbol{\theta} \in \mathbb{R}_+^K$, we have

$$\left| \frac{1}{1 + c\boldsymbol{\theta}^T \mathbf{h}(\mathtt{i}\epsilon)} \right| \leq \frac{1}{\Re(1 + c\boldsymbol{\theta}^T \mathbf{h}(\mathtt{i}\epsilon))} \leq 1.$$

Finally applying DCT in (10), we have

$$-zs(z) = 1 - \frac{1}{c} + \frac{1}{c} \int \frac{dG(\boldsymbol{\theta})}{1 + c\boldsymbol{\theta}^T \mathbf{h}(z)} = 1 - \frac{1}{c} + \frac{1 - \beta}{c} + \frac{\beta}{c} \int \frac{dG_1(\boldsymbol{\theta})}{1 + c\boldsymbol{\theta}^T \mathbf{h}(z)}$$

$$\implies \lim_{\epsilon\downarrow 0} \mathtt{i}\epsilon s(\mathtt{i}\epsilon) = 1 - \frac{\beta}{c} + \lim_{\epsilon\downarrow 0} \int \frac{dG_1(\boldsymbol{\theta})}{1 + c\boldsymbol{\theta}^T \mathbf{h}(\mathtt{i}\epsilon)} = 1 - \frac{\beta}{c}.$$

Similarly when $c_1 \leq 1$ i.e., $c \leq \beta/\alpha$, by interchanging the roles of (H,G), $(\alpha, \beta)$ and $(p, n)$ the L.S.D. of the dual has a point mass at 0 amounting to $\bar{F}(\{0\}) = 1 - \frac{\alpha}{1/c} = 1 - c\alpha$. Now noting the relationship between the Stieltjes transforms of $F$ and $\bar{F}$, we have

$$s(z) = \left(1 - \frac{1}{c}\right)\frac{1}{-z} + \frac{1}{c}\bar{s}(z) \implies \lim_{\epsilon\downarrow 0}\left(-\mathtt{i}\epsilon s(-\mathtt{i}\epsilon)\right) = 1 - \frac{1}{c} + \frac{1}{c}\lim_{\epsilon\downarrow 0}\left(-\mathtt{i}\epsilon\bar{s}(\mathtt{i}\epsilon)\right)$$

$$= 1 - \frac{1}{c} + \frac{1}{c}(1 - c\alpha) = 1 - \alpha.$$

$\square$

**Theorem 5.2. (Continuity):** *For a fixed $z \in \mathbb{C}_+$, the unique solutions of (9) and (11) can be considered as functions of $H$ and $G$. For any $D_0 < \infty$, the restriction of these functions on the*

*set*

$$\mathcal{D}_K = \{\mu_K : \mu_K \text{ is a probability measure on } \mathbb{R}_+^K; \int \lambda_r^2 d\mu_K(\boldsymbol{\lambda}) \leq D_0; r \in [K]\}$$

*are continuous.*

The proof has been done in Section F.3.

# 6 A few Special Cases

In this section, we present a few special cases in which our main result reduces to simplified versions. The proofs follow directly from Theorem 4.1 itself and are omitted except for the last one. We start with the case where all the marginal population spectral distributions are functionally related.

**Corollary 6.1.** *Fix $\boldsymbol{a} = (a_1, \ldots, a_K)$ and $\boldsymbol{b} = (b_1, \ldots, b_K)$ where $a_r, b_r > 0$ for all $r \in [K]$. Let $H, G$ be supported on $\mathcal{S}_H = \{(a_1\lambda, \ldots, a_K\lambda) : \lambda \sim H_1\}$ and $\mathcal{S}_G = \{(b_1\theta, \ldots, b_K\theta) : \theta \sim G_1\}$ respectively for some uni-variate distributions $H_1, G_1$ on $\mathbb{R}_+$. Let $\tilde{h}, \tilde{g}$ be analytic functions on $\mathbb{C}^+$ uniquely satisfying the integral equations:*

$$\tilde{h}(z) = \int \frac{\lambda dH_1(\lambda)}{-z + \lambda \boldsymbol{a}^T \boldsymbol{b} \int \frac{\theta dG_1(\theta)}{1+c\theta \boldsymbol{a}^T \boldsymbol{b} \tilde{h}(z)}} \ , \qquad \tilde{g}(z) = \int \frac{\theta dG_1(\theta)}{-z + c\theta \boldsymbol{a}^T \boldsymbol{b} \int \frac{\lambda dH_1(\lambda)}{1+\lambda \boldsymbol{a}^T \mathbf{b} \tilde{g}(z)}}, \quad z \in \mathbb{C}^+.$$

*Then with $h_r(z) = a_r\tilde{h}(z)$, and $g_r(z) = b_r\tilde{g}(z)$, for $r \in [K]$, $(\boldsymbol{h}(z), \boldsymbol{g}(z))$ is the unique analytic solution on $\mathbb{C}^+$ of the system of integral equations describing the Stieltjes transform of the L.S.D. as per Theorem 4.1.*

This shows that in the case that the marginal population spectral distributions are scale multiples of each other, the L.S.D. of the sample covariance reduces to that under a separable covariance model.

**Corollary 6.2.** *In theorem 4.1, suppose for all $r \in [K]$, we have $A_n^{(r)} = A_n$ and $B_n^{(r)}$ follow Assumptions **T3** and **T4**. Note that $H$ reduces to a univariate distribution over $\mathbb{R}_+$ in this case. The Stieltjes transform $s_F$, of the L.S.D. at $z \in \mathbb{C}_+$ simplifies to the following form:*

$$s_F(z) = \int_{\mathbb{R}_+} \frac{dH(\lambda)}{-z(1 + \lambda \sum_{r=1}^K g_r(z))},$$

*where $g_r(z)$ is given by (11). In addition to the conditions on $A_n^{(r)}$, suppose $B_n^{(r)} = \beta_r I_n$ with $\beta_r > 0$ for all $r \in [K]$. The Stieltjes transform retains the above expression with the following simplification for $g_r(z)$:*

$$g_r(z) = \frac{\beta_r}{-z(1 + ch(z) \sum_{s=1}^K \beta_s)}, \text{ for all } r \in [K].$$

**Corollary 6.3.** *Suppose for $r \in [K], n \in \mathbb{N}$, we have $A_n^{(r)} = \alpha_r I_p$ and $B_n^{(r)} = \beta_r I_n$ where $\alpha_r, \beta_r > 0$. Let $\gamma = \boldsymbol{\alpha}^T \boldsymbol{\beta}$. Then the L.S.D. of $F^{S_n}$ at $z \in \mathbb{C}_+$ is characterized by the following*

*Stieltjes Transform:*

$$s(z) = \left( \frac{\gamma(1-c) - z}{2cz\gamma} + \frac{\sqrt{(z - \gamma(1+\sqrt{c})^2)(z - \gamma(1-\sqrt{c})^2)}}{2cz\gamma} \right).$$

*Further, if the innovation entries are i.i.d. standard Gaussian and $0 < c < 1$, we have*

$$\lambda_{min}(S_n) \xrightarrow{a.s.} \gamma(1 - \sqrt{c})^2.$$

*In the expression for $s(z)$, we take that branch of the square root which ensures that $\Im s(z) > 0$ whenever $\Im(z) > 0$.*

*Proof.* Let $z \in \mathbb{C}_+$ be arbitrary and the unique solution of (9) be $\mathbf{h}(z) \in \mathbb{C}_+^K$. Denoting $\tilde{h}_r(z) = h_r(z)/\alpha_r$, we find that $\tilde{h}_r(z) = \tilde{h}_s(z) = s(z), r \neq s \in [K]$. This is because

$$h_{rn}(z) = \frac{1}{p} \text{trace}(A_n^{(r)} Q(z)) = \frac{\alpha_r}{p} \text{trace}(Q(z)) = \alpha_r s_n(z).$$

Since $h_{rn}(z) \xrightarrow{a.s.} h_r(z)$ and $s_n(z) \xrightarrow{a.s.} s(z)$, we have $s(z) = h_r(z)/\alpha_r$ for any $r \in [K]$. Therefore, for any $r \in [K]$, (9) and (11) implies that

$$
\begin{aligned}
-z\tilde{h}_r &= \left( 1 + \frac{\boldsymbol{\alpha}^T\boldsymbol{\beta}}{-z(1 + c\sum_{s=1}^K \beta_s h_s)} \right)^{-1} \\
\implies \tilde{h}_r &= \left( -z + \frac{\boldsymbol{\alpha}^T\boldsymbol{\beta}}{1 + c\boldsymbol{\alpha}^T\boldsymbol{\beta}\tilde{h}_r} \right)^{-1} \\
\implies \tilde{h}_r &= \frac{1 + c\gamma\tilde{h}_r}{-z(1 + c\gamma\tilde{h}_r) + \gamma} \\
\implies c\gamma z\tilde{h}_r^2 &+ \tilde{h}_r(z + (c-1)\gamma) + 1 = 0 \\
\implies \tilde{h}_r &= \frac{\gamma(1-c) - z}{2c\gamma z} + \frac{\sqrt{z - z_+}\sqrt{z - z_-}}{2c\gamma z},
\end{aligned}
$$

where $z_\pm = \gamma(1 \pm \sqrt{c})^2$. For the second result, note that

$$X_n = \sum_{r=1}^K \sqrt{\alpha_r \beta_r} Z_n^{(r)} \stackrel{d}{=} \sqrt{\gamma} Z \implies S_n = \frac{1}{n} X_n X_n^* \stackrel{d}{=} \gamma \left( \frac{1}{n} Z Z^* \right),$$

where $Z \in \mathbb{C}^{p \times n}$ is another standard Gaussian matrix independent of $Z_n^{(r)}$. From Theorem 2 of Bai and Yin (1993), it is clear that $\lambda_{min}(S_n) \xrightarrow{a.s.} \gamma(1 - \sqrt{c})^2$. $\square$

# 7 Relaxation of pairwise commutativity

We present a set of conditions which are strictly weaker than **T3**, but are sufficient for Theorem 4.1 to hold without exact pairwise commutativity within $\{A_n^{(r)}\}_r$ and within $\{B_n^{(r)}\}_r$. For simplicity, in this section we shall refer to $A_n^{(r)}, B_n^{(r)}, Z_n^{(r)}$ as $A_r, B_r, Z_r$ respectively, for all $r \in [K]$. For the result below, we denote the nuclear norm of a matrix $S$ by $||S||_*$.

We shall use the suffix $r \to s$ to indicate that the eigenbasis of the $r^{th}$ coordinate will be imposed on the spectrum of the $s^{th}$ coordinate. For $r \in [K]$, we construct a $K-$tuple of pairwise

commuting matrices as follows:

$$
A_{r \to s} := \begin{cases} P_r D_s P_r^* & \text{, if } r \neq s, \\ A_r & \text{, if } r = s, \end{cases} \quad and \quad B_{r \to s} := \begin{cases} Q_r E_s Q_r^* & \text{, if } r \neq s, \\ B_r & \text{, if } r = s. \end{cases}
$$

Then for any fixed $s \in [K]$, $(A_r, A_{r \to s})$ and $(B_r, B_{r \to s})$ share the same eigenbasis for all $r \in [K]$. We shall use the following notations:

$$
\mathbf{A}_{r \to} := (A_{r \to 1}, \ldots, A_{r \to K}) \text{ and } \mathbf{B}_{r \to} := (B_{r \to 1}, \ldots, B_{r \to K}); \ r \in [K].
$$

Thus for any $r \in [K]$, the above construction ensures that all the $K$ matrices in $\mathbf{A}_{r \to}$ and in $\mathbf{A}_{r \to}$ satisfy pairwise commutativity. This in particular, allows us to define the JESDs (recall 3.1) of these K-tuples of matrices.

**Theorem 7.1.** *Instead of **T3** of Theorem 4.1, suppose any of the following conditions hold:*

**C1:** *For some $r_0, s_0 \in [K]$, $\max_{r \in [K]} \{\operatorname{rank}(A_r - A_{r_0 \to r}), \operatorname{rank}(B_r - B_{s_0 \to r})\} = o(p)$,*

**C2:** *For some $r_0, s_0 \in [K]$, $\max_{r \in [K]} \{\operatorname{rank}(P_r - P_{r_0}), \operatorname{rank}(Q_r - Q_{s_0})\} = o(p)$, and*

**C3:** *For some $r_0, s_0 \in [K]$, $\max_{r \in [K]} \{||A_r - A_{r_0 \to r}||_*, ||B_r - B_{s_0 \to r}||_*\} = o(p)$.*

*Let $H_n = \operatorname{JESD}(\mathbf{A}_{r_0 \to})$ and $G_n = \operatorname{JESD}(\mathbf{B}_{s_0 \to})$ and suppose $H_n \xrightarrow{d} H$; $G_n \xrightarrow{d} G$ and $H, G$ satisfy the conditions in **T4** of Theorem 4.1. Then, the conclusion of Theorem 4.1 holds.*

The proof has been done in Section F.1.

**Remark 7.1.** *As a generalization of the Householder construction for unitary matrices, let $P_r = I_p - 2U_r U_r^*$, where $U_r$ is a $p \times k_r$ matrix with orthonormal vectors as columns. If $\max_{r \in [K]} \{k_r\} = o(p)$, then $P_r$ satisfy **C2** of Theorem 7.1. Therefore, if the eigen bases of $A_n^{(r)}, B_n^{(r)}; r \in [K]$ are constructed as above separately, the result of Theorem 4.1 holds even without commutativity among $A_n^{(r)}$ and among $B_n^{(r)}$.*

# 8   Simulation Study

We repeated these simulations for $K = 1, 2, 3$ and multiple values of $c = 0.5, 1, 2.5$. Fixing $p = 1000$, we get $n = p/c$. For each value of $K$ and $c$, the following steps were performed.

   **Construction of $Z_n^{(r)}$:** The entries of $Z_n^{(r)} \in \mathbb{C}^{p \times n}$ are sampled as i.i.d. standard complex Gaussian entries (**Example 1**) and t-distribution with 3 degrees of freedom (**Example 2**).

   **Construction of $A_n^{(r)}$:** We construct diagonal matrices $D_n^{(r)} \in \mathbb{R}^{p \times p}$ with i.i.d. samples from an exponential distribution with scale parameter $r$. A common unitary matrix, $P_n \in \mathbb{C}^{p \times p}$ is used for all $r \in [K]$. Then, define $A_n^{(r)} := P_n D_n^{(r)} P_n^*$.

   **Construction of $B_n^{(r)}$:** Similarly, diagonal matrix $E_n^{(r)} \in \mathbb{R}^{n \times n}$ are constructed with i.i.d. samples from an exponential distribution with scale parameter $2r$. As before, a common unitary matrix, $Q_n \in \mathbb{C}^{n \times n}$ is used for all $r \in [K]$. Then, define $B_n^{(r)} := Q_n E_n^{(r)} Q_n^*$.

Then the sample covariance matrix, $S_n$ is constructed as per (2). We generate the histogram of the eigenvalues of $S_n$ and superimpose the numerical approximation of the theoretical density from the Stieltjes Transform as predicted in Theorem 4.1.

We observe that the spread of the spectrum is larger when the innovations are distributed as i.i.d. $t$ distribution with 3 degrees of freedom (Figure 3) as compared with the one with standard Gaussian innovations (Figure 2). The histograms suggest that the bulk of the ESD largely coincides with its theoretical counterpart. However, the edge of the empirical spectrum seems to spread significantly to the right. The latter phenomenon is possibly due to slower convergence of the edge because of lack of higher moments of the innovations.

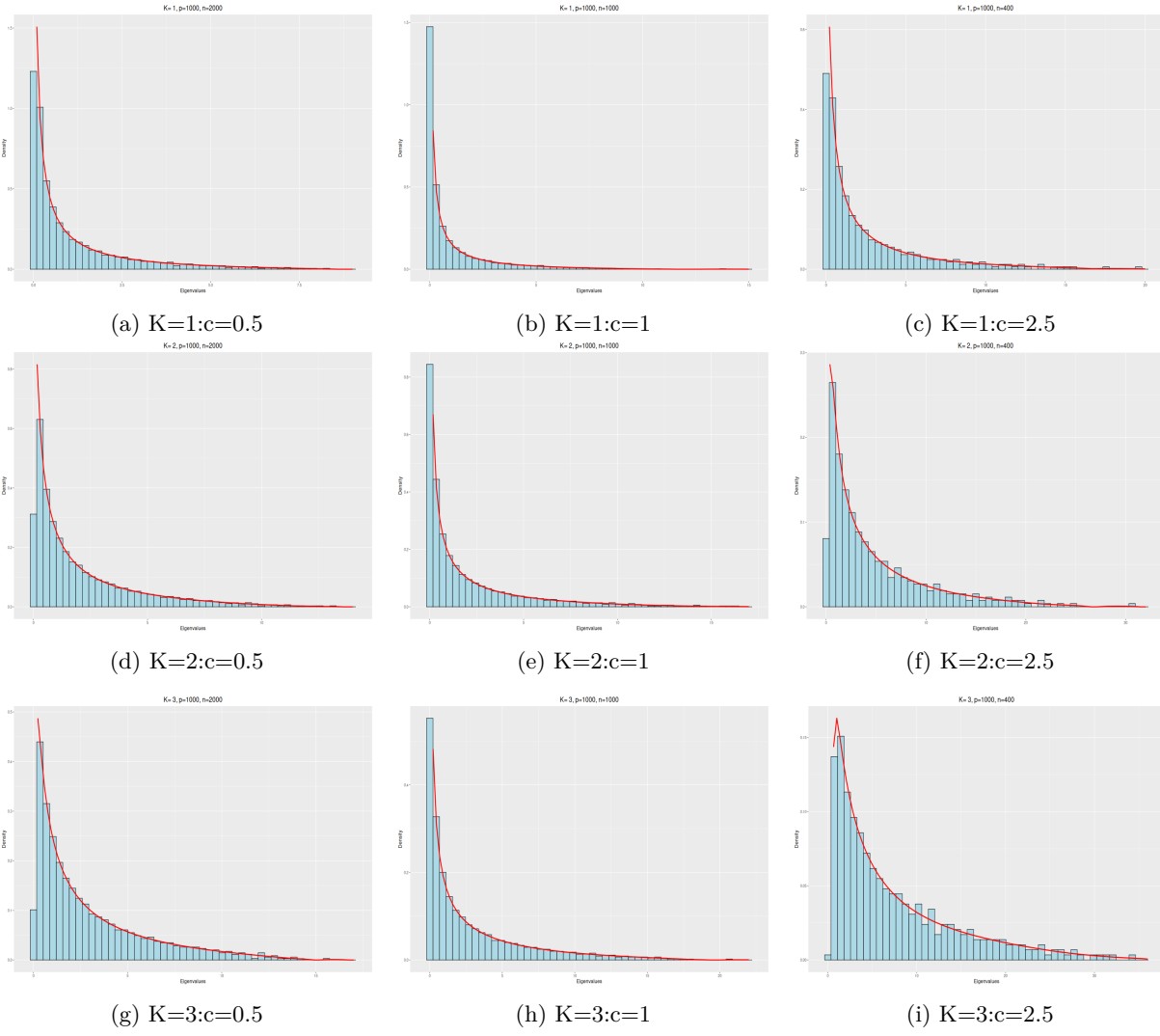

Figure 2: **Example 1**: Simulated vs. Theoretical limit distributions for various values of $K$ and $c$ with standard Gaussian innovations.

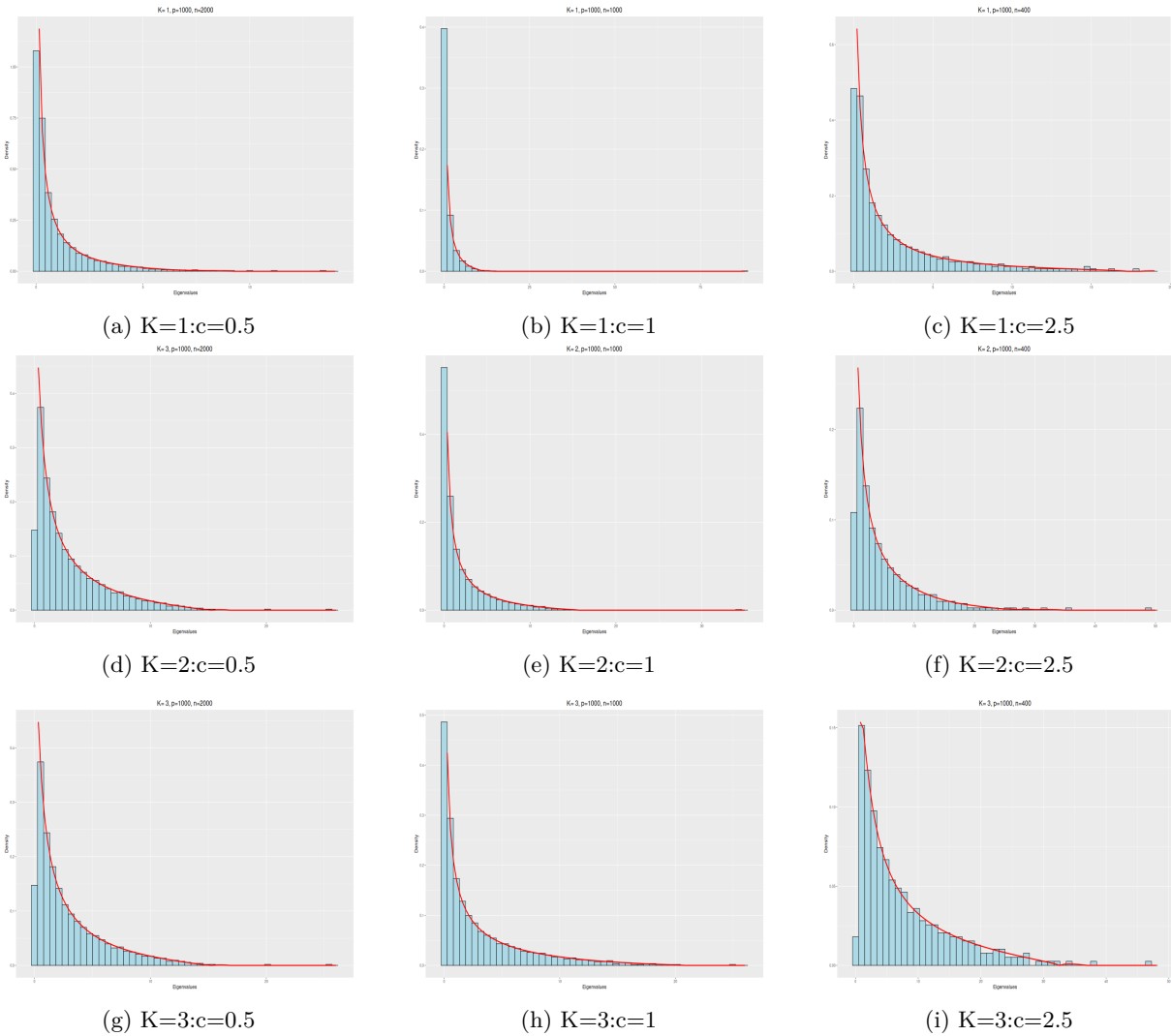

Figure 3: **Example 2**: Simulated vs. Theoretical limit distributions for various values of $K$ and $c$ when the innovations follow $t$-distribution with 3 degrees of freedom.

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

# A    Some basic results

**R0**: **Resolvent identity**:

$$A^{-1} - B^{-1} = A^{-1}(B - A)B^{-1} = B^{-1}(B - A)A^{-1}. \tag{27}$$

**R1**: For Hermitian matrices $A, B \in \mathbb{C}^{p \times p}$, by Lemma 2.4 of Silverstein and Bai (1995), we have

$$||F^A - F^B|| \leq \frac{1}{p} \operatorname{rank}(A - B). \tag{28}$$

**R2**: From Lemma 2.1 of Silverstein and Bai (1995), for a rectangular matrix, we have

$$\operatorname{rank}(A) \leq \sum_{i,j} \mathbb{1}_{\{a_{ij} \neq 0\}}. \tag{29}$$

**R3**: For rectangular matrices $A, B, P, Q, X$ of compatible dimensions, we have

$$\operatorname{rank}(AXB - PXQ) \leq \operatorname{rank}(A - P) + \operatorname{rank}(B - Q). \tag{30}$$

**R4**: For a p.s.d. matrix B and any square matrix A, we have

$$|\operatorname{trace}(AB)| \leq ||A||_{op} \operatorname{trace}(B). \tag{31}$$

**R5**: For $N \times N$ matrices $A, B$, we have

$$|\operatorname{trace}(AB)| \leq N ||A||_{op} ||B||_{op}. \tag{32}$$

**R6**: By Lemma B.18 of Bai and Silverstein (2009), we have

$$L(F, G) \leq ||F - G||_{\infty}. \tag{33}$$

**R7**: Let $A$ and $B$ be two $p \times n$ matrices and the ESDs of $S = AA^*$ and $\bar{S} = BB^*$ be denoted by $F^S$ and $F^{\bar{S}}$. Then, by Corollary A.42 of Bai and Silverstein (2009), we have

$$L^4(F^S, F^{\bar{S}}) \leq \frac{2}{p^2}(\operatorname{trace}(AA^* + BB^*))(\operatorname{trace}[(A - A^*)(B - B^*)]). \tag{34}$$

**R8**: Let $X = R + \mathbb{i}S$, where $R, S$ are real matrices of order $p \times n$. Then,

$$\frac{\partial AXB}{\partial r_{ij}} = A_{\cdot i}B_{j\cdot} \quad \text{and} \quad \frac{\partial AXB}{\partial s_{ij}} = \mathbb{i}A_{\cdot i}B_{j\cdot}. \tag{35}$$

**Lemma A.1.** *With $f$ defined in (72) and denoting $\partial_x$ instead of $\frac{d}{dx}$, we have*

$$\partial_x^3 f = \frac{1}{p} \operatorname{trace}\Big( -6\mathcal{Q}(\partial_x \mathcal{S})\mathcal{Q}(\partial_x \mathcal{S})\mathcal{Q}(\partial_x \mathcal{S})\mathcal{Q} + 3\mathcal{Q}(\partial_x \mathcal{S})\mathcal{Q}(\partial_x^2 \mathcal{S})\mathcal{Q} + 3\mathcal{Q}(\partial_x^2 \mathcal{S})\mathcal{Q}(\partial_x \mathcal{S})\mathcal{Q}\Big).$$

*Proof.* Following calculations done in Chatterjee (2006) and Supplementary Material of Liu et al. (2015), we differentiate both sides w.r.t. $x$ to get the following expression:

$$\mathcal{Q}(\mathcal{S} - zI) = I$$
$$\implies (\partial_x \mathcal{Q})\mathcal{S} + \mathcal{Q}\partial_x \mathcal{S} = z\partial_x \mathcal{Q}$$
$$\implies \mathcal{Q}(\partial_x \mathcal{S}) = -(\partial_x \mathcal{Q})(\mathcal{S} - zI)$$
$$\implies \partial_x \mathcal{Q} = -\mathcal{Q}(\partial_x \mathcal{S})\mathcal{Q}.$$

Differentiating the above w.r.t. $x$ again, we derive

$$\partial_x^2 \mathcal{Q} = 2\mathcal{Q}(\partial_x \mathcal{S})\mathcal{Q}(\partial_x \mathcal{S})\mathcal{Q} - \mathcal{Q}(\partial_x^2 \mathcal{S})\mathcal{Q}.$$

Differentiating once again and noting that $\partial_x^3 S = 0$, we derive the following expression:

$$\partial_x^3 \mathcal{Q} = -6\mathcal{Q}(\partial_x \mathcal{S})\mathcal{Q}(\partial_x \mathcal{S})\mathcal{Q}(\partial_x \mathcal{S})\mathcal{Q} + 3\mathcal{Q}(\partial_x \mathcal{S})\mathcal{Q}(\partial_x^2 \mathcal{S})\mathcal{Q} + 3\mathcal{Q}(\partial_x^2 \mathcal{S})\mathcal{Q}(\partial_x \mathcal{S})\mathcal{Q}. \tag{36}$$

This concludes the proof. $\square$

**Lemma A.2.** *Let $\{F_n, G_n\}_{n=1}^{\infty}$ be sequences of distribution functions on $\mathbb{R}$ with $s_{F_n}(z), s_{G_n}(z)$ denoting their respective Stieltjes transforms at $z \in \mathbb{C}_+$. If $L(F_n, G_n) \to 0$, then $|s_{F_n}(z) - s_{G_n}(z)| \to 0$.*

*Proof.* Let $\mathcal{P}(\mathbb{R})$ be the set of all probability distribution functions on $\mathbb{R}$. Then the bounded Lipschitz metric is defined as follows:

$$\beta : \mathcal{P}(\mathbb{R}) \times \mathcal{P}(\mathbb{R}) \to \mathbb{R}_+, \text{ where } \beta(F, G) := \sup\left\{ \left| \int h dF - \int h dG \right| : ||h||_{BL} \leq 1 \right\},$$

$$\text{and, } ||h||_{BL} = \sup\{|h(x)| : x \in \mathbb{R}\} + \sup_{x \neq y} \frac{|h(x) - h(y)|}{|x - y|}.$$

From Corollary 18.4 and Theorem 8.3 of Dudley (1976), we have the following relationship between Levy (L) and bounded Lipschitz ($\beta$) metrics:

$$\frac{1}{2}\beta(F, G) \leq L(F, G) \leq 3\sqrt{\beta(F, G)}. \tag{37}$$

Fix $z \in \mathbb{C}^R$ arbitrarily. Define $g_z(x) := (x - z)^{-1}$. Note that, $|g_z(x)| \leq 1/|\Im(z)| \ \forall x \in \mathbb{R}$. Therefore, we have

$$|g_z(x_1) - g_z(x_2)| = \left| \frac{1}{x_1 - z} - \frac{1}{x_2 - z} \right| = \frac{|x_1 - x_2|}{|x_1 - z||x_2 - z|} \leq \frac{|x_1 - x_2|}{\Im^2(z)}.$$

Note that, $||g_z||_{BL} \leq 1/|\Im(z)| + 1/\Im^2(z) < \infty$. Then, $||g||_{BL} = 1$ where, $g := g_z/||g_z||_{BL}$.

By (37) and (1), we have

$$L(F_n, G_n) \to 0 \longleftrightarrow \beta(F_n, G_n) \to 0$$

$$\implies \left| \int_{\mathbb{R}} g(x) dF_n(x) - \int_{\mathbb{R}} g(x) dG_n(x) \right| \to 0$$

$$\implies \left| \int_{\mathbb{R}} \frac{1}{x-z} dF_n(x) - \int_{\mathbb{R}} \frac{1}{x-z} dG_n(x) \right| \to 0$$

$$\implies |s_{F_n}(z) - s_{G_n}(z)| \longrightarrow 0.$$

$\square$

**Lemma A.3.** *Let $\{X_{jn}, Y_{jn} : 1 \leq j \leq n\}_{n=1}^{\infty}$ be triangular arrays of random variables. Suppose $\max_{1 \leq j \leq n} |X_{jn}| \xrightarrow{a.s.} 0$ and $\max_{1 \leq j \leq n} |Y_{jn}| \xrightarrow{a.s.} 0$. Then, $\max_{1 \leq j \leq n} |X_{jn} + Y_{jn}| \xrightarrow{a.s.} 0$.*

*Proof.* Let $A_x := \{\omega : \lim_{n \to \infty} \max_{1 \leq j \leq n} |X_{jn}(\omega)| = 0\}$, $A_y := \{\omega : \lim_{n \to \infty} \max_{1 \leq j \leq n} |Y_{jn}(\omega)| = 0\}$. Then $\mathbb{P}(A_x) = 1 = \mathbb{P}(A_y)$. Then, for all $\omega \in A_x \cap A_y$, we have $0 \leq |X_{jn}(\omega) + Y_{jn}(\omega)| \leq |X_{jn}(\omega)| + |Y_{jn}(\omega)|$. Hence, $\lim_{n \to \infty} \max_{1 \leq j \leq n} |X_{jn}(\omega) + Y_{jn}(\omega)| = 0$. But, $\mathbb{P}(A_x \cap A_y) = 1$. Therefore, the result follows. $\square$

**Lemma A.4.** *Let $\{A_{jn}, B_{jn}, C_{jn}, D_{jn} : 1 \leq j \leq n\}_{n=1}^{\infty}$ be triangular arrays of random variables. Suppose $\max_{1 \leq j \leq n} |A_{jn} - C_{jn}| \xrightarrow{a.s.} 0$ and $\max_{1 \leq j \leq n} |B_{jn} - D_{jn}| \xrightarrow{a.s.} 0$ and for some $B_1, B_2 \geq 0$, there exists $N_0 \in \mathbb{N}$ such that $|C_{jn}| \leq B_1$ a.s. and $|D_{jn}| \leq B_2$ a.s. when $n > N_0$. Then, $\max_{1 \leq j \leq n} |A_{jn} B_{jn} - C_{jn} D_{jn}| \xrightarrow{a.s.} 0$.*

*Proof.* Let $\Omega_1 = \{\omega : \lim_{n \to \infty} \max_{1 \leq j \leq n} |A_{jn}(\omega) - C_{jn}(\omega)| = 0\}$, $\Omega_2 = \{\omega : \lim_{n \to \infty} \max_{1 \leq j \leq n} |B_{jn}(\omega) - D_{jn}(\omega)| = 0\}$, $\Omega_3 = \{\omega : |C_{jn}(\omega)| \leq B_1 \text{ for } n > N_0\}$ and $\Omega_4 = \{\omega : |D_{jn}(\omega)| \leq B_2 \text{ for } n > N_0\}$. Then $\Omega_0 = \cap_{j=1}^{4} \Omega_j$ is a set of probability 1. Then, for all $\omega \in \Omega_0$, $\max_{1 \leq j \leq n} |B_{jn}(\omega)| \leq B_2$ eventually for large $n$. Therefore for $\omega \in \Omega_0$ and large $n$, we get the following:

$$\max_{1 \leq j \leq n} |A_{jn} B_{jn} - C_{jn} D_{jn}| \leq \max_{1 \leq j \leq n} |A_{jn} - C_{jn}||B_{jn}| + \max_{1 \leq j \leq n} |C_{jn}||B_{jn} - D_{jn}|$$

$$\leq B_2 \max_{1 \leq j \leq n} |A_{jn} - C_{jn}| + B_1 \max_{1 \leq j \leq n} |B_{jn} - D_{jn}| \xrightarrow{a.s.} 0.$$

$\square$

**Lemma A.5.** *Let $\{X_{jn}, Y_{jn} : 1 \leq j \leq n\}_{n=1}^{\infty}$ be triangular arrays of random variables such that $\max_{1 \leq j \leq n} |X_{jn} - Y_{jn}| \xrightarrow{a.s.} 0$. Then $|\frac{1}{n} \sum_{j=1}^{n} (X_{jn} - Y_{jn})| \xrightarrow{a.s.} 0$.*

*Proof.* Let $M_n := \max_{1 \leq j \leq n} |X_{jn} - Y_{jn}|$. We have,

$$\left| \frac{1}{n} \sum_{j=1}^{n} (X_{jn} - Y_{jn}) \right| \leq \frac{1}{n} \sum_{j=1}^{n} |X_{jn} - Y_{jn}| \leq M_n.$$

Let $\epsilon > 0$ be arbitrary. There exists $\Omega_0 \subset \Omega$ such that $\mathbb{P}(\Omega_0) = 1$ and for all $\omega \in \Omega_0$, we have $M_n(\omega) < \epsilon$ for sufficiently large $n \in \mathbb{N}$. Then,

$$\mathbb{P}\left( \{\omega : |\frac{1}{n} \sum_{j=1}^{n} (X_{jn} - Y_{jn})| < \epsilon\} \right) = 1.$$

Since $\epsilon > 0$ is arbitrary, the result follows. $\qquad\square$

We state the following result (Lemma B.26 of Bai and Silverstein (2009)) without proof.

**Lemma A.6.** *Let $A = (a_{ij})$ be an $n \times n$ non-random matrix and $x = (x_1, \ldots, x_n)^T$ be a vector of independent entries. Suppose $\mathbb{E}x_i = 0, \mathbb{E}|x_i|^2 = 1$, and $\mathbb{E}|x_i|^l \leq \nu_l$. Then for $k \geq 1$, there exists $C_k > 0$ independent of $n$ such that*

$$\mathbb{E}|x^*Ax - \mathrm{trace}(A)|^k \leq C_k\left((\nu_4 \, \mathrm{trace}(AA^*))^{\frac{k}{2}} + \nu_{2k} \, \mathrm{trace}\{(AA^*)^{\frac{k}{2}}\}\right).$$

For a deterministic matrix $A$ with $||A||_{op} < \infty$, let $B = A/||A||_{op}$. Then, $||B||_{op} = 1$ and by (32), $\mathrm{trace}(BB^*) \leq n||B||_{op}^2 = n$ and $\mathrm{trace}\{(BB^*)^{\frac{k}{2}}\} \leq n||B||_{op}^k = n$. Therefore, by Lemma A.6, we have

$$\mathbb{E}|x^*Bx - \mathrm{trace}(B)|^k \leq C_k\left((\nu_4 \, \mathrm{trace}(BB^*))^{\frac{k}{2}} + \nu_{2k} \, \mathrm{trace}\{(BB^*)^{\frac{k}{2}}\}\right) \qquad (38)$$

$$\implies \frac{\mathbb{E}|x^*Ax - \mathrm{trace}(A)|^k}{||A||_{op}^k} \leq C_k[(n\nu_4)^{\frac{k}{2}} + n\nu_{2k}]$$

$$\implies \mathbb{E}|x^*Ax - \mathrm{trace}(A)|^k \leq C_k||A||_{op}^k[(n\nu_4)^{\frac{k}{2}} + n\nu_{2k}].$$

We will be using this simplified form of the inequality going forward.

**Lemma A.7.** *Let $\{x_{jn} : j \in [n]\}_{n=1}^{\infty}$ be a triangular array of complex valued random vectors in $\mathbb{C}^p$ with independent entries. For $r \in [n]$, denote the $r^{th}$ element of $x_{jn}$ as $x_{jn}^{(r)}$. Suppose $\mathbb{E}x_{jn}^{(r)} = 0, \mathbb{E}|x_{jn}^{(r)}|^2 = 1$ and for $k \geq 1$ and $|x_{jn}| \leq n^b$ for some $0 < b < \frac{1}{2}$. Suppose $A_j \in \mathbb{C}^{p \times p}$ is independent of $x_{jn}$ and $\|A_j\|_{op} \leq B$ a.s. for some $B > 0$. Then,*

$$\max_{1 \leq j \leq n} \left| \frac{1}{n} x_{jn}^* A x_{jn} - \frac{1}{n} \operatorname{trace}(A_j) \right| \xrightarrow{a.s} 0.$$

*Proof.* We start by constructing the following bounds.

1. $\nu_4 := \sup_{j;n} \mathbb{E}|x_{jn}|^4 \leq \sup n^{2b} \mathbb{E}|x_{jn}|^2 = n^{2b}$.

2. In general, when $k \geq 2$, we similarly deduce that $\nu_{2k} = \sup_{j;n} \mathbb{E}|x_{jn}|^{2k} \leq n^{2b(k-1)}$.

For arbitrary $\delta > 0$ and $k \geq 1$, we have

$$
\begin{aligned}
p_n &:= \mathbb{P}\left( \max_{1 \leq j \leq n} \left| \frac{1}{n} x_{jn}^* A_j x_{jn} - \frac{1}{n} \operatorname{trace}(A_j) \right| > \delta \right) \\
&\leq \sum_{j=1}^{n} \mathbb{P}\left( \left| \frac{1}{n} x_{jn}^* A_j x_{jn} - \frac{1}{n} \operatorname{trace}(A_j) \right| > \delta \right), \text{ by union bound} \\
&\leq \sum_{j=1}^{n} \frac{\mathbb{E}\left| \frac{1}{n} x_{jn}^* A_j x_{jn} - \frac{1}{n} \operatorname{trace}(A_j) \right|^k}{\delta^k}, \text{ for any } k \in \mathbb{N} \\
&= \sum_{j=1}^{n} \frac{\mathbb{E}\left( \mathbb{E}\left[ |\frac{1}{n} x_{jn}^* A_j x_{jn} - \frac{1}{n} \operatorname{trace}(A_j)|^k \middle| A_j \right] \right)}{\delta^k} \\
&\leq \sum_{j=1}^{n} \frac{\mathbb{E}\|A_j\|_{op}^k C_k \left( (n\nu_4)^{\frac{k}{2}} + n\nu_{2k} \right)}{n^k \delta^k} \text{ by (38)} \\
&\leq \sum_{j=1}^{n} \frac{D_k \left[ (n^{1+2b})^{\frac{k}{2}} + n^{1+2b(k-1)} \right]}{n^k}, \text{ where } D_k = C_k \left( \frac{B}{\delta} \right)^k, \\
&= \frac{D_k}{n^{k(\frac{1}{2}-b)-1}} + \frac{D_k}{n^{k(1-2b)+2b-2}}.
\end{aligned}
$$

Since $b < 0.5$ and the above inequality holds for arbitrary $k \in \mathbb{N}$, we can choose $k \in \mathbb{N}$ large enough so that $\min\{k(0.5 - b) - 1, k(1 - 2b) + 2b - 2\} > 1$ to ensure that $\sum_{n=1}^{\infty} p_n$ converges. Therefore, by Borel Cantelli lemma, we have the result. $\square$

**Corollary A.1.** *Let $\{u_{jn}, v_{jn} : j \in [n]\}_{n=1}^{\infty}$ be triangular arrays and $A_j$ be complex matrices as in Lemma A.7 with $u_{jn}$ and $v_{jn}$ independent of each other. Then,*

$$\max_{1 \leq j \leq n} \left| \frac{1}{n} u_{jn}^* A_j v_{jn} \right| \xrightarrow{a.s.} 0.$$

*Proof.* Let $R_j(u,v) := \frac{1}{n} u_{jn}^* A_j v_{jn}$. Similarly, $R_j(v,v), R_j(u,u), R_j(v,u)$ are defined in the obvi-

ous manner. Let $x_{jn} = \frac{1}{\sqrt{2}}(u_{jn} + v_{jn})$. Now applying Lemma A.7, we get

$$\max_{1 \leq j \leq n} \left| \frac{1}{n} x_{jn}^* A_j x_{jn} - \frac{1}{n} \operatorname{trace}(A_j) \right| \xrightarrow{a.s.} 0 \tag{39}$$

$$\implies \max_{1 \leq j \leq n} \left| \frac{1}{2}(R_j(u,u) - T_j) + \frac{1}{2}(R_j(v,v) - T_j) + \frac{1}{2}(R_j(u,v) + R_j(v,u)) \right| \xrightarrow{a.s.} 0,$$

where $T_j := \frac{1}{n}\operatorname{trace}(A_j)$. Now setting $x_{jn} = \frac{1}{\sqrt{2}}(u_{jn} + \mathrm{i}v_{jn})$ and by Lemma A.7, we get

$$\max_{1 \leq j \leq n} \left| \frac{1}{n} x_{jn}^* A_j x_{jn} - \frac{1}{n} \operatorname{trace}(A_j) \right| \xrightarrow{a.s.} 0 \tag{40}$$

$$\implies \max_{1 \leq j \leq n} \left| \frac{1}{2}(R_j(u,u) - T_j) + \frac{1}{2}(R_j(v,v) - T_j) + \frac{1}{2}\mathrm{i}(R_j(u,v) - R_j(v,u)) \right| \xrightarrow{a.s.} 0.$$

Using Lemma A.3 on (39) and (40), we get $\max_{1 \leq j \leq n} |R_j(u,v)| \xrightarrow{a.s.} 0$. $\qquad\square$

# B  Existence of solution under A1-A2

**Lemma B.1.** *For $z \in \mathbb{C}_+$, $j \in [n]$ and a deterministic matrix $M \in \mathbb{C}^{p \times p}$, we have*

$$\operatorname{trace}\{M(Q(z) - Q_{-j}(z))\} \leq \frac{||M||_{op}}{\Im(z)}.$$

*Proof.* The proof is similar to that of Lemma 2.6 of Silverstein and Bai (1995). Note that the result holds even without **A1-A2**. $\qquad\square$

**Lemma B.2.** *Let $z \in \mathbb{C}_+$ and $r \in [K]$. Under **A1**, we have*

$$|h_{rn}(z) - \mathbb{E}h_{rn}(z)| \xrightarrow{a.s.} 0 \,, \quad |g_{rn}(z) - \mathbb{E}g_{rn}(z)| \xrightarrow{a.s.} 0 \,, \quad and \quad |s_n(z) - \mathbb{E}s_n(z)| \xrightarrow{a.s.} 0.$$

*Proof.* Define $\mathcal{F}_j = \sigma(\{X_{\cdot k} : j + 1 \leq k \leq n\})$ and for a measurable function $f$, we denote $\mathbb{E}_j f(X) := \mathbb{E}(f(X)|\mathcal{F}_j)$ for $0 \leq j \leq n - 1$ and $\mathbb{E}_n f(X) := \mathbb{E}f(X)$. For $r \in [K]$, we observe that

$$
\begin{aligned}
h_{rn}(z) - \mathbb{E}h_{rn}(z) &= \frac{1}{p}\operatorname{trace}(A_n^{(r)}Q(z)) - \mathbb{E}\left( \frac{1}{p}\operatorname{trace}(A_n^{(r)}Q(z)) \right) \\
&= \frac{1}{p}\sum_{j=1}^{n}(\mathbb{E}_{j-1} - \mathbb{E}_j)\operatorname{trace}(A_n^{(r)}Q(z)) \\
&= \frac{1}{p}\sum_{j=1}^{n}(\mathbb{E}_{j-1} - \mathbb{E}_j)\Big( \underbrace{\operatorname{trace}(A_n^{(r)}Q(z)) - \operatorname{trace}(A_n^{(r)}Q_{-j}(z))}_{:=Y_j} \Big) \\
&= \frac{1}{p}\sum_{j=1}^{n}\underbrace{(\mathbb{E}_{j-1} - \mathbb{E}_j)Y_j}_{:=D_j} = \frac{1}{p}\sum_{j=1}^{n}D_j.
\end{aligned}
$$

By Lemma B.1, we have $|Y_j| \leq \tau/\Im(z)$ for any $j \in [n]$. So, $|D_j| \leq 2\tau/\Im(z)$. By Lemma 2.12 of

Bai and Silverstein (2009), there exists $K_4$ depending only on $z \in \mathbb{C}_+$ such that

$$\mathbb{E}\,|h_{rn}(z) - \mathbb{E}h_{rn}(z)|^4 = \mathbb{E}\left|\frac{1}{p}\sum_{j=1}^{n} D_j\right|^4 \le \frac{K_4}{p^4}\mathbb{E}\left(\sum_{j=1}^{n}|D_j|^2\right)^2 \tag{41}$$

$$\le \frac{K_4}{p^4}\left(n\frac{4\tau^2}{\Im^2(z)}\right)^2 = \frac{K_4}{c_n^2 n^2}\frac{16\tau^4}{\Im^4(z)} \to 0.$$

By Borel Cantelli Lemma, we have $|h_{rn}(z) - \mathbb{E}h_{rn}(z)| \xrightarrow{a.s.} 0$. The second result follows similarly. For the last result,

$$s_n(z) - \mathbb{E}s_n(z) = \frac{1}{p}\sum_{j=1}^{p}(\mathbb{E}_{j-1} - \mathbb{E}_j)\Big(\operatorname{trace}(Q(z)) - \operatorname{trace}(Q_{-j}(z))\Big) = \frac{1}{p}\sum_{j=1}^{n} D_j.$$

By Lemma B.1, $|D_j| \le 2/\Im(z)$. Using the same arguments as above, the result follows. $\square$

**Lemma B.3.** *Let $z \in \mathbb{C}_+$ and $r \in [K]$. Under **A1-A2**, for sufficiently large $n$, we have*

$$\Im(h_{rn}(z)) \ge B_1(z) > 0 \text{ and } \Im(g_{rn}(z)) \ge B_2(z) > 0,$$

*where $B_i : \mathbb{C}_+ \to \mathbb{R}_+$ are deterministic functions.*

*Proof.* Fix $r \in [K]$ and Let $z = u + \mathrm{i}v$ with $v > 0$. Under **A1**, we have $||A_n^{(r)}||_{op} \le \tau$. Since $H_n$ and $H$ are compactly supported on (a subset of) $[0, \tau]^K$ and $H_n \xrightarrow{d} H$ a.s., we get

$$\int_{[0,\tau]^K} \lambda_r dH_n(\boldsymbol{\lambda}) \to \int_{[0,\tau]^K} \lambda_r dH(\boldsymbol{\lambda}); \quad r \in [K]. \tag{42}$$

Moreover, these limits must be positive since none of the marginals of $H$ is degenerate at 0. Therefore for large $n$, we have

$$\frac{1}{p}\operatorname{trace}(A_n^{(r)}) = \int_{[0,\tau]^K} \lambda_r dH_n(\boldsymbol{\lambda}) \to \int_{[0,\tau]^K} \lambda_r dH(\boldsymbol{\lambda}) > 0. \tag{43}$$

Recalling Remark 3.2, we have

$$h_{rn}(z) = \frac{1}{p}\sum_{j=1}^{p}\frac{d_{jj}^{(r)}}{\lambda_j - z},$$

where $\{\lambda_j\}_{j=1}^{p}$ are the real eigenvalues of $S_n$ and $d_{jj}^{(r)}$ are the diagonal elements of $D_n^{(r)} = P_n^* A_n^{(r)} P_n$ with $P_n \Lambda_n P_n^*$ being a spectral decomposition of $S_n$. For any $\delta > 0$, we see that

$$||S_n||_{op} = ||\frac{1}{n}X_n||_{op}^2 \tag{44}$$

$$\le \sum_{r=1}^{K}(||\frac{1}{n}U_n^{(r)}Z_n^{(r)}V_n^{(r)}||_{op})^2$$

$$\le K\sum_{r=1}^{K}(||U_n^{(r)}||_{op}||V_n^{(r)}||_{op}||Z_n^{(r)}||_{op})^2$$

$$\le 2K\tau(1 + \sqrt{c_n})^2 + \delta.$$

Let $B = 2K\tau(1 + \sqrt{c})^2$. Then $\mathbb{P}(|\lambda_j| > B \ \text{i.o.}) = 0$.

[1] Define $B^* := \begin{cases} -B \operatorname{sgn}(u) & \text{if } u \neq 0, \\ B & \text{if } u = 0. \end{cases}$

Then $(\lambda_j - u)^2 \leq (B^* - u)^2$. Therefore,

$$\Im(h_{rn}(z)) = \frac{1}{p} \sum_{j=1}^{p} \frac{d_{jj}^{(r)} v}{(\lambda_j - u)^2 + v^2}$$

$$\geq \frac{1}{p} \sum_{j=1}^{p} \frac{d_{jj}^{(r)} v}{(B^* - u)^2 + v^2}$$

$$= \frac{v}{(B^* - u)^2 + v^2} \left( \frac{1}{p} \sum_{j=1}^{p} d_{jj}^{(r)} \right)$$

$$= \frac{v}{(B^* - u)^2 + v^2} \left( \frac{1}{p} \operatorname{trace}(A_n^{(r)}) \right), \text{ as } \operatorname{trace}(D_n^{(r)}) = \operatorname{trace}(A_n^{(r)})$$

$$\longrightarrow \frac{v}{(B^* - u)^2 + v^2} \left( \int_{[0,\tau]^K} \lambda_r dH(\boldsymbol{\lambda}) \right) := M_r > 0 \text{ from (43)}.$$

Define $B_1(z) := \min_{r \in [K]} M_r$. This lower bound works for $\Im(h_{rn}(z))$ for all $r \in [K]$. The proof for $\Im(g_{rn}(z))$ is similar. $\qquad\square$

**Corollary B.1.** *Under **A1-A2** and assuming $A_n^{(r)}$ are diagonal, for $z \in \mathbb{C}_+$, a deterministic equivalent for $\tilde{Q}(z)$ is given by*

$$\bar{\bar{Q}}(z) = \left( -zI_n + c_n \sum_{r=1}^{K} B_n^{(r)} \left( \frac{1}{p} \operatorname{trace}\{A_n^{(r)}[I_p + \sum_{s=1}^{K} \mathbb{E}g_{sn}(z)A_n^{(s)}]^{-1}\} \right) \right)^{-1}. \tag{45}$$

**Remark B.1.** *The proof is similar to that of Theorem 4.4.*

**Definition B.1.** *Similar to (18-20), we define the following quantities:*

$$\tilde{g}_{rn}(z) := \frac{1}{n} \operatorname{trace}\{B_n^{(r)} \bar{\bar{Q}}(z)\} = \int \frac{\theta_r dG_n(\boldsymbol{\theta})}{-z(1 + c_n \boldsymbol{\theta}^T \boldsymbol{O}(z, \mathbb{E}\boldsymbol{g}_n, H_n))}, \tag{46}$$

$$\bar{\bar{\bar{Q}}}(z) := \left( -zI_n + c_n \sum_{r=1}^{K} B_n^{(r)} \left( \frac{1}{p} \operatorname{trace}\{A_n^{(r)}[I_p + \sum_{s=1}^{K} \tilde{g}_{sn}(z)A_n^{(s)}]^{-1}\} \right) \right)^{-1}, \tag{47}$$

$$\tilde{\tilde{g}}_{rn}(z) := \frac{1}{n} \operatorname{trace}\{B_n^{(r)} \bar{\bar{\bar{Q}}}(z)\} = \int \frac{\lambda_r dH_n(\boldsymbol{\lambda})}{-z(1 + c_n \boldsymbol{\theta}^T \boldsymbol{O}(z, \tilde{\boldsymbol{g}}_n, H_n))}. \tag{48}$$

**Lemma B.4.** *Let $\boldsymbol{h}_n, \boldsymbol{g}_n$ be as defined in (3.8) and (3.9) respectively and $H_n, G_n$ be as defined in (3). For each $z \in \mathbb{C}_+$, let $\{\boldsymbol{p}_n(z), \boldsymbol{q}_n(z) \in \mathbb{C}_+^K\}_{n=1}^{\infty}$ be sequences (deterministic or random) such that $||\boldsymbol{h}_n - \boldsymbol{p}_n||_1 \to 0$ a.s. and $||\boldsymbol{g}_n - \boldsymbol{q}_n||_1 \to 0$ a.s. Under **A1-A2**, we have the following results:*

*(1) $||\boldsymbol{O}(z, c_n\boldsymbol{h}_n(z), G_n) - \boldsymbol{O}(z, c_n\boldsymbol{p}_n(z), G_n)||_1 \xrightarrow{a.s} 0$, and*

*(2) $||\boldsymbol{O}(z, \boldsymbol{g}_n(z), H_n) - \boldsymbol{O}(z, \boldsymbol{q}_n(z), H_n)||_1 \xrightarrow{a.s.} 0$.*

*Further if $\boldsymbol{x}^{\infty}, \boldsymbol{y}^{\infty}$ are any subsequential limits of $\{\boldsymbol{h}_{n_k}, \boldsymbol{g}_{n_k}\}_{k=1}^{\infty}$ respectively and $H, G$ are as defined in Theorem 4.1, we have*

---

[1]$\operatorname{sgn}(x)$ is the sign function

(3) $||\boldsymbol{O}(z, c_{n_k}\boldsymbol{h}_{n_k}(z), G_{n_k}) - \boldsymbol{O}(z, c\boldsymbol{x}^\infty(z), G)||_1 \xrightarrow{a.s.} 0$, and

(4) $||\boldsymbol{O}(z, \boldsymbol{g}_{n_k}(z), H_{n_k}) - \boldsymbol{O}(z, \boldsymbol{y}^\infty(z), H)||_1 \xrightarrow{a.s.} 0$.

**Remark B.2.** *Here $\boldsymbol{h}_n(\cdot), \boldsymbol{p}_n(\cdot), \boldsymbol{x}^\infty(\cdot), \boldsymbol{y}^\infty(\cdot)$ are all (analytic) mappings from $\mathbb{C}_+^K$ to $\mathbb{C}_+^K$. So once we fix a $z \in \mathbb{C}_+$, we will almost exclusively refer to the complex vectors $\boldsymbol{h}_n(z), \boldsymbol{g}_n(z) \in \mathbb{C}_+^K$ by $\boldsymbol{h}_n, \boldsymbol{g}_n$ respectively unless stated otherwise. The same convention will be followed for $\tilde{\boldsymbol{h}}_n(\cdot), \tilde{\tilde{\boldsymbol{h}}}_n(\cdot), \tilde{\boldsymbol{g}}_n(\cdot), \tilde{\tilde{\boldsymbol{g}}}_n(\cdot)$ and their (subsequential) limits.*

*Proof.* Fix $z \in \mathbb{C}_+$. By Lemma B.3, for any $r \in [K]$, we $\Im(h_{rn}(z)) \geq B_1(z) > 0$ for sufficiently large $n$. Similarly for any $r \in [K]$, we also have $\Im(p_{rn}(z)) \geq B_1(z) > 0$ since $||\mathbf{h}_n - \mathbf{p}_n||_1 \xrightarrow{a.s.} 0$. Therefore, we see that

$$||\mathbf{O}(z, c_n\mathbf{h}_n, G_n) - \mathbf{O}(z, c_n\mathbf{p}_n, G_n)||_1$$

$$\leq \frac{1}{|z|} \sum_{r=1}^K \int \left| \frac{\theta_r}{1 + c_n\boldsymbol{\theta}^T\mathbf{h}_n} - \frac{\theta_r}{1 + c_n\boldsymbol{\theta}^T\mathbf{p}_n} \right| dG_n(\boldsymbol{\theta})$$

$$\leq \frac{1}{|z|} \sum_{r=1}^K \sum_{s=1}^K c_n|h_{sn} - p_{sn}| \int \left| \frac{\theta_r\theta_s}{(1 + c_n\boldsymbol{\theta}^T\mathbf{h}_n)(1 + c_n\boldsymbol{\theta}^T\mathbf{p}_n)} \right| dG_n(\boldsymbol{\theta})$$

$$\leq \frac{1}{|z|} \sum_{r=1}^K \sum_{s=1}^K c_n|h_{sn} - p_{sn}| \int \left| \frac{1}{c_n^2\Im(h_{rn})\Im(p_{sn})} \right| dG_n(\boldsymbol{\theta})$$

$$\leq \frac{1}{|z|} \frac{Kc_n}{c_n^2 B_1^2(z)} ||\mathbf{h}_n - \mathbf{p}_n||_1 \leq \frac{2K}{c|z|B_1^2(z)} ||\mathbf{h}_n - \mathbf{p}_n||_1.$$

The establishes the first result. The second result follows similarly from the second result of Lemma B.3.

For the third and fourth results, recall that Theorem 4.3 guarantees the existence of subsequential limits of $\{\mathbf{h}_n\}$ and $\{\mathbf{g}_n\}$. We observe that

$$||\mathbf{O}(z, c_{n_k}\mathbf{h}_{n_k}, G_{n_k}) - \mathbf{O}(z, c\mathbf{x}^\infty, G)||_1$$

$$\leq ||\mathbf{O}(z, c_{n_k}\mathbf{h}_{n_k}, G_{n_k}) - \mathbf{O}(z, c_{n_k}\mathbf{h}_{n_k}, G)||_1 + ||\mathbf{O}(z, c_{n_k}\mathbf{h}_{n_k}, G) - \mathbf{O}(z, c\mathbf{x}^\infty, G)||_1$$

$$\leq \frac{1}{|z|} \sum_{r=1}^K \left( \int \left| \frac{\theta_r}{1 + c_{n_k}\boldsymbol{\theta}^T\mathbf{h}_{n_k}} \right| d\{G_{n_k}(\boldsymbol{\theta}) - G(\boldsymbol{\theta})\} + \int \left| \frac{\theta_r}{1 + c_{n_k}\boldsymbol{\theta}^T\mathbf{h}_{n_k}} - \frac{\theta_r}{1 + c\boldsymbol{\theta}^T\mathbf{x}^\infty} \right| dG(\boldsymbol{\theta}) \right).$$

Since $||\mathbf{h}_{n_k} - \mathbf{x}^\infty||_1 \to 0$, we also have $\Im(x_r^\infty(z)) \geq B_1(z) > 0$. This leads to the following bounds on the integrands associated with the $r^{th}$ term of the above expression:

$$\left| \frac{\theta_r}{1 + c_{n_k}\boldsymbol{\theta}^T\mathbf{h}_{n_k}} \right| \leq \frac{1}{c_n\Im(h_{r,n_k})} \leq \frac{2}{cB_1(z)} \text{ and } \left| \frac{\theta_r}{1 + c\boldsymbol{\theta}^T\mathbf{x}^\infty} \right| \leq \frac{1}{c\Im(h_r)} \leq \frac{1}{cB_1(z)}.$$

Applying DCT and using the fact that $G_n \xrightarrow{d} G$, the third result follows. The fourth result follows similarly. $\square$

**Lemma B.5.** *Let $z \in \mathbb{C}_+$ and $r \in [K]$. Under **A1-A2**, we have*

$$|\tilde{h}_{rn}(z) - \tilde{\tilde{h}}_{rn}(z)| \to 0 \text{ and } |\tilde{g}_{rn}(z) - \tilde{\tilde{g}}_{rn}(z)| \to 0.$$

*Proof.* Let $r \in [K]$. From Theorem 4.4 and Lemma B.2, we have $||\tilde{\mathbf{h}}_n - \mathbb{E}\mathbf{h}_n|| \to 0$ as $n \to \infty$. Under **A1**, we have $\boldsymbol{\lambda} \in [0, \tau]^K$. By Lemma B.4 and the below string of inequalities the first result is immediate.

$$
\left| \tilde{h}_{rn}(z) - \tilde{\tilde{h}}_{rn}(z) \right| = \left| \int \frac{\lambda_r dH_n(\boldsymbol{\lambda})}{-z(1 + \boldsymbol{\lambda}^T \mathbf{O}(z, c_n \mathbb{E}\mathbf{h}_n, G_n))} - \int \frac{\lambda_r dH_n(\boldsymbol{\lambda})}{-z(1 + \boldsymbol{\lambda}^T \mathbf{O}(z, c_n \tilde{\mathbf{h}}_n, G_n))} \right|
$$

$$
\leq |z| \int \frac{\left| \lambda_r \boldsymbol{\lambda}^T (\mathbf{O}(z, c_n \mathbb{E}\mathbf{h}_n, G_n) - \mathbf{O}(z, c_n \tilde{\mathbf{h}}_n, G_n)) \right| dH_n(\boldsymbol{\lambda})}{| - z - z\boldsymbol{\lambda}^T \mathbf{O}(z, c_n \mathbb{E}\mathbf{h}_n, G_n)| \, | - z - z\boldsymbol{\lambda}^T \mathbf{O}(z, c_n \tilde{\mathbf{h}}_n, G_n)|}
$$

$$
\leq \frac{|z|}{v^2} \sum_{s=1}^{K} |O_s(z, c_n \mathbb{E}\mathbf{h}_n, G_n) - O_s(z, c_n \tilde{\mathbf{h}}_n, G_n)| \int \lambda_r \lambda_s dH_n(\boldsymbol{\lambda})
$$

$$
\leq \frac{|z|\tau^2}{v^2} ||\mathbf{O}(z, c_n \mathbb{E}\mathbf{h}_n, G_n) - \mathbf{O}(z, c_n \tilde{\mathbf{h}}_n, G_n))||_1 \to 0.
$$

The second result follows similarly. $\qquad\square$

**Lemma B.6.** *Uniform Convergence Results: Recall the definition of $E_j(r, s)$ and $F_j(r, s)$ from (51) and (52) respectively for $r, s \in [K]$. Under **A1-A2**, we have the following uniform convergence results.*

$$
\begin{cases}
\max_{1 \leq j \leq n} |F_j(r, r) - b_j^{(r)} m_{rn}| \xrightarrow{a.s.} 0; & \max_{1 \leq j \leq n} |F_j(r, s)| \xrightarrow{a.s.} 0, r \neq s; \\
\max_{1 \leq j \leq n} |E_j(r, r) - c_n b_j^{(r)} h_{rn}| \xrightarrow{a.s.} 0; & \max_{1 \leq j \leq n} |E_j(r, s)| \xrightarrow{a.s.} 0, r \neq s.
\end{cases}
$$

*Proof.* We begin with the following simplification of $E_j(r, s)$:

$$
E_j(r, s) = \frac{1}{n} (X_{\cdot j}^{(r)})^* Q_{-j} X_{\cdot j}^{(s)} = \frac{1}{n} b_j^{(r)} Z_{\cdot j} U_n^{(r)} Q_{-j} U_n^{(r)} Z_{\cdot j}^{(s)}.
$$

Recall that $U_n^{(r)} = (A_n^{(r)})^{\frac{1}{2}}$ and the fact that under **A1-A2**, we can use Lemma A.7 and Lemma B.1 to get

$$
\max_{1 \leq j \leq n} \left| E_j(r, r) - \frac{1}{n} b_j^{(r)} \operatorname{trace}(A_n^{(r)} Q_{-j}) \right| \xrightarrow{a.s.} 0
$$

$$
\implies \max_{1 \leq j \leq n} \left| E_j(r, r) - c_n b_j^{(r)} \left( \frac{1}{p} \operatorname{trace}(A_n^{(r)} Q_{-j}) \right) \right| \xrightarrow{a.s.} 0
$$

$$
\implies \max_{1 \leq j \leq n} \left| E_j(r, r) - c_n b_j^{(r)} \left( \frac{1}{p} \operatorname{trace}(A_n^{(r)} Q) \right) \right| \xrightarrow{a.s.} 0
$$

$$
\implies \max_{1 \leq j \leq n} \left| E_j(r, r) - c_n b_j^{(r)} h_{rn} \right| \xrightarrow{a.s.} 0.
$$

For $r \neq s$, $\max_{1 \leq j \leq n} |E_j(r, s)| \xrightarrow{a.s.} 0$ follows from Lemma A.1 and Lemma B.1. Consider the below expansion of $F_j(r, s)$:

$$
F_j(r, s) = \frac{1}{n} X_{\cdot j}^{(r)} \bar{Q} M_n Q_{-j} X_{\cdot j}^{(s)} = \frac{1}{n} \sqrt{b_j^{(r)} b_j^{(s)}} (Z_{\cdot j}^{(r)})^* U_n^{(r)} \bar{Q} M_n Q_{-j} U_n^{(s)} Z_{\cdot j}^{(s)}.
$$

By Lemma A.7 and Lemma B.1, we have

$$\max_{1 \le j \le n} \left| F_j(r,r) - \frac{1}{n} b_j^{(r)} \operatorname{trace}(A_n^{(r)} \bar{Q} M_n Q_{-j}) \right| \xrightarrow{a.s.} 0$$

$$\implies \max_{1 \le j \le n} \left| F_j(r,r) - \frac{1}{n} b_j^{(r)} \operatorname{trace}(A_n^{(r)} \bar{Q} M_n Q) \right| \xrightarrow{a.s.} 0$$

$$\implies \max_{1 \le j \le n} \left| F_j(r,r) - b_j^{(r)} m_{rn} \right| \xrightarrow{a.s.} 0.$$

The last result follows from Lemma A.1 and Lemma B.1. $\qquad\square$

## B.1 Proof of Theorem 4.4

*Proof.* Fix $z \in \mathbb{C}_+$. Define $F(z) := \left( \bar{Q}(z) \right)^{-1}$. We will use $Q, \bar{Q}, Q_{-j}$ for simplicity. From the resolvent identity (27), we have

$$Q - \bar{Q} = Q\left( F + zI_p - \frac{1}{n} \sum_{j=1}^n X_{\cdot j} X_{\cdot j}^* \right) \bar{Q}. \tag{49}$$

Using the above, we get

$$\frac{1}{p} \operatorname{trace}\{(Q - \bar{Q})M_n\} \tag{50}$$

$$= \frac{1}{p} \operatorname{trace}\{Q(F + zI_p)\bar{Q}M_n\} - \frac{1}{p} \operatorname{trace}\{Q\left( \frac{1}{n} \sum_{j=1}^n X_{\cdot j} X_{\cdot j}^* \right)\bar{Q}M_n\}$$

$$= \frac{1}{p} \operatorname{trace}\{(F + zI_p)\bar{Q}M_nQ\} - \frac{1}{p} \operatorname{trace}\{\left( \frac{1}{n} \sum_{j=1}^n X_{\cdot j} X_{\cdot j}^* \right)\bar{Q}M_nQ\}$$

$$= \underbrace{\frac{1}{p} \operatorname{trace}\{(F + zI_p)\bar{Q}M_nQ\}}_{Term_1} - \underbrace{\frac{1}{p} \sum_{j=1}^n \frac{1}{n} X_{\cdot j}^* \bar{Q}M_nQ X_{\cdot j}}_{Term_2}.$$

Since we have assumed that $B_n^{(r)}$ are diagonal, let $b_j^{(r)}$ represent the $j^{th}$ diagonal element of $B_n^{(r)}$. In particular, this means that the $j^{th}$ column of $X_n^{(r)}$ can be expressed as $X_{\cdot j}^{(r)} = (U_n^{(r)} Z_n^{(r)} V_n^{(r)})_{\cdot j} = \sqrt{b_j^{(r)}} U_n^{(r)} Z_n^{(r)}$. To establish $Term_1 - Term_2 \xrightarrow{a.s.} 0$, we define the following quantities.

For $j \in [n], r, s \in [K]$, define

$$E_j(r,s) := \frac{1}{n}(X_{\cdot j}^{(r)})^* Q_{-j} X_{\cdot j}^{(s)} = \frac{1}{n} \sqrt{b_j^{(r)} b_j^{(s)}} (Z_{\cdot j}^{(r)})^* U_n^{(r)} Q_{-j} U_n^{(s)} Z_{\cdot j}^{(s)}, \tag{51}$$

$$F_j(r,s) := \frac{1}{n}(X_{\cdot j}^{(r)}) * \bar{Q} M_n Q_{-j} X_j^{(s)} = \frac{1}{n} \sqrt{b_j^{(r)} b_j^{(s)}} (Z_{\cdot j}^{(r)})^* U_n^{(r)} \bar{Q} M_n Q_{-j} U_n^{(s)} Z_{\cdot j}^{(s)}, \tag{52}$$

$$m_{rn}(z) := \frac{1}{n} \operatorname{trace}\{A_n^{(r)} \bar{Q} M_n Q\} \text{ for } r \in [K]. \tag{53}$$

The following relationship between $Q$ and $Q_{-j}$ for $j \in [n]$ is a direct consequence of Wood-

bury's formula:

$$QX_{\cdot j} = \frac{Q_{-j}X_{\cdot j}}{1 + \frac{1}{n}X_{\cdot j}^* Q_{-j}X_{\cdot j}}. \tag{54}$$

Using (54) and noting that $X_{\cdot j} = \sum_{r=1}^{K} X_{\cdot j}^{(r)}$, $Term_2$ of (50) can be simplified as follows:

$$Term_2 = \frac{1}{p}\sum_{j=1}^{n}\frac{1}{n}X_{\cdot j}^* \bar{Q}M_n Q X_{\cdot j} = \frac{1}{p}\sum_{j=1}^{n}\frac{\frac{1}{n}X_{\cdot j}^* \bar{Q}M_n Q_{-j}X_{\cdot j}}{1 + \frac{1}{n}X_{\cdot j}^* Q_{-j}X_{\cdot j}} = \frac{1}{p}\sum_{j=1}^{n}\frac{\sum_{r,s=1}^{K}F_j(r,s)}{1 + \sum_{r,s=1}^{K}E_j(r,s)}. \tag{55}$$

To proceed further, we need the limiting behaviour of $F_j(r,s), E_j(r,s)$. From Lemma B.6, we have the results given below.

$$\begin{cases} \max\limits_{1 \le j \le n}|F_j(r,r) - b_j^{(r)}m_{rn}| \xrightarrow{a.s.} 0; & \max\limits_{1 \le j \le n}|F_j(r,s)| \xrightarrow{a.s.} 0, r \ne s; \\ \max\limits_{1 \le j \le n}|E_j(r,r) - c_n b_j^{(r)}h_{rn}| \xrightarrow{a.s.} 0; & \max\limits_{1 \le j \le n}|E_j(r,s)| \xrightarrow{a.s.} 0, r \ne s. \end{cases}$$

Using them in (55), we get

$$\left|Term_2 - \frac{1}{p}\sum_{j=1}^{n}\sum_{r=1}^{K}\frac{b_j^{(r)}m_{rn}}{1 + c_n\sum_{s=1}^{K}b_j^{(s)}h_{sn}}\right| \xrightarrow{a.s.} 0. \tag{56}$$

Now note that

$$\frac{1}{p}\sum_{j=1}^{n}\sum_{r=1}^{K}\frac{b_j^{(r)}m_{rn}}{1 + c_n\sum_{s=1}^{K}b_j^{(s)}h_{sn}}$$

$$= \frac{1}{p}\sum_{j=1}^{n}\frac{\sum_{r=1}^{K}\frac{1}{n}\text{trace}\{b_j^{(r)}A_n^{(r)}\bar{Q}M_n Q\}}{1 + c_n\sum_{s=1}^{K}b_j^{(s)}h_{sn}}$$

$$= \frac{1}{p}\text{trace}\left\{\sum_{r=1}^{K}A_n^{(r)}\left(\sum_{j=1}^{n}\frac{1}{n}\frac{b_j^{(r)}}{1 + c_n\sum_{s=1}^{K}b_j^{(s)}h_{sn}}\right)\bar{Q}M_n Q\right\}$$

$$= \frac{1}{p}\text{trace}\left(\sum_{r=1}^{K}A_n^{(r)} \times \frac{1}{n}\text{trace}\{B_n^{(r)}[I_n + c_n\sum_{s=1}^{K}h_{sn}B_n^{(s)}]^{-1}\}\right)\bar{Q}M_n Q.$$

Finally using Lemma B.2, we get

$$\left|Term_2 - \frac{1}{p}\text{trace}\left(\sum_{r=1}^{K}A_n^{(r)} \times \frac{1}{n}\text{trace}\{B_n^{(r)}[I_n + c_n\sum_{s=1}^{K}h_{sn}B_n^{(s)}]^{-1}\}\right)\bar{Q}M_n Q\right| \xrightarrow{a.s.} 0$$

$$\implies \left|Term_2 - \left(zI_p - zI_p + \frac{1}{p}\text{trace}\left(\sum_{r=1}^{K}A_n^{(r)} \times \frac{1}{n}\text{trace}\{B_n^{(r)}[I_n + c_n\sum_{s=1}^{K}\mathbb{E}h_{sn}B_n^{(s)}]^{-1}\}\right)\bar{Q}M_n Q\right)\right| \xrightarrow{a.s.} 0$$

$$\implies \left|Term_2 - \frac{1}{p}\text{trace}\{(zI_p + F(z))\bar{Q}M_n Q\}\right| \xrightarrow{a.s.} 0$$

$$\implies |Term_2 - Term_1| \xrightarrow{a.s.} 0.$$

This concludes the proof. □

**Lemma B.7.** *Let $\boldsymbol{x}(\cdot) = \{x_r(\cdot)\}_{r=1}^{K}$ and $\boldsymbol{y}(\cdot) = \{y_r(\cdot)\}_{r=1}^{K}$ be Stieltjes Transforms of some*

*measures on* $\mathbb{R}$. *Using definition (7) and* $c, H, G$ *as defined in Theorem 4.1, the below functions are holomorphic for any* $r \in [K]$:

$$\eta_r : \mathbb{C}_+ \to \mathbb{C}_+; \quad \eta_r(z) = O_r(z, c\boldsymbol{x}(z), G),$$

$$\nu_r : \mathbb{C}_+ \to \mathbb{C}_+; \quad \nu_r(z) = O_r(z, \boldsymbol{y}(z), H).$$

*Proof.* Fix $r \in [K]$. Let $\gamma$ be an arbitrary closed curve in $\mathbb{C}_+$. There exists a compact set $M \subset \mathbb{C}_+$ that contains $\gamma$. Since $x_r(\cdot)$ is holomorphic, $x_r(M) \subset \mathbb{C}_+$ is compact. Define $B_M := \inf_{z \in M} |z| \Im(x_r(z))$. Clearly $B_M > 0$ and this allows us to establish the following bound:

$$\left| \frac{\theta_r}{-z(1 + c\boldsymbol{\theta}^T \mathbf{x}(z))} \right| \le \frac{1}{c|z|\Im(x_r(z))} \le \frac{1}{cB_M} < \infty.$$

Now applying Fubini, Cauchy and Morera, we have the following result:

$$\int_\gamma O_r(z, c\mathbf{x}(z), G) dz = \int_\gamma \int \frac{\theta_r dG(\boldsymbol{\theta})}{-z(1 + c\boldsymbol{\theta}^T \mathbf{x}(z))} dz$$

$$= \int \left( \int_\gamma \frac{\theta_r}{-z(1 + c\boldsymbol{\theta}^T \mathbf{x}(z))} dz \right) dG(\boldsymbol{\theta}) = 0.$$

The proof for the other part is similar. $\qquad \square$

## B.2 Proof of Theorem Existence under A1-A2

*Proof.* By Theorem 4.3, we can choose a common subsequence $\{n_k\}_{k=1}^\infty$ such that $\{\mathbf{h}_{n_k}, \mathbf{g}_{n_k}\}_{k=1}^\infty$ converge uniformly within every compact subset of $\mathbb{C}_+$. For convenience, we denote these subsequences as $\{\mathbf{x}_k, \mathbf{y}_k\}_{k=1}^\infty$ and the corresponding limits as $\mathbf{x}^\infty, \mathbf{y}^\infty$ respectively. For convenience, we introduce the following notations along the subsequence $\{n_k\}_{k=1}^\infty$:

1. $\tilde{\mathbf{x}}_k = \tilde{\mathbf{h}}_{n_k}; \quad \tilde{\tilde{\mathbf{x}}}_k = \tilde{\tilde{\mathbf{h}}}_{n_k}$,

2. $\tilde{\mathbf{y}}_k = \tilde{\mathbf{g}}_{n_k}; \quad \tilde{\tilde{\mathbf{y}}}_k = \tilde{\tilde{\mathbf{g}}}_{n_k}$,

3. $d_k = c_{n_k}$, and

4. $P_k = H_{n_k}; \quad Q_k = G_{n_k}$.

Fix $z = u + \mathrm{i}v \in \mathbb{C}_+$ and let $r \in [K]$. From Lemma B.5, we have

$$\tilde{x}_{rk}(z) - \tilde{\tilde{x}}_{rk}(z) \to 0$$

$$\Longrightarrow \tilde{x}_{rk}(z) - \int \frac{\lambda_r dP_k(\boldsymbol{\lambda})}{-z(1 + \boldsymbol{\lambda}^T \mathbf{O}(z, d_k \tilde{\mathbf{x}}_k(z), Q_k))} \longrightarrow 0$$

$$\Longrightarrow \tilde{x}_{rk}(z) - \int \frac{\lambda_r d\{P_k(\boldsymbol{\lambda}) - H(\boldsymbol{\lambda})\}}{-z(1 + \boldsymbol{\lambda}^T \mathbf{O}(z, d_k \tilde{\mathbf{x}}_k(z), Q_k))} - \int \frac{\lambda_r dH(\boldsymbol{\lambda})}{-z(1 + \boldsymbol{\lambda}^T \mathbf{O}(z, d_k \tilde{\mathbf{x}}_k(z), Q_k))} \longrightarrow 0.$$

Note that by **A1** and Lemma 4.1, the common integrand in the second and third terms are

bounded.

$$\left| \frac{\lambda_r}{-z(1 + \boldsymbol{\lambda}^T \mathbf{O}(z, d_k \tilde{\mathbf{x}}_k(z), Q_k))} \right| \le \frac{\tau}{v}. \tag{57}$$

Since $P_k \xrightarrow{d} H$ by **T4**, the second term vanishes in the limit. Moreover $Q_k \xrightarrow{d} G$ and $\tilde{\mathbf{x}}_k \to \mathbf{x}^\infty$. Therefore, by Lemma B.4, we have $\mathbf{O}(z, d_k \tilde{\mathbf{x}}_k(z), Q_k) \to \mathbf{O}(z, d_k \mathbf{x}^\infty(z), G)$. Finally, applying DCT, we get

$$x_r^\infty(z) = \int \frac{\lambda_r dH(\boldsymbol{\lambda})}{-z(1 + \boldsymbol{\lambda}^T \mathbf{O}(z, c\mathbf{x}^\infty(z), G))}. \tag{58}$$

This implies that any subsequential limit of $\mathbf{h}_n(z)$ satisfies equation (9). Similarly, starting from the second result of Lemma B.5, we can establish that any subsequential limit of $\mathbf{g}_n(z)$ satisfies equation (11).

$$y_r^\infty(z) = \int \frac{\theta_r dG(\boldsymbol{\lambda})}{-z(1 + c\boldsymbol{\theta}^T \mathbf{O}(z, \mathbf{y}^\infty(z), H))}. \tag{59}$$

Now we will show that $\mathbf{x}^\infty$ and $\mathbf{y}^\infty$ satisfy (12). Using Lemma B.4, we observe that

$$
\begin{aligned}
&\|\mathbf{x}^\infty(z) - \mathbf{O}(z, \mathbf{y}^\infty(z), H)\|_1 \\
&= \lim_{k\to\infty} \|\mathbf{x}_k(z) - \mathbf{O}(z, \mathbf{y}_k(z), P_k)\|_1 \\
&= \lim_{k\to\infty} \|\tilde{\mathbf{x}}_k(z) - \mathbf{O}(z, \mathbf{y}_k(z), P_k)\|_1 \\
&= \sum_{r=1}^{K} \lim_{k\to\infty} |\tilde{x}_{rk}(z) - O_r(z, \mathbf{y}_k(z), P_k)|.
\end{aligned} \tag{60}
$$

Making use of Lemma B.2 and Lemma B.4, we expand the $r^{th}$ term in the above expansion as follows.

$$
\begin{aligned}
&|\tilde{x}_{rk}(z) - O_r(z, \mathbf{y}_k(z), P_k)| \tag{61}\\
&= \left| \int \frac{\lambda_r dP_k(\boldsymbol{\lambda})}{-z(1 + \boldsymbol{\lambda}^T \mathbf{O}(z, d_k \mathbb{E}\mathbf{x}_k(z), Q_k))} - \int \frac{\lambda_r dP_k(\boldsymbol{\lambda})}{-z(1 + d_k \boldsymbol{\lambda}^T \mathbf{y}_k(z))} \right| \\
&= |z| \left| \int \frac{\boldsymbol{\lambda}^T(\mathbf{y}_k(z) - \mathbf{O}(z, d_k \mathbb{E}\mathbf{x}_k(z), Q_k)) \lambda_r dP_k(\boldsymbol{\lambda})}{(-z - d_k z \boldsymbol{\lambda}^T \mathbf{O}(z, \mathbb{E}\mathbf{x}_k(z), Q_k))(-z - z\boldsymbol{\lambda}^T \mathbf{y}_k(z))} \right| \\
&\le |z| \sum_{s=1}^{K} |y_{sk}(z) - O_s(z, d_k \mathbb{E}\mathbf{x}_k(z), Q_k)| \int \left| \frac{\lambda_r \lambda_s}{(z + z\boldsymbol{\lambda}^T \mathbf{O}(z, d_k \mathbb{E}\mathbf{x}_k(z), Q_k))(z + z\boldsymbol{\lambda}^T \mathbf{y}_k(z))} \right| dP_k(\boldsymbol{\lambda}) \\
&\le \frac{|z|\tau^2}{v^2} \sum_{s=1}^{K} |y_{sk}(z) - O_s(z, d_k \mathbb{E}\mathbf{x}_k(z), Q_k)| \\
&\le \frac{|z|\tau^2}{v^2} \sum_{s=1}^{K} |y_{sk}(z) - O_s(z, d_k \mathbf{x}_k(z), Q_k)| + \epsilon, \text{ for sufficiently large } k \\
&= \frac{|z|\tau^2}{v^2} \|\mathbf{y}_k - \mathbf{O}(z, d_k \mathbf{x}_k(z), Q_k)\|_1 + \epsilon.
\end{aligned}
$$

Similarly, we expand the $s^{th}$ term in the above expansion and observe that

$$
|y_{sk}(z) - O_s(z, d_k \mathbf{x}_k(z), Q_k)| \tag{62}
$$

$$
= \left| \int \frac{\theta_s dQ_k(\boldsymbol{\theta})}{-z(1 + d_k \boldsymbol{\theta}^T \mathbf{O}(z, d_k \mathbb{E} \mathbf{y}_k(z), P_k))} - \int \frac{\theta_s dQ_k(\boldsymbol{\theta})}{-z(1 + d_k \boldsymbol{\theta}^T \mathbf{x}_k(z))} \right|
$$

$$
= |z| \left| \int \frac{\boldsymbol{\theta}^T (\mathbf{x}_k(z) - \mathbf{O}(z, \mathbb{E} \mathbf{y}_k(z), P_k)) \theta_s dQ_k(\boldsymbol{\theta})}{(-z - d_k z \boldsymbol{\theta}^T \mathbf{O}(z, \mathbb{E} \mathbf{y}_k(z), P_k))(-z - d_k z \boldsymbol{\theta}^T \mathbf{x}_k(z))} \right|
$$

$$
\leq |z| \sum_{t=1}^{K} |x_{tk}(z) - O_t(z, \mathbb{E} \mathbf{y}_k(z), P_k)| \int \left| \frac{\theta_s \theta_t}{(z + d_k z \boldsymbol{\theta}^T \mathbf{O}(z, \mathbb{E} \mathbf{y}_k(z), P_k))(z + d_k z \boldsymbol{\theta}^T \mathbf{x}_k(z))} \right| dQ_k(\boldsymbol{\theta})
$$

$$
\leq \frac{|z| \tau^2}{v^2} \sum_{t=1}^{K} \left( |x_{tk}(z) - O_t(z, \mathbf{y}_k(z), P_k)| + \epsilon \right), \text{ for large } k
$$

$$
= \frac{|z| \tau^2}{v^2} ||\mathbf{x}_k - \mathbf{O}(z, \mathbf{y}_k(z), P_k)||_1 + \frac{K \epsilon |z| \tau^2}{v^2}.
$$

Therefore, combining (61) and (62), we have

$$
||\mathbf{y}_k(z) - \mathbf{O}(z, d_k \mathbf{x}_k(z), Q_k)||_1 \leq \frac{K|z| \tau^2}{v^2} ||\mathbf{x}_k - \mathbf{O}(z, \mathbf{y}_k(z), P_k)||_1 + \frac{K^2 \epsilon |z| \tau^2}{v^2}
$$

$$
\implies |\tilde{x}_{rk}(z) - O_r(z, \mathbf{y}_k(z), P_k)| \leq \frac{K|z|^2 \tau^4}{v^4} ||\mathbf{x}_k - \mathbf{O}(z, \mathbf{y}_k(z), P_k)||_1 + \frac{K^2 \epsilon |z|^2 \tau^4}{v^4} + \epsilon.
$$

Choose $z \in \mathbb{C}_+$ such that $|u| \leq v$ which in particular implies that $|z|^2 \leq 2v^2$. For sufficiently large $k$, Theorem 4.4 implies that

$$
|x_{rk}(z) - O_r(z, \mathbf{y}_k(z), P_k)| \leq |\tilde{x}_{rk}(z) - O_r(z, \mathbf{y}_k(z), P_k)| + \epsilon \tag{63}
$$

$$
\implies ||\mathbf{x}_k(z) - \mathbf{O}(z, \mathbf{y}_k(z), P_k)||_1 \leq \frac{2K^2 \tau^4}{v^2} ||\mathbf{x}_k(z) - \mathbf{O}(z, \mathbf{y}_k(z), P_k)||_1 + \frac{2\tau^4 K^3}{v^2} \epsilon + K\epsilon.
$$

Using Lemma B.4, we take limits on both sides to get the following.

$$
||\mathbf{x}^\infty(z) - \mathbf{O}(z, \mathbf{y}^\infty(z), H)||_1 \leq \frac{2K^2 \tau^4}{v^2} ||\mathbf{x}^\infty(z) - \mathbf{O}(z, \mathbf{y}^\infty(z), H)||_1 + \frac{2\tau^4 K^3}{v^2} \epsilon + K\epsilon. \tag{64}
$$

Choose $\epsilon > 0$ arbitrarily small and then for large enough $v > 0$, we will end up with a contradiction unless $||\mathbf{x}^\infty(z) - \mathbf{O}(z, \mathbf{y}^\infty(z), H)||_1 = 0$. Similarly, we can derive that $||\mathbf{y}^\infty(z) - O(z, c\mathbf{x}^\infty(z), G)||_1 = 0$. For any $r \in [K]$, $x_r^\infty(\cdot)$ is holomorphic as it is a Stieltjes Transform and $O_r(\cdot)$ as a function of $z \in \mathbb{C}_+$ is holomorphic by Lemma B.7. Since these two agree on an open subset of $\mathbb{C}_+$, they must be equal on the whole of $\mathbb{C}_+$.

So $\mathbf{x}^\infty$ and $\mathbf{y}^\infty$ satisfy (9), (11) and (12). By Theorem 4.2, all these subsequential limits must coincide which we will refer to as $\mathbf{h}^\infty$ and $\mathbf{g}^\infty$ going forward. Moreover, $h_r^\infty$ and $g_r^\infty$ themselves are Stieltjes Transforms of real measures due to Theorem 4.3. This establishes (1) and (2) of Theorem 4.5.

We now show that $s_n(z) \xrightarrow{a.s.} s_F(z)$ where $s_F(z)$ is defined in (8). From Theorem 4.4, we have

$$
|s_n(z) - \frac{1}{p} \text{trace}(\bar{Q}(z))| \xrightarrow{a.s.} 0.
$$

Therefore, all that remains is to show that

$$\left| \frac{1}{p} \text{trace}(\bar{Q}(z)) - \int \frac{dH(\boldsymbol{\lambda})}{-z(1 + \boldsymbol{\lambda}^T \mathbf{O}(z, c\mathbf{h}^\infty(z), G))} \right| \to 0.$$

Due to **T3** of Theorem 4.1, we have

$$\frac{1}{p} \text{trace}(\bar{Q}(z)) = \int \frac{dH_n(\boldsymbol{\lambda})}{-z(1 + \boldsymbol{\lambda}^T \mathbf{O}(z, c_n \mathbb{E}\mathbf{h}_n(z), G_n))} \tag{65}$$

$$= \int \frac{d\{H_n(\boldsymbol{\lambda}) - H(\boldsymbol{\lambda})\}}{-z(1 + \boldsymbol{\lambda}^T \mathbf{O}(z, c_n \mathbb{E}\mathbf{h}_n(z), G_n))} + \int \frac{dH(\boldsymbol{\lambda})}{-z(1 + \boldsymbol{\lambda}^T \mathbf{O}(z, c_n \mathbb{E}\mathbf{h}_n(z), G_n))}.$$

By Lemma 4.1, the common integrand in both the terms is bounded by $1/v$. The second term goes to 0, since $H_n \xrightarrow{d} H$. By Lemma B.2, we now have $||\mathbb{E}\mathbf{h}_n(z) - \mathbf{h}^\infty(z)||_1 \xrightarrow{a.s.} 0$. Therefore, by Lemma B.4, we conclude $||\mathbf{O}(z, c_n \mathbb{E}\mathbf{h}_n(z), G_n)) - \mathbf{O}(z, c_n \mathbf{h}_n^\infty(z), G))||_1 \to 0$. Finally applying DCT in the second term of (65) gives us

$$\lim_{n \to \infty} \int \frac{dH(\boldsymbol{\lambda})}{-z(1 + \boldsymbol{\lambda}^T \mathbf{O}(z, c_n \mathbb{E}\mathbf{h}_n(z), G))} = \int \frac{dH(\boldsymbol{\lambda})}{-z(1 + \boldsymbol{\lambda}^T \mathbf{O}(z, c\mathbf{h}^\infty(z), G))} = s_F(z). \tag{66}$$

Therefore, $s_n(z) \xrightarrow{a.s.} s_F(z)$. This establishes the equivalence between (8) and (13).

Now, we will show that $s_F(\cdot)$ satisfies the conditions of Theorem 2.1. From Lemma 4.2, we have

$$|\mathrm{i}y(\mathbf{h}^\infty(\mathrm{i}y))^T \mathbf{g}^\infty(\mathrm{i}y)| \leq \sum_{r=1}^{K} y|h_r^\infty(\mathrm{i}y)| \, |g_r^\infty(\mathrm{i}y)| \leq Ky \frac{C_0}{y} \frac{C_0}{y} \to 0 \text{ as } y \to \infty. \tag{67}$$

Thus using (13), we have

$$z s_F(z) = -z(z^{-1} + \mathbf{h}^T(z)\mathbf{g}(z)) \implies \lim_{y \to \infty} \mathrm{i}y s_F(\mathrm{i}y) = -1. \tag{68}$$

Since $s_F(.)$ satisfies the necessary and sufficient condition from Proposition 2.1, it is the Stieltjes transform of some probability distribution. By Proposition 2.1, this underlying measure $F$ is the L.S.D. of $F^{S_n}$. This completes the proof of Theorem 4.1 under Assumptions 4.1.1. $\square$

# C  Results related to implementation of Lindeberg's Principle

**Theorem C.1.** *Let $c \in (0, \infty)$ and $\lambda_+ = (1 + \sqrt{c})^2$. Suppose that the entries of the matrix $\boldsymbol{X}_n = (x_{jkn}, j \leq p, k \leq n)$ are independent (not necessarily identically distributed) and satisfy*

*1 $\mathbb{E}x_{jkn} = 0$,*

*2 $|x_{jkn}| \leq n^b; b \in (0, \frac{1}{2})$,*

*3 $\sup_{j,k,n} |\mathbb{E}|x_{jkn}|^2 - 1| \longrightarrow 0$, and*

*4 $\mathbb{E}|x_{jkn}|^l \leq d(\delta_n \sqrt{n})^{l-2}$ for all $l \geq 2$ for some $d > 0$,*

*where $\delta_n \to 0$ and $b > 0$. Let $\boldsymbol{S}_n = \frac{1}{n}\boldsymbol{X}_n\boldsymbol{X}_n^*$. Then for any $\epsilon > 0$,*

$$\mathbb{P}(\lambda_{max}(\boldsymbol{S}_n) > \lambda_+ + \epsilon, \ i.o.) = 0.$$

*Proof.* Fix arbitrary $\epsilon > 0$. We have $\lambda_{max}(\mathbf{S}_n) \leq \operatorname{tr}(\mathbf{S}_n)$. In particular, for $t = \lambda_+ + \epsilon$ and any $k \in \mathbb{N}$, we have

$$\mathbb{P}(\lambda_{max}(\mathbf{S}_n) > t) \leq \mathbb{P}(\operatorname{tr}(\mathbf{S}_n^k) > t^k) \leq \frac{\mathbb{E}\operatorname{tr}(\mathbf{S}_n^k)}{t^k}. \tag{69}$$

Expanding the tracial term above, we get

$$\operatorname{tr}(S_n^k) = \frac{1}{n^k}\sum_{i_1,\ldots,i_k}\sum_{j_1,\ldots,j_k} x_{i_1 j_1}\, \overline{x_{i_2 j_1}}\, x_{i_2 j_2}\, \overline{x_{i_3 j_2}}\cdots x_{i_k j_k}\, \overline{x_{i_1 j_k}}. \tag{70}$$

Here, $i_l \in [p]$ and $j_l \in [n]$ for each $l \in [k]$. Note that if $\mathbf{i}$ and $\mathbf{j}$ are such that $x_{i_l,j_l}$ or $\overline{x_{i_l,j_l}}$ appears only once for some $l \in [k]$, then that term does not contribute anything to $\mathbb{E}(S_n^k)$. So the contributing terms in the RHS of (70) are those whose unique (upto conjugacy, i.e. $x_{ij}$ and $\overline{x}_{ij}$ will be counted as the same element) components appear at least twice.

Let $s$ be the number of unique terms for a particular combination of $\mathbf{i}$ and $\mathbf{j}$ indices. Note that $1 \leq s \leq k+1$ where $s = 1$ denotes the case where we have a single node and $s = k+1$ denotes where the bi-partite graph becomes a tree without any cycles. For all other values of $s$, the graph has cycles.

Let the number of unique row indices be $u$ and the number of unique column indices be $v$. There are $O((cn)^u)$ ways to choose the unique row indices and $n^v$ ways to choose the column indices. By **a basic graph theoretic result**, (i.e., a connected graph $(V, E)$ satisfies $|V| \leq |E| + 1$) we have $u + v \leq s + 1$. So, the total number of $\mathbf{i,j}$ tuples that involve $s$ distinct $x_{ij}$ elements are given by $c^u n^{u+v} \leq c^k n^{s+1}$. Let $m_l$ be the multiplicity of $x_{i_l,j_l}$. Then, $\sum_{l=1}^s m_l = 2k$. By Condition 4, we get the following bound on the contribution of individual terms of (70)

$$\left|\mathbb{E}\prod_{l=1}^s x_{i_l,j_l}^{m_l}\right| \leq d^s(n^b)^{2(k-s)}.$$

Without loss of generality, we can assume $c \leq 1$. This is because we only need an almost sure upper bound for $||\frac{1}{\sqrt{n}}X||_{op}$. If $c > 1$, we note that $||\frac{1}{\sqrt{n}}X||_{op} = \sqrt{\frac{p}{n}}||\frac{1}{\sqrt{p}}X^*||_{op}$. Thus, we need to multiply the bound by $2\sqrt{c}$ (the factor of 2 is for good measure) to get a valid upper bound.

For $s = k+1$, a bound for overall contribution is given by

$$T_{n+1} := \frac{1}{n^k}c^k n^{k+2} d^{k+1} n^{2b(k-k-1)} = d(dc)^k n^{2(1-b)} \leq d^{k+1} n^{2(1-b)}.$$

Similarly, for $s = k$, a bound for overall contribution is given by

$$T_n := \frac{1}{n^k}c^k n^{k+1} d^{k+1} n^{2b(k-k)} = d(dc)^k n^0 \leq d^{k+1} n.$$

Since $2(1-b) > 1$, we have $T_{n+1} >> T_n$. The other contributions are also negligible.

Therefore, for large enough $n$, we have, $\mathbb{E}\operatorname{tr}(S_n^k) = O(d^{k+1}n^{2(1-b)})$. Plugging this into (69), we get

$$\mathbb{P}(\lambda_{max}(S_n) > t) \leq \frac{d^{k+1}n^{2(1-b)}}{t^k} = d(d/t)^k n^{2(1-b)}.$$

We will choose $k$ (depending on $n$) suitably to impose a polynomial decay of fixed order $N \geq 2$ on the tail probability of $\lambda_{max}$. Note that we can choose $d$ to be $1 + \epsilon/2$ because of Condition 3. Since $\lambda_+ = (1 + \sqrt{c})^2 > 1$, we always have $t = \lambda_+ + \epsilon > 1 + \epsilon/2 = d$. In particular, $\log_n(d/t) < 0$. For this note that

$$\frac{d^k n^{2(1-b)}}{t^k} \leq \frac{1}{n^N}$$

$$\Longleftrightarrow k\log_n(d/t) + 2(1-b) \leq -N$$

$$\Longleftrightarrow k \geq -\frac{N + 2(1-b)}{\log_n(d/t)}.$$

**Remark C.1.** *Note that Condition 4 is implied by Conditions 2 and 3. Moreover, Condition 3 is implied if we assume a uniform $2 + \eta_0$ moment bound on the innovations.*

$\square$

**Theorem C.2.** *Suppose $\boldsymbol{Z}$ and $\boldsymbol{W}$ are random vectors in $\mathbb{R}^N$ with $\boldsymbol{W}$ having independent components. For $1 \leq i \leq N$, let*

1. *$A_i := \mathbb{E}|\mathbb{E}(z_i|z_1, \ldots, z_{i-1}) - \mathbb{E}w_i|; \quad B_i := \mathbb{E}|\mathbb{E}(z_i^2|z_1, \ldots, z_{i-1}) - \mathbb{E}w_i^2|,$*

2. *$\Phi_i := [z_1, \ldots, z_i, w_{i+1}, \ldots, w_N]; \quad \Phi_i^0 := [z_1, \ldots, z_{i-1}, 0, w_{i+1}, \ldots, w_N],$*

3. *$\Psi_i^\lambda := \lambda\Phi_i^0 + (1-\lambda)\Phi_i, \lambda \in [0, 1].$*

*Let $M_3$ be a bound on $\max_i \mathbb{E}|X_i|^3 + \mathbb{E}|Y_i|^3$. Suppose $f : \mathbb{R}^N \to \mathbb{R}$ is a thrice continuously differentiable function. For $r = 1, 2$, let $L_r(f)$ be a constant such that $|\partial_i^r f(\boldsymbol{x})| \leq L_r(f)$ for each $i$ and $\boldsymbol{x}$, where $\partial_i^r$ denotes the $r$-fold derivative in the $i^{th}$ coordinate. Then*

$$|\mathbb{E}f(\boldsymbol{Z}) - \mathbb{E}f(\boldsymbol{W})| \leq \sum_{i=1}^N (A_i L_1(f) + B_i L_2(f)) + \frac{M_3}{6}\mathbb{E}\left(\sup_{\lambda \in [0,1]} |\partial_i^3 f(\Psi_i^\lambda)| + \sup_{\lambda \in [0,1]} |\partial_i^3 f(\Psi_{i-1}^\lambda)|\right). \tag{71}$$

*Proof.* The proof is very similar to that of Theorem 1.1 of Chatterjee (2006). $\square$

## C.1 Proof of Theorem 4.6

*Proof.* Separating the real and imaginary parts of $Z_n^{(r)}, W_n^{(r)}$, we reshape them into vectors as follows:

1. $\mathbf{W} = \operatorname{vec}(\Re(W_n^{(1)}), \Im(W_n^{(1)}), \ldots, \Re(W_n^{(K)}), \Im(W_n^{(K)})) \in \mathbb{R}^N,$

2. $\mathbf{Z} = \operatorname{vec}(\Re(Z_n^{(1)}), \Im(Z_n^{(1)}), \ldots, \Re(Z_n^{(K)}), \Im(Z_n^{(K)})) \in \mathbb{R}^N.$

For a fixed $z = u + \mathring{\imath}v \in \mathbb{C}_+$, let $\mathcal{Q}(\mathbf{m}) = (\mathcal{S}(\mathbf{m}) - zI_p)^{-1}$. Also, let

$$f : \mathbb{R}^N \to \mathbb{C}; \quad f(\mathbf{m}) := \frac{1}{p}\operatorname{trace}(\mathcal{Q}(\mathbf{m})). \tag{72}$$

Clearly, $f(\mathbf{Z}) = s_n(z)$ and $f(\mathbf{W}) = t_n(z)$. We will use a modified version of Theorem 1.1 from Chatterjee (2006) to achieve this. As per the notations of Theorem C.2, we have $A_i = 0, B_i = 0$ for $i \in [N]$ since the entries of $\mathbf{Z}$ are independent and the first two moments of $\mathbf{Z}$ and $\mathbf{W}$ match. Clearly $M_3 < \infty$ in this case. So all that remains is to show is that the last term in (71) can be made arbitrarily small for large $n$.

For arbitrary $l \in [N]$ and $\lambda \in [0,1]$, we evaluate $\sup_{\lambda \in [0,1]} |\partial_l^3 f(\Phi_l^\lambda)|$. Denote $\mathbf{m} = \Psi_l^\lambda$. Using (36), we have

$$p|\partial_l^3 f(\mathbf{m})| \le 6||\mathcal{Q}(\mathbf{m})||_{op}^4||\partial_l \mathcal{S}(\mathbf{m})||_F^3 + 6||\mathcal{Q}(\mathbf{m})||_{op}^3||\partial_l^2 \mathcal{S}(\mathbf{m})||_F||\partial_l \mathcal{S}(\mathbf{m})||_F \tag{73}$$
$$\implies |\partial_l^3 f(\mathbf{m})| \le \frac{6}{pv^4}||\partial_l \mathcal{S}(\mathbf{m})||_F^3 + \frac{6}{pv^3}||\partial_l^2 \mathcal{S}(\mathbf{m})||_F||\partial_l \mathcal{S}(\mathbf{m})||_F.$$

Here we used the fact that $||\mathcal{Q}(\mathbf{m})||_{op} \le 1/v$ holds for any $\mathbf{m}$. Since $l \in [N] = [2Kpn]$, there exists $k \in [K]$ such that $np(2k-2)+1 \le l \le 2npk$.

**Case1:** $np(2k-2)+1 \le l \le np(2k-1)$ for some $k \in [K]$. Then, $m_l$ is an entry in the real part of $\mathcal{X}_k(\mathbf{m})$, say the $(i,j)^{th}$ entry. Using Lemma (35), we derive an expression for $\partial_l S(\mathbf{m})$.

$$
\begin{aligned}
n\partial_l \mathcal{S}(\mathbf{m}) &= (\partial_l \mathcal{X}(\mathbf{m}))\mathcal{X}(\mathbf{m})^* + \mathcal{X}(\mathbf{m})(\partial_l \mathcal{X}(\mathbf{m})^*) \\
&= \sum_{r=1}^K (\partial_l \mathcal{X}_r(\mathbf{m}))\mathcal{X}(\mathbf{m})^* + \mathcal{X}(\mathbf{m})\sum_{r=1}^K(\partial_l \mathcal{X}_r(\mathbf{m})^*) \\
&= (\partial_l \mathcal{X}_k(\mathbf{m}))\mathcal{X}(\mathbf{m})^* + \mathcal{X}(\mathbf{m})(\partial_l \mathcal{X}_k(\mathbf{m})^*) \\
&= U_{\cdot i}^{(k)}V_{j\cdot}^{(k)}\mathcal{X}(\mathbf{m})^* + \mathcal{X}(\mathbf{m})V_{\cdot j}^{(k)}U_{i\cdot}^{(k)}.
\end{aligned}
$$

**Case2:** Suppose $np(2k-1)+1 \le l \le 2npk$. Let $l$ be the $(i,j)^{th}$ entry as before. In this case, we have

$$n\partial_l \mathcal{S}(\mathbf{m}) = \mathring{\imath}\left(U_{\cdot i}^{(k)}V_{j\cdot}^{(k)}\mathcal{X}(\mathbf{m})^* - \mathcal{X}(\mathbf{m})V_{\cdot j}^{(k)}U_{i\cdot}^{(k)}\right).$$

In either case, using basic inequalities involving operator and Frobenius norms, we observe

$$
\begin{aligned}
||\partial_l \mathcal{S}(\mathbf{m})||_F &\le (||U_{\cdot i}^{(k)}V_{j\cdot}^{(k)}\mathcal{X}(\mathbf{m})^*||_F + ||\mathcal{X}(\mathbf{m})V_{\cdot j}^{k}U_{i\cdot}^{(k)}||_F)/n \\
&\le (||U_{\cdot i}^{(k)}||_2\,||V_{j\cdot}^{(k)}\mathcal{X}(\mathbf{m})^*||_2 + ||\mathcal{X}(\mathbf{m})V_{\cdot j}^{(j)}||_2\,||U_{i\cdot}^{(k)}||_2)/n \\
&\le \sum_{r=1}^K (||U_{\cdot i}^{(k)}||_2\,||V_{j\cdot}^{(k)}\mathcal{X}_r(\mathbf{m})^*||_2 + ||\mathcal{X}_r(\mathbf{m})V_{\cdot j}^{(k)}||_2\,||U_{i\cdot}^{(k)}||_2)/n \\
&\le \sum_{r=1}^K ||U_{\cdot i}^{(k)}||_2\,||V_{j\cdot}^{(k)}||_2||\mathcal{X}_r(\mathbf{m})||_{op}/n + ||V_{\cdot j}^{(k)}||_2\,||U_{i\cdot}^{(k)}||_2||\mathcal{X}_r(\mathbf{m})||_{op}/n \\
&\le 2K\tau n^{-\frac{1}{2}}||\mathcal{X}_r(\mathbf{m})/\sqrt{n}||_{op}.
\end{aligned}
$$

By Theorem C.1, for sufficiently large $n$ we have the following bound for $||\partial_l \mathcal{S}(\mathbf{m})||_F$.

$$||\mathcal{X}_r(\mathbf{m})/\sqrt{n}||_{op} = ||\lambda \mathcal{X}_r(\Phi_l^0)/\sqrt{n} + (1-\lambda)\mathcal{X}_r(\Phi_l)/\sqrt{n}||_{op} \leq 4C\tau(1+\sqrt{c}) \text{ a.s.}$$
$$\implies ||\partial_l \mathcal{S}(\mathbf{m})||_F \leq 8KC\tau^2 n^{-0.5}(1+\sqrt{c}) := C_1 n^{-0.5}.$$

Simplifying $\partial_l^2 \mathcal{S}(\mathbf{m})$ using (35) when $l$ corresponds to the $(i,j)^{th}$ entry in the real part of $\mathcal{X}_k(\mathbf{m})$, we have

$$n\partial_l^2 \mathcal{S}(\mathbf{m}) = \partial_l(U_{\cdot i}^{(k)} V_{j\cdot}^{(k)} \mathcal{X}(\mathbf{m})^* + \mathcal{X}(\mathbf{m}) V_{\cdot j}^{(k)} U_{i\cdot}^{(k)})$$
$$= \sum_{r=1}^K \partial_l(U_{\cdot i}^{(k)} V_{j\cdot}^{(k)} \mathcal{X}_r(\mathbf{m})^* + \mathcal{X}_r(\mathbf{m}) V_{\cdot j}^{(k)} U_{\cdot i}^{(k)})$$
$$= \partial_l(U_{\cdot i}^{(k)} V_{j\cdot}^{(k)} \mathcal{X}_k(\mathbf{m})^* + \mathcal{X}_k(\mathbf{m}) V_{\cdot j}^{(k)} U_{\cdot i}^{(k)}) = 2U_{\cdot i}^{(k)} V_{j\cdot}^{(k)} V_{\cdot j}^{(k)} U_{\cdot i}^{(k)}$$
$$\implies ||\partial_l^2 S(\mathbf{m})||_F \leq 2\tau^2/n.$$

The same bound works even when $l$ corresponds to the $(i,j)^{th}$ entry of the imaginary part of $\mathcal{X}_k(\mathbf{m})$. Therefore, an upper bound for the third partial derivative of $f$ is given by

$$|\partial_l^3 f(\mathbf{m})| \leq \frac{6}{pv^4}(C_1 n^{-0.5})^3 + \frac{6}{pv^3}\frac{2\tau^2}{n} C_1 n^{-0.5} = C(c, K, v, \tau)n^{-2.5}.$$

Since $l \in [N]$ and $\lambda \in [0,1]$ were arbitrary, we use Theorem C.2 to conclude that

$$|\mathbb{E}f(\mathbf{Z}) - \mathbb{E}f(\mathbf{W})| \leq \sum_{l=1}^N |\mathbb{E}f(\Phi_l) - \mathbb{E}f(\Phi_{l-1})| \leq 2Kpn \times \frac{M_3}{6} \frac{C(c, K, v, \tau)}{n^{2.5}} \to 0.$$

$\square$

# D   Existence of solution under general conditions

**Lemma D.1.** *For any $z \in \mathbb{C}_+$ and $r \in [K]$ and $\tau > 0$, we have $\Im(zg_r^\tau(z)) \geq 0$.*

*Proof.* Note that by definition, $g_{rn}^\tau(\cdot)$ are Stieltjes Transforms of measures on subsets of positive reals. In particular, this implies that $\Im(zg_{rn}^\tau(z)) \geq 0$. By Theorem 4.5, for a fixed $\tau > 0$, we have $g_{rn}^\tau \to g_r^\tau$ where the convergence is uniform in any compact subset of $\mathbb{C}_+$. Therefore we have $\Im(zg_r^\tau(z)) \geq 0$. $\square$

## D.1   Proof of Step7 and Step8 of Theorem 4.7

*Proof.* For any fixed $\tau > 0$ and $r \in [K]$, note that $|h_{rn}^\tau(z)| \leq C_0/v$ implies that $|h_r^\tau(z)| \leq C_0/v$. So using arguments similar to those used in Theorem 4.3 and Montel's Theorem, $\mathcal{H}_r = \{h_r^\tau : \tau > 0\}$ and $\mathcal{G}_r = \{g_r^\tau : \tau > 0\}$ are normal families for any $r \in [K]$. Let $x_r^\infty$ and $y_r^\infty$ be any subsequential limit points corresponding to the subsequences $\{\mathbf{h}^{\tau_n}\}_{n=1}^\infty$ and $\{\mathbf{g}^{\tau_n}\}_{n=1}^\infty$ resepctively where $\tau_n \to \infty$.

Note that $x_r^\infty$ is a Stieltjes Transform by Theorem 2.2. In particular, this implies that for any $z \in \mathbb{C}_+$, we must have $\Im(x_r^\infty(z)) > 0$ by a basic property of Stieltjes Transforms. For large

$n$, we have the following bound.

$$\left|\frac{\theta_r}{-z(1 + c\boldsymbol{\theta}^T\mathbf{h}^{\tau_n})}\right| \leq \frac{1}{c|z|\Im(h_r^{\tau_n}(z))} \rightarrow \frac{1}{c|z|\Im(x_r^\infty(z))} < \infty. \tag{74}$$

Also, we have

$$\left|\frac{\theta_r}{-z(1 + c\boldsymbol{\theta}^T\mathbf{x}^\infty)}\right| \leq \frac{1}{c|z|\Im(x_r^\infty(z))} < \infty. \tag{75}$$

Thus we have

$$|O_r(z, c\mathbf{h}^{\tau_n}, G^{\tau_n}) - O_r(z, c\mathbf{x}^\infty, G)| \tag{76}$$
$$\leq |O_r(z, c\mathbf{h}^{\tau_n}, G^{\tau_n}) - O_r(z, c\mathbf{h}^{\tau_n}, G)| + |O_r(z, c\mathbf{h}^{\tau_n}, G) - O_r(z, \mathbf{x}^\infty, G)|$$
$$= \left|\int \frac{\theta_r}{-z(1 + c\boldsymbol{\theta}^T\mathbf{h}^{\tau_n})}d\{G^{\tau_n}(\boldsymbol{\theta}) - G(\boldsymbol{\theta})\}\right| + \left|\int \frac{\theta_r dG(\boldsymbol{\theta})}{-z(1 + c\boldsymbol{\theta}^T\mathbf{h}^{\tau_n})} - \int \frac{\theta_r dG(\boldsymbol{\theta})}{-z(1 + c\boldsymbol{\theta}^T\mathbf{x}^\infty)}\right|.$$

Noting that $G^{\tau_n} \xrightarrow{d} G$ and applying DCT, we therefore have $\mathbf{O}(z, c\mathbf{h}^{\tau_n}, G^{\tau_n}) \rightarrow \mathbf{O}(z, c\mathbf{x}^\infty, G)$. By Theorem 4.5, for any $n \in \mathbb{N}$, we have $\mathbf{g}^{\tau_n} = \mathbf{O}(z, c\mathbf{h}^{\tau_n}, G^{\tau_n})$. Since $\mathbf{g}^{\tau_n} \rightarrow \mathbf{y}^\infty$, we therefore have $\mathbf{y}^\infty = \mathbf{O}(z, c\mathbf{x}^\infty, G)$.

Similarly, we observe that $y_r^\infty$ is a Stieltjes Transform by Theorem 2.2 which implies, in particular, that $\Im(y_r^\infty(z)) > 0$ for any $z \in \mathbb{C}_+$. For large $n$, we have the following bound.

$$\left|\frac{\lambda_r}{-z(1 + \boldsymbol{\lambda}^T\mathbf{g}^{\tau_n})}\right| \leq \frac{1}{|z|\Im(g_r^{\tau_n}(z))} \rightarrow \frac{1}{|z|\Im(y_r^\infty(z))} < \infty. \tag{77}$$

Also, we have

$$\left|\frac{\lambda_r}{-z(1 + \boldsymbol{\lambda}^T\mathbf{y}^\infty)}\right| \leq \frac{1}{|z|\Im(y_r^\infty(z))} < \infty. \tag{78}$$

Now we try to establish

$$|O_r(z, \mathbf{g}^{\tau_n}, H^{\tau_n}) - O_r(z, \mathbf{y}^\infty, H)| \tag{79}$$
$$\leq |O_r(z, \mathbf{g}^{\tau_n}, H^{\tau_n}) - O_r(z, \mathbf{g}^{\tau_n}, H)| + |O_r(z, \mathbf{g}^{\tau_n}, H) - O_r(z, \mathbf{y}^\infty, H)|$$
$$= \left|\int \frac{\lambda_r}{-z(1 + \boldsymbol{\lambda}^T\mathbf{g}^{\tau_n})}d\{H^{\tau_n}(\boldsymbol{\lambda}) - H(\boldsymbol{\lambda})\}\right| + \left|\int \frac{\lambda_r dH(\boldsymbol{\lambda})}{-z(1 + \boldsymbol{\lambda}^T\mathbf{g}^{\tau_n})} - \int \frac{\lambda_r dH(\boldsymbol{\lambda})}{-z(1 + \boldsymbol{\lambda}^T\mathbf{y}^\infty)}\right|.$$

Noting that $H^{\tau_n} \xrightarrow{d} H$ and applying DCT, we therefore have $\mathbf{O}(z, \mathbf{g}^{\tau_n}, H^{\tau_n}) \rightarrow \mathbf{O}(z, \mathbf{y}^\infty, H)$. By Theorem 4.5, for any $n \in \mathbb{N}$, we have $\mathbf{h}^{\tau_n}(z) = \mathbf{O}(z, \mathbf{g}^{\tau_n}(z), H^{\tau_n})$. Since $\mathbf{h}^{\tau_n} \rightarrow \mathbf{x}^\infty$, we therefore have $\mathbf{x}^\infty(z) = \mathbf{O}(z, \mathbf{y}^\infty(z), H)$.

Thus we see that any (pair of) subsequential limit $(\mathbf{x}^\infty, \mathbf{y}^\infty)$ satisfies (9), (11) and (12). By Theorem 4.2, all subsequential points must coincide which we will denote as $\mathbf{h}^\infty, \mathbf{g}^\infty$. This establishes (1) and (2) of Theorem 4.7.

We now show that $s^\tau(z) \xrightarrow{a.s.} s_F(z)$ where $s_F(z)$ is defined in (8). Note that

$$|s^\tau(z) - s_F(z)| \tag{80}$$

$$= \left| \int \frac{dH^\tau(\boldsymbol{\lambda})}{-z(1 + \boldsymbol{\lambda}^T \mathbf{g}^\tau(z))} - \int \frac{dH(\boldsymbol{\lambda})}{-z(1 + \boldsymbol{\lambda}^T \mathbf{g}^\infty(z))} \right|$$

$$\leq \left| \int \frac{1}{-z(1 + \boldsymbol{\lambda}^T \mathbf{g}^\tau(z))} d\{H^\tau(\boldsymbol{\lambda}) - H(\boldsymbol{\lambda})\} \right| +$$

$$\int \left| \frac{1}{-z(1 + \boldsymbol{\lambda}^T \mathbf{g}^\tau(z))} - \frac{1}{-z(1 + \boldsymbol{\lambda}^T \mathbf{g}^\infty(z))} \right| dH(\boldsymbol{\lambda}).$$

Note that for any $r \in [K]$, $\Im(zg_r^\tau(z)) \geq 0$ as $g_r^\tau(\cdot)$ are Stieltjes Transforms of measures on subsets of positive reals. Since $g_r^\tau \to g_r^\infty$, we also have $\Im(zg_r^\infty(z)) \geq 0$. Therefore, the integrands associated in the above expansion are bounded by $1/\Im(z)$ and $2/\Im(z)$ respectively. Thus, by application of DCT and using the fact that $H^\tau \xrightarrow{d} H$, we get $|s^\tau(z) - s_F(z)| \to 0$. This establishes the equivalence between (8) and (13).

Finally, we will show that $s_F(\cdot)$ satisfies the conditions of Theorem 2.1. From Lemma 4.2, we have

$$|\mathrm{i}y(\mathbf{h}^\infty(\mathrm{i}y))^T \mathbf{g}^\infty(\mathrm{i}y)| \leq \sum_{r=1}^K y|h_r^\infty(\mathrm{i}y)| \, |g_r^\infty(\mathrm{i}y)| \leq Ky \frac{C_0}{y} \frac{C_0}{y} \to 0 \text{ as } y \to \infty. \tag{81}$$

Thus using (13), we have

$$zs_F(z) = -z(z^{-1} + \mathbf{h}^T(z)\mathbf{g}(z)) \implies \lim_{y \to \infty} \mathrm{i}ys_F(\mathrm{i}y) = -1. \tag{82}$$

This concludes the proof of Step7 and Step8 of Theorem 4.7. $\qquad\square$

## D.2 Truncation, Centralization and Rescaling Steps

### Impact of spectral truncation of scaling matrices

Let $H_r$ and $G_r$ be the marginal distributions of $H$ and $G$ respectively for $r \in [K]$. Using

(28), (30), we observe that

$$\|F^{S_n} - F^{S_n^\tau}\| \le \frac{1}{p} \operatorname{rank}(S_n - S_n^\tau) \tag{83}$$

$$\le \frac{2}{p} \operatorname{rank}\left(\sum_{r=1}^{K} U_n^{(r)} Z_n^{(r)} V_n^{(r)} - \sum_{r=1}^{K} U_n^{r,\tau} Z_n^{(r)} V_n^{r,\tau}\right)$$

$$\le \frac{2}{p} \sum_{r=1}^{K} \operatorname{rank}\left(U_n^{(r)} Z_n^{(r)} V_n^{(r)} - U_n^{r,\tau} Z_n^{(r)} V_n^{r,\tau}\right)$$

$$\le \frac{2}{p} \sum_{r=1}^{K} \left(\operatorname{rank}(U_n^{(r)} - U_n^{r}(\tau)) + \operatorname{rank}(V_n^{(r)} - V_n^{r}(\tau))\right)$$

$$= 2 \sum_{r=1}^{K} \left((1 - H_{rn}(\tau)) + (1 - G_{rn}(\tau))\right)$$

$$\longrightarrow 2K \sum_{r=1}^{K} (1 - H_r(\tau) + 1 - G_r(\tau)) \longrightarrow 0, \text{ as } \tau \to \infty.$$

Here we used the fact that $\tau > 0$ was chosen in Step1 of Theorem 4.7 such that $(\tau, \ldots, \tau)$ is a continuity point of $H$.

**Truncation of innovation entries**

Again using (28), (30), we have

$$\|F^{S_n^\tau} - F^{\hat{S}_n}\| \le \frac{1}{p} \operatorname{rank}(S_n^\tau - \hat{S}_n) \tag{84}$$

$$\le \frac{2}{p} \operatorname{rank}\left(\sum_{r=1}^{K} U_n^{(r)}(\tau) Z_n^{(r)} V_n^{(r)}(\tau) - \sum_{r=1}^{K} U_n^{(r)}(\tau) \hat{Z}_n^{(r)} V_n^{(r)}(\tau)\right)$$

$$\le \frac{2}{p} \sum_{r=1}^{K} \operatorname{rank}\left(U_n^{(r)}(\tau)(Z_n^{(r)} - \hat{Z}_n^{(r)}) V_n^{(r)}(\tau)\right)$$

$$\le \frac{2}{p} \sum_{r=1}^{K} \operatorname{rank}(Z_n^{(r)} - \hat{Z}_n^{(r)}).$$

**Remark D.1.** *Note that these results hold even if the scaling matrices are not necessarily diagonal.*

Recall that $z_{ij}^{(r)}$ and $\hat{z}_{ij}^{(r)}$ denote the $ij^{th}$ elements of $Z^{(r)}$ and $\hat{Z}^{(r)}$, respectively. For $r \in [K]$, define $I_{ij}^{(r)} := \mathbb{1}_{\{z_{ij}^{(r)} \ne \hat{z}_{ij}^{(r)}\}} = \mathbb{1}_{\{|z_{ij}^{(r)}| > n^b\}}$ where $b$ is defined in **A2**. Using (29), we have

$$\operatorname{rank}(Z_r - \hat{Z}_r) \le \sum_{ij} I_{ij}^{(r)}.$$

Now observe that

$$\mathbb{P}(I_{ij}^{(r)} = 1) = \mathbb{P}(|z_{ij}^{(r)}| > n^b) \le \frac{\mathbb{E}|z_{ij}^{(r)}|^{2+\eta_0}}{n^{b(2+\eta_0)}} \le \frac{M_{2+\eta_0}}{n^{b(2+\eta_0)}}.$$

Since $\frac{1}{2+\eta_0} < b < \frac{1}{2}$, we have

$$\frac{1}{p} \sum_{i,j} \mathbb{P}(I_{ij}^{(r)} = 1) \leq \frac{npM_{2+\eta_0}}{pn^{b(2+\eta_0)}} \longrightarrow 0.$$

Also, we have $\operatorname{Var} I_{ij}^{(r)} \leq \mathbb{P}(I_{ij}^{(r)} = 1)$. For arbitrary $\epsilon > 0$, and large enough $n$, we must have $\sum_{i,j} \operatorname{Var} I_{ij}^{(r)} \leq p\epsilon/2$. Finally we use Bernstein's Inequality to get the following bound:

$$\begin{aligned}
\mathbb{P}\left(\frac{1}{p}\sum_{i,j} I_{ij}^{(r)} > \epsilon\right) &\leq \mathbb{P}\left(\sum_{i,j}(I_{ij}^{(r)} - \mathbb{P}(I_{ij}^{(r)} = 1)) > \frac{p\epsilon}{2}\right) \\
&\leq 2\exp\left(-\frac{p^2\epsilon^2/4}{2(p\epsilon/2 + \sum_{i,j} \operatorname{Var} I_{ij}^{(r)})}\right) \\
&\leq 2\exp\left(-\frac{p^2\epsilon^2/4}{2(p\epsilon/2 + p\epsilon/2)}\right) = 2\exp\left(-\frac{p\epsilon}{8}\right).
\end{aligned}$$

By Borel Cantelli lemma, $\frac{1}{p}\sum_{ij} I_{ij}^{(r)} \xrightarrow{a.s.} 0$ and thus

$$\frac{1}{p}\operatorname{rank}(Z_n^{(r)} - \hat{Z}_n^{(r)}) \xrightarrow{a.s.} 0. \tag{85}$$

Combining this with (84), we have $||F^{S_n^\tau} - F^{\hat{S}_n}|| \xrightarrow{a.s.} 0$.

**Centering of entries**

The next result to be proved is $||F^{\hat{T}_n} - F^{\tilde{T}_n}|| \xrightarrow{a.s.} 0$.

Define $\check{W}_n^{(r)} = (\check{w}_{ij}^{(r)}) := (w_{ij}^{(r)} \mathbb{1}_{\{w_{ij}^{(r)} > n^a\}})$ for $r \in [K]$. Similar to (85), we have

$$\frac{1}{p}\operatorname{rank} \check{W}_n^{(r)} = \frac{1}{p}\operatorname{rank}(W_n^{(r)} - \hat{W}_n^{(r)}) \xrightarrow{a.s.} 0. \tag{86}$$

Therefore, we see that

$$\begin{aligned}
||F^{\hat{T}_n} - F^{\tilde{T}_n}|| &\leq \frac{1}{p}\operatorname{rank}(\hat{T}_n - \tilde{T}_n) \tag{87} \\
&\leq \frac{1}{p}\sum_{r=1}^{K}\operatorname{rank}\left(U_n^{(r)}(\tau)\hat{W}_n^{(r)}V_n^{(r)}(\tau) - U_n^{(r)}(\tau)\tilde{W}_n^{(r)}V_n^{(r)}(\tau)\right) \\
&\leq \frac{1}{p}\sum_{r=1}^{K}\operatorname{rank}(\hat{W}_n^{(r)} - \tilde{W}_n^{(r)}) \\
&= \frac{1}{p}\sum_{r=1}^{K}\operatorname{rank}(\mathbb{E}\hat{W}_n^{(r)}) \\
&= \frac{1}{p}\sum_{r=1}^{K}\operatorname{rank}(\mathbb{E}\check{W}_n^{(r)}) \longrightarrow 0,
\end{aligned}$$

where the second last inequality follows since $\tilde{W}_n^{(r)} = \hat{W}_n^{(r)} - \mathbb{E}\hat{W}_n^{(r)}$ and the last equality is due to the fact that $\mathbf{0} = \mathbb{E}W_n^{(r)} = \mathbb{E}\hat{W}_n^{(r)} + \mathbb{E}\check{W}_n^{(r)}$ and the convergence is due to (86).

**Impact of rescaling**

We will now show that $L(F^{\tilde{T}_n}, F^{\ddot{T}_n}) \xrightarrow{a.s.} 0$. Recall from Step5 of Theorem 4.7 that $\rho_n^2 :=$

$\text{var}(\tilde{w}_{11}^{(1)})$. It is clear that $\rho_n \to 1$. From (34), we have

$$L^4(F^{\tilde{T}_n}, F^{\ddot{T}_n}) \le \frac{2}{p^2} \operatorname{trace}\left(\frac{1}{n}\tilde{Y}_n\tilde{Y}_n^* + \frac{1}{n}\ddot{Y}_n\ddot{Y}_n^*\right) \times \operatorname{trace}\left(\frac{1}{n}(\tilde{Y}_n - \ddot{Y}_n)(\tilde{Y}_n - \ddot{Y}_n)^*\right)$$

$$\le \frac{2\tau^4}{p^2} \operatorname{trace}\left(\frac{1}{n}\tilde{W}_n\tilde{W}_n^* + \frac{1}{n}\ddot{W}_n\ddot{W}_n^*\right) \times \operatorname{trace}\left(\frac{1}{n}(\tilde{W}_n - \ddot{W}_n)(\tilde{W}_n - \ddot{W}_n)^*\right)$$

$$\le \frac{2\tau^4}{p^2 n^2}\left((1 + \rho_n^{-2})\sum_{i,j}|\tilde{w}_{ij}|^2\right)\left(\sum_{i,j}|\tilde{w}_{ij}|^2(1 - \rho_n^{-1})^2\right) \xrightarrow{a.s.} 0.$$

This concludes the proof.

# E  Supporting Results for Uniqueness

**Lemma E.1.** *Osgood's Lemma: A continuous function of several complex variables is holomorphic if it is holomorphic separately in every variable.*

**Lemma E.2.** *Let $D \subset \mathbb{C}$ be a domain. For $j \in [n]$, let $f_j : D^m \to \mathbb{C}$ be holomorphic. Then, $\boldsymbol{f} : D^m \to \mathbb{C}^n$ is holomorphic.*

*Proof.* We will show that $\mathbf{f}$ is Fréchet differentiable. Denote $V = \mathbb{C}^m$ and $W = \mathbb{C}^n$. Fix $\mathbf{z} \in V$. For any $j \in [n]$, we have

$$\lim_{\mathbf{h}\to 0} \frac{|f_j(\mathbf{z} + \mathbf{h}) - f_j(\mathbf{z}) - \nabla f_j(\mathbf{z})\mathbf{h}|}{||\mathbf{h}||_V} = 0.$$

Let $A^{(\mathbf{z})} \in \mathbb{C}^{n \times m}$ be a matrix with $A_{j\cdot}^{(\mathbf{z})} = \nabla f_j(\mathbf{z})$. As $W$ is a finite dimensional Banach space, all norms are equivalent. So let $||\mathbf{w}||_W = \sum_{j=1}^n |w_j|$. Then, we have

$$\lim_{\mathbf{h}\to 0} \frac{||\mathbf{f}(\mathbf{z} + \mathbf{h}) - \mathbf{f}(\mathbf{z}) - A^{\mathbf{z}}\mathbf{h}||_W}{||\mathbf{h}||_V} = \sum_{j=1}^n \lim_{\mathbf{h}\to 0}\left(\frac{|f_j(\mathbf{z} + \mathbf{h}) - f_j(\mathbf{z}) - \nabla f_j(\mathbf{z})\mathbf{h}|}{||\mathbf{h}||_V}\right) = 0.$$

$\square$

**Lemma E.3.** *Recall the definition of $\boldsymbol{O}(\cdot)$ from (7). Then, $\boldsymbol{f} : \mathbb{C}_+^K \to \mathbb{C}_+^K; \quad \boldsymbol{f}(\boldsymbol{h}) = \boldsymbol{O}(z, c\boldsymbol{h}, G)$ is a holomorphic function of $\boldsymbol{h}$ and $\boldsymbol{e} : \mathbb{C}_+^K \to \mathbb{C}_+^K; \quad \boldsymbol{e}(\boldsymbol{g}) = \boldsymbol{O}(z, \boldsymbol{g}, H)$ is a holomorphic function of $\boldsymbol{g}$.*

*Proof.* Note that $\mathbf{f}$ is a K-variate function. For each $j \in [K]$, first we show continuity of $f_j$ and then show that $f_j$ is holomorphic separately in each coordinate. By Osgood's Lemma the proof will be complete.

For arbitrary $\mathbf{h} \in \mathbb{C}_+^K$, there exists $\epsilon > 0$ such that $\mathcal{B} = B(\mathbf{h}; \epsilon) \subset \mathbb{C}_+^K$. Therefore, there

exists $M_0 > 0$ such that $\Im(\tilde{h}_r) \geq M_0$ for all $r \in [K]$ whenever $\tilde{\mathbf{h}} \in \mathcal{B}$. Therefore,

$$
\begin{aligned}
\left| f_j(\mathbf{h}) - f_j(\tilde{\mathbf{h}}) \right| &= \left| \int \frac{\theta_j dG(\boldsymbol{\theta})}{-z(1 + c\boldsymbol{\theta}^T \mathbf{h})} - \int \frac{\theta_j dG(\boldsymbol{\theta})}{-z(1 + c\boldsymbol{\theta}^T \tilde{\mathbf{h}})} \right| \\
&\leq \frac{1}{|z|} \sum_{r=1}^K |h_r - \tilde{h}_r| \int \frac{\theta_j \theta_r dG(\boldsymbol{\theta})}{(1 + c\boldsymbol{\theta}^T \mathbf{h})(1 + c\boldsymbol{\theta}^T \tilde{\mathbf{h}})} \\
&\leq \frac{1}{|z| M_0^2} \| \mathbf{h} - \tilde{\mathbf{h}} \|_1 \\
\implies \| \mathbf{f}(\mathbf{h}) - f(\tilde{\mathbf{h}}) \|_1 &\leq \frac{K}{|z| M_0^2} \| \mathbf{h} - \tilde{\mathbf{h}} \|_1.
\end{aligned}
$$

This establishes continuity of $\mathbf{f}$ at $\mathbf{h}$. For the next part, we will show that for every $j \in [K]$, $f_j(h_r; \mathbf{h}_{-r})$ is holomorphic in $h_r$ for each $r \in [K]$. The notation indicates that we are keeping all $K$ coordinates of $\mathbf{h}$ fixed except the $r^{th}$ one. Let $\gamma$ be any closed curve in $\mathbb{C}_+$. Then, there exists $M_1 > 0$ such that $\Im(h_r) \geq M_1 > 0$ for $h_r \in \gamma$. Then we have

$$
\left| \frac{\theta_j}{1 + c\boldsymbol{\theta}^T \mathbf{h}} \right| \leq \frac{1}{\Im(h_j)} \leq \begin{cases} 1/M_1 & \text{if } r = j, \\ 1/\Im(h_j) & \text{if } r \neq j. \end{cases}
$$

In either case, we can interchange the two integrals by Fubini. Thus by Cauchy and Morera's Theorem,

$$
\int_\gamma f_j(h_r; \mathbf{h}_{-r}) dh_r = \frac{1}{-z} \int_\gamma \int \frac{\theta_j dG(\boldsymbol{\theta})}{1 + c\boldsymbol{\theta}^T \mathbf{h}} dh_r = \frac{1}{-z} \int \int_\gamma \frac{\theta_j}{1 + c\boldsymbol{\theta}^T \mathbf{h}} dh_r dG(\boldsymbol{\theta}) = 0.
$$

This establishes that $f_j$ is holomorphic by Osgood's Lemma. Finally by Lemma E.2, $\mathbf{f}$ itself is holomorphic as well. The proofs for $\mathbf{g}$ are similar. $\qquad \square$

**Lemma E.4.** *Recall the definitions of $\boldsymbol{P}_z, \boldsymbol{Q}_z$ from (15). $\boldsymbol{P}_z, \boldsymbol{Q}_z$ are holomorphic functions.*

*Proof.* Writing $\mathbf{f}(\mathbf{h}) = \mathbf{O}(z, c\mathbf{h}, G)$ as in Lemma E.3. Then, we denote the $j^{th}$ coordinate of $\boldsymbol{P}_z$ as

$$
p_j(\mathbf{h}) = \int \frac{\lambda_j dH(\boldsymbol{\lambda})}{-z(1 + \boldsymbol{\lambda}^T \mathbf{O}(z, c\mathbf{h}, G))} = \int \frac{\lambda_j dH(\boldsymbol{\lambda})}{-z(1 + \boldsymbol{\lambda}^T \mathbf{f}(\mathbf{h}))}.
$$

Since $p_j$ is K-variate, we will once again apply Osgood's Lemma. To show that $p_j$ is continuous, let $\mathbf{h} \in \mathbb{C}_+^K$ be arbitrary. Then there exists $\epsilon > 0$ such that $\mathcal{B} = B(\mathbf{h}; \epsilon) \subset \mathbb{C}_+^K$. Then since $\mathbf{f}$ is analytic by Lemma E.3, there exists $M_2 > 0$ such that for all $r \in [K]$, $\Im(f_r(\tilde{\mathbf{h}})) \geq M_2$

whenever $\tilde{\mathbf{h}} \in \mathcal{B}$. As a result, we see that

$$
\begin{aligned}
|p_j(\mathbf{h}) - p_j(\tilde{\mathbf{h}})| &= \left| \int \frac{\lambda_j dH(\boldsymbol{\lambda})}{-z(1 + \boldsymbol{\lambda}^T \mathbf{f}(\mathbf{h}))} - \int \frac{\lambda_j dH(\boldsymbol{\lambda})}{-z(1 + \boldsymbol{\lambda}^T \mathbf{f}(\tilde{\mathbf{h}}))} \right| \\
&\leq \sum_{r=1}^{K} |f_r(\mathbf{h}) - f_r(\tilde{\mathbf{h}})| \int \frac{\lambda_j \lambda_r dH(\boldsymbol{\lambda})}{|z(1 + \boldsymbol{\lambda}^T \mathbf{f}(\mathbf{h}))(1 + \boldsymbol{\lambda}^T \mathbf{f}(\tilde{\mathbf{h}}))|} \\
&\leq ||\mathbf{f}(\mathbf{h}) - \mathbf{f}(\tilde{\mathbf{h}})||_1 \frac{1}{|z| \Im(f_j(\mathbf{h})) \Im(f_r(\tilde{\mathbf{h}}))} \\
&\leq \frac{1}{|z| M_2^2} ||\mathbf{f}(\mathbf{h}) - \mathbf{f}(\tilde{\mathbf{h}})||_1 \\
\implies ||P_z(\mathbf{h}) - P_z(\tilde{\mathbf{h}})||_1 &\leq \frac{K}{|z| M_2^2} ||\mathbf{f}(\mathbf{h}) - \mathbf{f}(\tilde{\mathbf{h}})||_1.
\end{aligned}
$$

This establishes continuity of $\mathbf{P}_z$ at $\mathbf{h}$. Now, we will show that $p_j(h_r; \mathbf{h}_{-r})$ is a holomorphic function of $h_r$ for each $r \in [K]$ keeping $\mathbf{h}_{-r}$ fixed. Let $\gamma$ be an arbitrary closed curve in $\mathbb{C}_+$. Since $f_j$ is holomorphic, there exists $M_3 > 0$ such that $\Im(f_j(h_r; \mathbf{h}_{-r})) \geq M_3$ whenever $h_r \in \gamma$. Note that

$$
\left| \frac{\lambda_j}{-z(1 + \boldsymbol{\lambda}^T \mathbf{f}(h_r; \mathbf{h}_{-r}))} \right| \leq \frac{1}{|z| M_3}.
$$

Therefore by Fubini and Cauchy,

$$
\int_\gamma p_j(h_r; \mathbf{h}_{-r}) dh_r = \int_\gamma \int p_j(h_r; \mathbf{h}_{-r}) dH(\boldsymbol{\lambda}) dh_r = \int \int_\gamma p_j(h_r; \mathbf{h}_{-r}) dh_r dH(\boldsymbol{\lambda}) = \int 0 \, dH(\boldsymbol{\lambda}) = 0.
$$

By Morera's theorem, $p_j$ is holomorphic in $h_r$ keeping $\mathbf{h}_{-r}$ fixed. Thus by Osgood's Lemma, $p_j$ is holomorphic in $\mathbb{C}_+^K$. Finally by Lemma E.2, $\mathbf{P}_z$ itself is holomorphic as well. The proof for $\mathbf{Q}_z$ is similar. $\qquad\square$

# F   Miscellaneous Proofs

## F.1   Proof of Theorem 7.1

*Proof.* Recall from (2) that $S_n = \frac{1}{n} X_n X_n^*$. Analogous to $X_n \in \mathbb{C}^{p \times n}$, we define $Y_{r_0, s_0} \in \mathbb{C}^{p \times n}$ for $r_0, s_0 \in [K]$ as follows:

$$
Y_{r_0, s_0} := A_{r_0 \to 1}^{\frac{1}{2}} Z_1 B_{s_0 \to 1}^{\frac{1}{2}} + \ldots + A_{r_0 \to K}^{\frac{1}{2}} Z_K B_{s_0 \to K}^{\frac{1}{2}}.
$$

So, $Y_{r_0, s_0}$ is a version of $X_n$ such that the row and the column scaling matrices used across the $K$ coordinates commute among themselves. The sample covariance matrix of $Y_{r_0, s_0}$ is

$$
M_{r_0, s_0} := \frac{1}{n} Y_{r_0, s_0} Y_{r_0, s_0}^*.
$$

Using (28),(30), we observe that,

$$\|F^{S_n} - F^{M_{r_0,s_0}}\| \le \frac{1}{p} \operatorname{rank}(S_n - M_{r_0,s_0})$$

$$\le \frac{2}{p} \operatorname{rank}(X_n - Y_{r_0,s_0})$$

$$\le \frac{2}{p} \sum_{r=1}^{K} \left( \operatorname{rank}(A_r^{\frac{1}{2}} - A_{r_0 \to r}^{\frac{1}{2}}) + \operatorname{rank}(B_r^{\frac{1}{2}} - B_{s_0 \to r}^{\frac{1}{2}}) \right)$$

$$\le \frac{4}{p} \sum_{r=1}^{K} \left( \operatorname{rank}(P_r - P_{r_0}) + \operatorname{rank}(Q_r - Q_{s_0}) \right).$$

Therefore, under conditions **C1** or **C2** of the theorem, $F^{S_n}$ almost surely shares the same weak limit as that of $F^{M_{r_0,s_0}}$. Note that $M_{r_0,s_0}$ is an analog of our original sample covariance matrix $S_n$ but the components of $M_{r_0,s_0}$ satisfy all the conditions (including **T3**) of the Theorem 4.1. The almost sure limit of $F^{M_{r_0,s_0}}$ is characterized in the theorem.

To show sufficiency of **C3**, note that the deterministic equivalent for the resolvent $Q(z) = (S_n - zI_p)^{-1}; z \in \mathbb{C}_+$ from Theorem 4.4 was as follows:

$$\bar{Q}(z) = \left( -zI_p + \sum_{r=1}^{K} A_r \left( \frac{1}{n} \operatorname{trace}\{B_r[I_n + c_n \sum_{s=1}^{K} \mathbb{E}h_{sn}(z)B_s]^{-1}\} \right) \right)^{-1}.$$

We introduce some efficient wrapper functions to compress the notation. We will denote $\mathbf{A} = (A_1, \ldots, A_K)$ and $\mathbf{B} = (B_1, \ldots, B_K)$. For fixed $z \in \mathbb{C}_+$ and p.s.d. matrices $B_r$ of dimension $n$, we denote

$$\mathbf{V}(\mathbf{B}) := \left( I_n + c_n \sum_{s=1}^{K} \mathbb{E}h_{sn}(z)B_s \right)^{-1},$$

$$U_r(\mathbf{B}) := \frac{1}{n} \operatorname{trace}\left( B_r \mathbf{V}(\mathbf{B}) \right).$$

With this, we have the following simpler expression for $\bar{Q}(z)$ as follows:

$$\bar{Q}(z) = \left( -zI_p + \sum_{r=1}^{K} U_r(z, \mathbf{B})A_r \right)^{-1}.$$

With $r_0, s_0 \in [K]$ from **C3**, we define an analogous version of $\bar{Q}(z)$ as follows:

$$R_{r_0,s_0}(z) := \left( -zI_p + \sum_{r=1}^{K} A_{r_0 \to r} \left( \frac{1}{n} \operatorname{trace}\{B_{s_0 \to r}[I_n + c_n \sum_{s=1}^{K} \mathbb{E}h_{sn}(z)B_{s_0 \to s}]^{-1}\} \right) \right)^{-1}$$

$$= \left( -zI_p + \sum_{r=1}^{K} U_r(\mathbf{B}_{s_0 \to})A_{r_0 \to r} \right)^{-1}$$

$$= \left( -zI_p + \sum_{r=1}^{K} A_{r_0 \to r} \int \frac{\theta_r dG_n(\boldsymbol{\theta})}{1 + c_n \boldsymbol{\theta}^T \mathbb{E}\mathbf{h}_n(z)} \right)^{-1}.$$

Fixing $r_0, s_0 \in [K]$ from C3, we will now refer to $R_{r_0,s_0}(\cdot)$ as $R(\cdot)$. Since $\mathbf{A}_{r_0 \to}$ and $\mathbf{B}_{s_0 \to}$ satisfy **T3**, $R(z)$ is an ideal candidate for a deterministic equivalent of $Q(z)$ and it is clear that,

using it instead of $\bar{Q}(z)$ will lead exactly to the same conclusions of Theorem 4.1. Thus, it suffices to show that under **C3** and **A1** of Assumptions 4.1.1, the following holds:

$$\left| \frac{1}{p} \operatorname{trace}(\bar{Q}(z) - R(z)) \right| \to 0. \tag{88}$$

Let $0 < v = \Im(z)$. The proof of (88) depends on the following interim results.

1. $|U_r(\mathbf{B}_{s_0 \to})|$ is uniformly (in $n$) bounded as a function of $z$ and $\tau$ from **A1**. More specifically,

$$|U_r(\mathbf{B}_{s_0 \to})| \leq \frac{|z|\tau}{v}. \tag{89}$$

2. The second result we need is the following:

$$|U_r(\mathbf{B}_{s_0 \to}) - U_r(\mathbf{B})| \to 0 \text{ as } n \to \infty. \tag{90}$$

3. The third result is as follows:

$$\max\{||R(z)||_{op}, ||\bar{Q}(z)||_{op}\} \leq \frac{1}{v}. \tag{91}$$

By (27), (89), (90) and (91) and **C3** we observe that

$$\left| \frac{1}{p} \operatorname{trace}(\bar{Q}(z) - R(z)) \right|$$

$$= \left| \frac{1}{p} \operatorname{trace} \left( R(z)\bar{Q}(z) \sum_{r=1}^{K} [U_r(\mathbf{B}_{s_0 \to})A_{r_0 \to r} - U_r(\mathbf{B})A_r] \right) \right|$$

$$\leq ||R(z)\bar{Q}(z)||_{op} \sum_{r=1}^{K} \frac{1}{\sqrt{p}} ||U_r(\mathbf{B}_{s_0 \to})A_{r_0 \to r} - U_r(\mathbf{B})A_r||_F$$

$$\leq \frac{1}{v^2} \sum_{r=1}^{K} |U_r(\mathbf{B}_{s_0 \to})| \times \frac{1}{\sqrt{p}} ||A_{r_0 \to r} - A_r||_F + \frac{1}{v^2} \sum_{r=1}^{K} |U_r(\mathbf{B}_{s_0 \to}) - U_r(\mathbf{B})| \times \frac{1}{\sqrt{p}} ||A_r||_F \to 0.$$

Here we used the fact that $||M||_F \leq ||M||_{op} ||M||_*$ and $||A_r||_{op}, ||A_{r_0 \to r}||_{op} \leq \tau$ by **A1**.

**Proof of (89):** Since $h_{sn}(\cdot)$ is the Stieltjes transform of some *positive* measure, we have $\Im(zh_{sn}(z)) \geq 0$ for any $s \in [K], z \in \mathbb{C}_+$. Therefore, we have

$$|U_r(\mathbf{B}_{s_0 \to})| = \left| \frac{1}{n} \operatorname{trace} \left( B_{s_0 \to}[I_n + \sum_{s=1}^{K} \mathbb{E}(c_n h_{sn})B_{s_0 \to s}]^{-1} \right) \right|$$

$$= \left| \int \frac{\theta_r dG_n(\boldsymbol{\theta})}{1 + c_n \langle \mathbb{E}\mathbf{h}_n, \boldsymbol{\theta} \rangle} \right|$$

$$\leq \left| \int \frac{|z|\theta_r dG_n(\boldsymbol{\theta})}{|z + c_n z \sum_{s=1}^{K} \mathbb{E}h_{sn}\theta_s|} \right|$$

$$\leq \frac{|z|\tau}{|\Im(z) + c_n z \sum_{s=1}^{K} \theta_s \Im(\mathbb{E}h_{sn})|} \leq \frac{|z|\tau}{v}.$$

**Proof of (90):** Note that

$$|U_r(\mathbf{B}_{s_0 \to}) - U_r(\mathbf{B})|$$

$$= \left| \frac{1}{n} \operatorname{trace} \left( B_r(V(\mathbf{B}_{s_0 \to}) - V(\mathbf{B})) \right) \right| + \left| \frac{1}{n} \operatorname{trace} \left( (B_{s_0 \to r} - B_r)V(\mathbf{B}_{s_0 \to}) \right) \right| = I + II.$$

We claim that $||V(\mathbf{B}_{s_0 \to})||_{op} \le |z|/v$. Let $\{\theta_{sj}\}_{j=1}^n$ denote the eigenvalues of $B_{s_0 \to s}$ (and also $B_s$). Therefore, for any $j \in [n]$, we have

$$\left| \frac{1}{1 + c_n \sum_{s=1}^K \mathbb{E}h_{sn}\theta_{sj}} \right| \le \frac{|z|}{|\Im(z) + c_n \sum_{s=1}^K \theta_{sj}\mathbb{E}h_{sn}|} \le \frac{|z|}{v}.$$

Thus, the term $II$ in the above expansion goes to 0 on using **C3** and basic inequalities related to traces of matrices. Similarly, we note that $V(\mathbf{B}) = z(zI_n + c_n \sum_{s=1}^K \mathbb{E}(zh_{sn})B_s)^{-1}$. Again using the fact that $\Im(zh_{sn}(z)) \ge 0$, we observe that

$$||V(\mathbf{B})||_{op} \le \frac{|z|}{|\Im(z + c_n \sum_{s=1}^K \mathbb{E}(zh_{sn}(z))B_s)|} \le \frac{|z|}{v}.$$

Now, analyzing the term $I$, for large $n$, we have

$$\left| \frac{1}{n} \operatorname{trace} \left( B_r(V(\mathbf{B}_{s_0 \to}) - V(\mathbf{B})) \right) \right|$$

$$= \left| \frac{1}{n} \operatorname{trace} \left( V(\mathbf{B})B_r V(\mathbf{B}_{s_0 \to}) \sum_{s=1}^K \mathbb{E}(c_n h_{sn})(B_s - B_{s_0 \to s}) \right) \right|$$

$$= ||V(\mathbf{B})B_r V(\mathbf{B}_{s_0 \to})||_{op} \sum_{s=1}^K \frac{1}{\sqrt{n}} |\mathbb{E}(c_n h_{sn}(z))| \times ||B_s - B_{s_0 \to s}||_F$$

$$\le \tau \left( \frac{|z|}{v} \right)^2 \frac{2cC_0}{v} \sum_{s=1}^K \frac{1}{\sqrt{n}} ||B_s - B_{s_0 \to s}||_F \longrightarrow 0, \text{ by } \mathbf{C3}.$$

**Proof of (91):** We have

$$U_r(\mathbf{B}_{s_0 \to}) = \int \frac{\theta_r dG_n(\boldsymbol{\theta})}{1 + c_n \langle \mathbb{E}\mathbf{h}_n, \boldsymbol{\theta} \rangle} = -zO_r(z, \mathbb{E}(c_n \mathbf{h}_n), G_n),$$

as per the notation developed in (7). By Lemma 4.1 and (90), for large $n$, we have

$$\bar{Q}(z)^{-1} = -(vI_p + \Pi) + \mathfrak{i}(-uI_p + \Psi),$$

where $\Pi$ and $\Psi$ are p.s.d. matrices. Thus, we have $||\bar{Q}(z)||_{op} \le 1/v$. The proof of $||R(z)||_{op} \le 1/v$ easily follows, as the *spatial and temporal components* of $R(z)$ commute.

For any $p \times p$ matrix $A$, by Cauchy-Schwarz we have

$$|\operatorname{trace}(A)| \le \sqrt{p \operatorname{trace}(A^*A)},$$

and when B is p.s.d, we have

$$| \operatorname{trace}(AB)| \leq ||A||_{op} \operatorname{trace}(B).$$

We also used the fact that $||A||_F^2 \leq ||A||_{op}||A||_*$. $\qquad \square$

## F.2 Proof of Theorem 5.1

*Proof.* The main result related to the L.S.D. holds as long as the innovation entries satisfy **T2**. Throughout this proof, we assume without loss of generality that the innovation entries are i.i.d. standard Gaussian. This is done, in particular, to access the results established in Corollary 6.3.

**Notation**: The innovation matrices $Z_n^{(r)}$ shall be denoted as $Z_r$ for simplicity. For $n \in \mathbb{N}$, denote $\mathcal{P}(n)$ to be the set of all the permutations of $[n]$. Let $\delta$ in $\mathcal{P}(p)$ and $U = \operatorname{diag}(u_1, \ldots, u_p)$ be a diagonal matrix of order $p$. Then, we define

$$U^\delta := \operatorname{diag}(u_{\delta_1}, \ldots, u_{\delta_p}).$$

We start with the observation that if $\{X_i\}_{i=1}^n$ are identically distributed non-negative random variables, then

$$X_1 \overset{stoc}{\geq} \frac{1}{n} \sum_{i=1}^n X_i, \tag{92}$$

where, $\overset{stoc}{\geq}$ denotes the stochastic ordering.

Let $Z_r; r \in [K]$ be some fixed realization of the innovation matrices and $\mathbf{x} \in \mathbb{C}^p$ be such that $||\mathbf{x}|| = 1$. Let $\mathcal{D} := (\mathbb{C}^{p \times p})^K \times (\mathbb{C}^{n \times n})^K$ and define the mapping

$$L : \mathcal{D} \to \mathbb{C}^n \quad L((U_r, V_r)_{r=1}^K) := \frac{1}{\sqrt{n}} \sum_{r=1}^K V_r Z_r^* U_r \mathbf{x}.$$

Clearly, $L$ is linear in $U_r, V_r$. Then, $QF((A_r, B_r)_{r=1}^K) := \mathbf{x}^* S_n \mathbf{x} = L^* L$ is jointly convex in $U_r, V_r, r \in [K]$. We now state a few distributional equivalence relationships.

1. For $r \in [K]$, let $\delta_r \in \mathcal{P}(p)$ and $\sigma_r \in \mathcal{P}(n)$. Then we have $V_r Z_r^* U_r \overset{d}{=} V_r^{\sigma_r} Z_r^* U_r^{\delta_r}$.

2. Combining everything, we have

$$QF(U_1, \ldots, U_K, V_1, \ldots, V_K) \overset{d}{=} QF(U_1^{\delta_1}, \ldots, U_K^{\delta_K}, V_1^{\sigma_1}, \ldots, V_K^{\sigma_K}).$$

For $r \in [K]$, let $\bar{U}_r = \left(\frac{1}{p} \operatorname{trace}(U_r)\right) I_p$ and $\bar{V}_r = \left(\frac{1}{n} \operatorname{trace}(V_r)\right) I_n$. Therefore,

$$\mathbf{x}^* S_n \mathbf{x} = QF((U_r, V_r)_{r=1}^K) \tag{93}$$

$$\overset{stoc}{\geq} \frac{1}{(p!)^K (n!)^K} \sum_{\delta_i \in \mathcal{P}(p)} \sum_{\sigma_j \in \mathcal{P}(n)} QF(U_1^{\delta_1}, \ldots, U_K^{\delta_K}, V_1^{\sigma_1}, \ldots, V_K^{\sigma_K})$$

$$\geq \frac{1}{(p!)^K} \sum_{\delta_i \in \mathcal{P}(p)} QF(U_{\pi_1}^{\delta_1}, \ldots, U_{\pi_K}^{\delta_K}, \bar{V}_1, \ldots, \bar{V}_K)$$

$$\geq QF(\bar{U}_1, \ldots, \bar{U}_K, \bar{V}_1, \ldots, \bar{V}_K).$$

where the first inequality follows due to (92) and the subsequent inequalities follow from applying convexity separately in each of the $2K$ coordinates of $QF(\cdot)$. Due to the conditions on $H$ and $G$ imposed in Theorem 4.1, there exists $\epsilon > 0$ such that

$$\liminf_{n \to \infty} \min_{r \in [K]} \left\{ \frac{1}{p} \operatorname{trace}(U_r), \frac{1}{n} \operatorname{trace}(V_r) \right\} \geq \epsilon.$$

By Corollary 6.3, when $0 < c < 1$, we have

$$\liminf_{n \to \infty} QF(\bar{U}_1, \ldots, \bar{U}_K, \bar{V}_1, \ldots, \bar{V}_K)$$

$$\geq \liminf_{n \to \infty} QF(\epsilon I_p, \ldots, \epsilon I_p, \epsilon I_n, \ldots, \epsilon I_n) \geq K \epsilon^2 (1 - \sqrt{c})^2 > 0.$$

Thus when $0 < c < 1$, for any fixed $\mathbf{x} \in \mathbb{C}^p$, we must have $x^* S_n x \overset{stoc}{\geq} \epsilon^2 (1 - \sqrt{c})^2$. This implies that the smallest eigen value is bounded away from 0 in the limiting sense. As a result, we have $F_c(\{0\}) = 0$ where the L.S.D. is denoted as $F_c$ to emphasize the role of $c > 0$. Let $F_c^{\bar{S}_n} \overset{d}{\to} \bar{F}_c$ where $\bar{F}_c$ denotes the L.S.D. of the dual sample covariance matrix $\bar{S}_n$. Utilizing the relationship between $F_c$ and $\bar{F}_c$, we have

$$F_c(\{0\}) = \left(1 - \frac{1}{c}\right) \delta_0(\{0\}) + \frac{1}{c} \bar{F}_c(\{0\}) \implies \bar{F}_c(\{0\}) = 1 - c. \tag{94}$$

This establishes the result when $c < 1$. When $c > 1$, by interchanging the roles of $(S_n, \bar{S}_n)$ and $(c, \frac{1}{c})$, we get $\bar{F}_c(\{0\}) = 0$ and therefore from (94), we have $F_c(\{0\}) = 1 - 1/c$.

Now we address the case when c = 1. From (93), we have

$$x^* S_n x \overset{stoc}{\geq} QF(\epsilon I_p, \ldots, \epsilon I_p, \epsilon I_n, \ldots, \epsilon I_n). \tag{95}$$

Let $T_n$ be the sample covariance matrix when $A_n^{(r)} = \epsilon I_p$ and $B_n^{(r)} = \epsilon I_n$ for all $r \in [K]$. Since (95) holds for any $||\mathbf{x}|| = 1$, stochastically we must have $S_n \underset{l\ddot{o}wner}{\geq} T_n$ or $S_n - T_n$ is stochastically a p.s.d matrix. Therefore, $F^{S_n}(t) \leq F^{T_n}(t)$ for any $t \in \mathbb{R}$. Recall that $F^{S_n} \overset{d}{\to} F^{c=1}$ and from Corollary 6.3, we have $F^{T_n} \overset{d}{\to} F^{MP,c=1}$ where $F^{MP,c=1}$ denotes the Marchenko-Pastur type limiting law for the given set of scaling matrices. If $X_n \overset{stoc}{\geq} Y_n$, $X_n \overset{d}{\to} X$ and $Y_n \overset{d}{\to} Y$, then $X \overset{stoc}{\geq} Y$. In particular, $F_X(t) \leq F_Y(t)$ for any $t \in \mathbb{R}$. Thus, $F^{c=1}(0, a) \leq F^{MP,c=1}(0, a) \to 0$ as $a \to 0$. The convergence to 0 follows since the Marchenko-Pastur type law derived in Corollary

[6.3](#) exhibits $1/\sqrt{t}$ behavior when $t \approx 0$. This proves the result for $c = 1$. $\qquad\square$

**Lemma F.1.** *In Theorem [5.1](#), when $c \geq 1$, we must have $\lim_{\epsilon\downarrow0} h_{r_0}(\mathtt{i}\epsilon) = \infty$ for some $r_0 \in [K]$.*

*Proof.* Suppose not, then there exists $M > 0$ such that for all $r \in [K]$, $|h_r(\mathtt{i}\epsilon)| \leq M$ for any $\epsilon > 0$. In that case, we have

$$\frac{1}{1 + cKM||\boldsymbol{\theta}||_2} \leq \frac{1}{1 + c|\boldsymbol{\theta}^T\mathbf{h}(\mathtt{i}\epsilon)|} \leq \left|\frac{1}{1 + c\boldsymbol{\theta}^T\mathbf{h}(\mathtt{i}\epsilon)}\right|. \tag{96}$$

Denoting $f(\epsilon; \boldsymbol{\theta}) = (1 + c\boldsymbol{\theta}\mathbf{h}(\mathtt{i}\epsilon))^{-1}$, we observe that $\Re(f(\epsilon; \boldsymbol{\theta})) \geq 0$ and $\Im(f(\epsilon; \boldsymbol{\theta})) \leq 0$. In light of this, we have

$$\lim_{\epsilon\downarrow0} \int |f(\epsilon; \boldsymbol{\theta})| \, dG(\boldsymbol{\theta}) = \lim_{\epsilon\downarrow0} \int f(\epsilon; \boldsymbol{\theta}) dG(\boldsymbol{\theta}) = 0.$$

The last equality follows from [(13)](#) and Theorem [5.1](#). Thus, when $c \geq 1$, we have

$$\lim_{\epsilon\downarrow0} \int \frac{dG(\boldsymbol{\theta})}{1 + c\boldsymbol{\theta}^T\mathbf{h}(\mathtt{i}\epsilon)} = 0.$$

Define $R := \{||\boldsymbol{\theta}||_2 \leq r/M\}$ where $r > 0$ is chosen such that $G(R) > 0$. This is possible due to the conditions on $G$. Finally, we arrive at a contradiction by shrinking $\epsilon$ to 0 since the quantity on the right converges to 0 as shown below:

$$0 < \frac{1}{1 + crK} \times G(R) \leq \int_R \frac{dG(\boldsymbol{\theta})}{1 + cM||\boldsymbol{\theta}||_2} \leq \int_{R\cup R^c} \frac{dG(\boldsymbol{\theta})}{1 + cM||\boldsymbol{\theta}||_2} \leq \int \frac{dG(\boldsymbol{\theta})}{\left|1 + c\boldsymbol{\theta}^T\mathbf{h}(\mathtt{i}\epsilon)\right|}.$$

$\qquad\square$

## F.3   Proof of Theorem [5.2](#)

*Proof.* Fix $\epsilon > 0$, $z \in \mathbb{C}_+$ and $H, G$ with unique solutions denoted by $\mathbf{h}, \mathbf{g}$. Let $(\tilde{H}, \tilde{G})$ be any other pair of distribution functions with corresponding unique solutions given by $\tilde{\mathbf{h}}, \tilde{\mathbf{g}}$. We will show that if $L(H, \tilde{H})$, $L(G, \tilde{G})$ are sufficiently small, then $||\mathbf{h} - \tilde{\mathbf{h}}||_1$ and $||\mathbf{g} - \tilde{\mathbf{g}}||_1$ are also small.

$$\begin{aligned}
|h_r - \tilde{h}_r| &= \frac{1}{|z|}\left|\int \frac{\lambda_r dH(\boldsymbol{\lambda})}{1 + \boldsymbol{\lambda}^T\mathbf{g}} - \int \frac{\lambda_r d\tilde{H}(\boldsymbol{\lambda})}{1 + \boldsymbol{\lambda}^T\tilde{\mathbf{g}}}\right| \\[2mm]
&\leq \frac{1}{|z|}\int \left|\frac{\lambda_r}{1 + \boldsymbol{\lambda}^T\mathbf{g}}\right| d\{H(\boldsymbol{\lambda}) - \tilde{H}(\boldsymbol{\lambda})\} + |z|\int \left|\frac{\lambda_r}{-z(1 + \boldsymbol{\lambda}^T\mathbf{g})} - \frac{\lambda_r}{-z(1 + \boldsymbol{\lambda}^T\tilde{\mathbf{g}})}\right| d\tilde{H}(\boldsymbol{\lambda}) \\[2mm]
&\leq \frac{\epsilon}{|z|K} + \sum_{j=1}^K |g_j - \tilde{g}_j|\int \left|\frac{\lambda_r\lambda_j}{(-z - z\boldsymbol{\lambda}^T\mathbf{g})(-z - z\boldsymbol{\lambda}^T\tilde{\mathbf{g}})}\right| d\tilde{H}(\boldsymbol{\lambda}) \\[2mm]
&\leq \frac{\epsilon}{|z|K} + \frac{D_0}{v^2}\sum_{j=1}^K |g_j - \tilde{g}_j|.
\end{aligned} \tag{97}$$

The last inequality follows as

$$\left|\frac{\lambda_r}{1 + \boldsymbol{\lambda}^T\mathbf{g}}\right| \leq \frac{1}{\Im(g_r)} < \infty.$$

By Lemma 4.1, we have $\Im(zg_k(z)) \geq 0$ and $\Im(z\tilde{g}_k(z))$ for any $k \in [K]$. Similarly, when $L(G, \tilde{G})$ is sufficiently small, the $j^{th}$ term in the second expression can be bounded above.

$$
\begin{aligned}
|g_j - \tilde{g}_j| &= \left| \int \frac{\theta_j dG(\boldsymbol{\theta})}{-z(1 + c\boldsymbol{\theta}^T \mathbf{h})} - \int \frac{\theta_j d\tilde{G}(\boldsymbol{\theta})}{-z(1 + c\boldsymbol{\theta}^T \tilde{\mathbf{h}})}) \right| \\
&\leq \frac{1}{|z|} \int \left| \frac{\theta_j}{1 + c\boldsymbol{\theta}^T \mathbf{h}} \right| d\{G(\boldsymbol{\theta}) - \tilde{G}(\boldsymbol{\theta})\} + \int \left| \frac{\theta_j}{-z(1 + c\boldsymbol{\theta}^T \mathbf{h})} - \frac{\theta_j}{-z(1 + c\boldsymbol{\theta}^T \tilde{\mathbf{h}})} \right| d\tilde{G}(\boldsymbol{\theta}) \\
&\leq \frac{1}{|z|} \int \left| \frac{\theta_j}{1 + c\boldsymbol{\theta}^T \mathbf{h}} \right| d\{G(\boldsymbol{\theta}) - \tilde{G}(\boldsymbol{\theta})\} + \sum_{s=1}^{K} |z(h_s - \tilde{h}_s)| \int \left| \frac{\theta_j \theta_s}{(-z - zc\boldsymbol{\theta}^T \mathbf{h})(-z - zc\boldsymbol{\theta}^T \tilde{\mathbf{h}})} \right| d\tilde{G}(\boldsymbol{\theta}) \\
&\leq \frac{\epsilon}{K|z|} + \frac{D_0}{v^2} \sum_{s=1}^{K} |z(h_s - \tilde{h}_s)| = \frac{\epsilon}{|z|K} + \frac{D_0|z|}{v^2} ||\mathbf{h} - \tilde{\mathbf{h}}||_1.
\end{aligned}
\tag{98}
$$

Again, the first term goes to 0 as $L(G, \tilde{G}) \to 0$ by DCT since

$$
\left| \frac{\theta_j}{1 + c\boldsymbol{\theta}^T \mathbf{h}} \right| \leq \frac{1}{c\Im(h_j)} < \infty.
$$

Combining everything, we get

$$
\begin{aligned}
|h_r - \tilde{h}_r| &\leq \frac{\epsilon}{|z|K} + \frac{D_0}{v^2} \sum_{j=1}^{K} \left( \frac{\epsilon}{|z|K} + \frac{D_0|z|}{v^2} ||\mathbf{h} - \tilde{\mathbf{h}}||_1 \right) \\
\implies ||\mathbf{h} - \tilde{\mathbf{h}}||_1 &\leq \frac{\epsilon}{|z|} + \frac{KD_0}{v^2} \left( \frac{\epsilon}{|z|} + \frac{KD_0|z|}{v^2} ||\mathbf{h} - \tilde{\mathbf{h}}||_1 \right) \\
&= \frac{\epsilon}{|z|} \left( 1 + \frac{KD_0}{v^2} \right) + \frac{K^2 D_0^2 |z|}{v^4} ||\mathbf{h} - \tilde{\mathbf{h}}||_1.
\end{aligned}
\tag{99}
$$

For every $\epsilon > 0$, there exists $\delta > 0$ such that the above holds whenever

$$
\max\{L(H, \tilde{H}), L(G, \tilde{G})\} < \delta.
$$

For $\epsilon = 1/n$, there exists $\tilde{H}_n, \tilde{G}_n, \tilde{\mathbf{h}}_n, \tilde{\mathbf{g}}_n$ such that the above inequality holds. Repeating the arguments in (16), $\{\tilde{\mathbf{h}}_n\}$ has a convergent subsequence, say $\{\tilde{\mathbf{h}}_{n_k}\}$ by *Montel's Theorem*. But any subsequential limit satisfies the below.

$$
\lim_{k \to \infty} ||\mathbf{h} - \tilde{\mathbf{h}}_{n_k}||_1 \leq \frac{K^2 D_0^2 |z|}{v^4} \lim_{k \to \infty} ||\mathbf{h} - \tilde{\mathbf{h}}_{n_k}||_1.
\tag{100}
$$

Choose $z = u + \mathbf{i}v$ with $|u| \leq v$ and $v \geq V_0$, where $V_0 := (2K^2 D_0^2)^{\frac{1}{3}}$. Then, for such $z \in \mathbb{C}_+$, it is easy to see by a contraction argument that we must have $\lim_{n \to \infty} ||\mathbf{h}(z) - \tilde{\mathbf{h}}_{n_k}(z)|| = 0$ for any subsequence. Therefore, we have $\lim_{k \to \infty} ||\mathbf{h}(z) - \tilde{\mathbf{h}}_n(z)|| = 0$. Similarly, we can show that $\lim_{k \to \infty} ||\mathbf{g}(z) - \tilde{\mathbf{g}}_n(z)|| = 0$. Now, noting that $\tilde{\mathbf{h}}_n(\cdot), \mathbf{h}(\cdot)$ are locally bounded (e.g. see proof of Theorem 4.3), the result can be extended to any $z \in \mathbb{C}_+$ by Theorem 2.3. $\qquad \square$

