# OpenReview forum: "L.S.D. of sample covariances of superposition of matrices with separable covariance structure"
_SLADS/Section_A — Accepted by SLADS_Section_A_

### Review · Reviewer_54Vb · 2026-05-27

**Summary Of Contributions:**

The paper investigates the LSD of sample covariance matrices derived from a sum of $K$ matrices, each possessing a separable (Kronecker-type) covariance structure in the high-dimensional regime ($p/n\rightarrow c \in (0,\infty)$). The authors characterize the L.S.D. using a system of Stieltjes transform equations and extend the results to cases where scaling matrices are not strictly commutative. This work bridges the gap between the traditional separable covariance model (K=1) and more complex, non-separable spatio-temporal interactions.

**Audience:**

Yes

**Broader Impact Concerns:**

The model assumes K is fixed. If were to grow with p and n,, the current system of equations might become computationally intractable or theoretically invalid. A brief sentence on the limitations regarding the size of K relative to p would be beneficial for future research.

**Claims And Evidence:**

Yes

**Requested Changes:**

1. Clarify Independence: Formally add to Theorem 4.1 the assumption that the innovation matrices
$Z_n^{(r)}$ are independent across the index $r$. Currently, this is only mentioned in the introduction and variance calculations.
2. Outlier Discussion: Add a remark regarding the moment assumption (T2).
3. The citation "Wang and Xiang (2026)" is a manuscript, pls update the status, if necessary.
4. Typos: "digaonalizable" → "diagonalizable". "beyond of scope" → "beyond the scope".

**Strengths And Weaknesses:**

Strengths:
1. Significant Generalization: The transition from one to K components is a mathematically non-trivial extension. It addresses a major limitation in spatio-temporal statistics where simple separability is often an unrealistic assumption.
2. Theoretical Rigor: The authors demonstrate a sophisticated use of RMT tools, including the Stieltjes transform method, Montel’s Theorem for subsequential limits, and the Lindeberg principle for non-Gaussian innovations.
3. Novel Non-Commutative Extension: Section 7 provides a meaningful relaxation of the simultaneous diagonalizability requirement. By using low-rank or nuclear-norm perturbations, the authors make the model more robust for practical applications where strict commutativity between spatial covariance matrices rarely holds.
4. Well-Structured Proofs: The "Sketch of the Proof" (Section 4.1) is helpful in navigating the complex derivations that follow, and the use of the Earle-Hamilton Theorem for uniqueness is an elegant application of complex analysis to RMT.

Weakness:
1. In the expansion of $S_n=n^{-1}(\sum X_n^{(r)}) (\sum X_n^{(r)})^*$,  the cross-terms $n^{-1}(X_n^{(r)}) (X_n^{(s)})^*$ are not Hermitian and possess complex dependencies. While the authors rely on the Lindeberg principle to show these terms vanish in the limit, the explicit spectral analysis of these terms is somewhat buried. A more transparent discussion on why these terms do not shift the support of the L.S.D. would strengthen the proof’s accessibility.
2.  The authors assume $2+\eta_0$ moments in (T2). While this is sufficient for the convergence of the L.S.D., it is well known (Bai-Yin law) that 4th moments are typically required to prevent outlier eigenvalues from appearing outside the support. Would be helpful to provide more discussions.
3. The use of the Earle-Hamilton Theorem requires the holomorphic map to map the domain "strictly inside" itself. The authors should clarify if this condition holds uniformly as the imaginary part of the transform approaches the real axis, which is where the spectral density is actually defined.

---

> ### Author Response · Authors · 2026-06-25
> **All points under Requested Changes have been addressed in the revised manuscript. Here is our response to the points under weaknesses and Broader Impact Concerns.**
>
> 1.  $\textbf{Contribution of cross-terms:}$ We have explicitly established that the contribution of the cross-terms is almost surely negligible for large $p,n$ on page $34$ of the revised manuscript. Please refer to equations ($55$), ($56$) and the use of Lemma B.6 in the interim. In this Lemma, we have separately handled the uniform convergence of a finite number of cross-terms (since $K$ is finite) to $0$. As an add-on to the results in the manuscript, we establish the rate of convergence here. Recall the proof of Lemma A.7 (page 27) where we established the following tail bound for quadratic forms valid for any $k \ge 1$ and $D_k(\delta)$ is a constant depending on $k$ and $\delta$.
>
>
> $$p_n:= P\bigg(\underset{1 \leq j \leq n}{\max} \left|\dfrac{1}{n}x_{jn}^*A_jx_{jn} - \dfrac{1}{n}\operatorname{trace}(A_j)\right| > \delta\bigg)
> \leq \frac{D_k(\delta)}{n^{k(\frac{1}{2}-b) - 1}} + \frac{D_k(\delta)}{n^{k(1 - 2b) + 2b-2}}.$$
>
> Take $\delta = n^{-\alpha}$ where $\alpha$ is any number smaller than $0.5 - b$. Then, for slightly modified constants $E_k$ depending on $k$, we have the following tail probability bound:
>
> $$p_n:= P\bigg(\underset{1 \leq j \leq n}{\max} \left|\dfrac{1}{n}x_{jn}^*A_jx_{jn} - \dfrac{1}{n}\operatorname{trace}(A_j)\right| > n^{-\alpha}\bigg)
> \leq \frac{E_k}{n^{k(\frac{1}{2}-b - \alpha) - 1}} + \frac{E_k}{n^{k(1 - 2b - \alpha) + 2b-2}}.
> $$
>
> We note that $\sum_n p_n$ still converges since the above holds for any $k \ge 1$. Therefore, the rate of convergence is at least $O(n^{-\alpha})$. The following expressions explain the critical transition from equation (55) to (56) in the manuscript.
>
> $$\bigg|\frac{1}{p}\sum_{j=1}^n \frac{\sum_{r,s} F_j(r,s)}{1 + \sum_{r,s} E_j(r,s)} - \frac{1}{p}\sum_{j=1}^n \frac{\sum_r \mathbb{E}F_j(r,s))}{1+ \sum_r \mathbb{E}E_j(r,s)} \bigg| = \bigg|\frac{1}{p}\sum_{j=1}^n \frac{\sum_r \mathbb{E}F_j(r,r) + K^2 O(n^{-\alpha})}{1+ \sum_r \mathbb{E}E_j(r,r) + K^2O(n^{-\alpha})} - \frac{1}{p}\sum_{j=1}^n\frac{\sum_r \mathbb{E}F_j(r,s))}{1+ \sum_r \mathbb{E}E_j(r,s)} \bigg|\longrightarrow 0 \text{ a.s.}$$
>
>
> 2. $\textbf{Outlier Discussion:}$ We have added Remark 4.1 in the revised manuscript regarding moment conditions on page 7.
>
> 3. $\textbf{Use of Earle-Hamilton Theorem:}$ In Lemma 4.2 of the manuscript, we show that for any fixed $z \in C_+$, the quantities $P_z(\cdot), Q_z(\cdot)$ defined in equation ($15$) are bounded by a quantity that depends on $\Im(z)$. For a fixed $z \in C_+$, we essentially have
>     $$P_z(C_+^K) \subset S_z^K; Q_z(C_+^K) \subset S_z^K, \text{ where } S_z = \{w \in C_+: |w| \leq C/\Im(z)\}.$$
> Therefore, this is not a uniform (in $z$) bound. However, the capacity in which Earle-Hamilton Theorem has been employed is to merely establish the uniqueness of the fixed point solutions of the functional equations (9) and (11) for any arbitrary $z \in \mathbf{C}_+$. These unique fixed points also turn out to be Stieltjes transforms (equation (9)) and are a crucial ingredient to derive the Stieltjes transform (equation (8)) of the Limiting Spectral Distribution of interest. Once this Stieltjes transform is established, we employ Inversion tools to recover the underlying distribution as shown in a few numerical illustrations in Section 8 of the manuscript. The non-uniformness of this bound in question does not impact the limiting distribution.

---

> > ### Author Response · Authors · 2026-06-25
> > **Our Response to Broader Impact Concerns**
> >
> > We explored the possibility of extending the results on the existence of LSD to the regime where $K$ grows with $p,n$. We will start with the computational issues. As in Section $8$ of the manuscript, we evaluated the LSD under various values of $K$. The spatial and temporal limiting joint spectral distribution (i.e. $H$ and $G$) were constructed in a way such that all the marginals followed independent standard exponential distributions. The limiting density was evaluated at $50$ grid points along the real axis. We state below the times taken by a R program run on on an Ubuntu 24.04.4 LTS system with 12th Gen Intel Core i7 and 16 GiB RAM. We keep p and n fixed at 1000. For $K = 3$, it took $2.36$ minutes, $K = 4$, it took $3.34$ minutes, $K = 5$, it took $4.22$ minutes, $K = 6$, it took $6.15$ minutes, $K = 7$, it took $7.51$ minutes to complete. At least for reasonably small $K$, the growth in complexity appears to be linear in $K$.
> >
> > In light of the explanation provided in our comment on the contribution of the cross-terms, it appears that we can possibly take $K=K_n$ depending on $n$ as follows:
> >     $$K_n \ll O(n^{\alpha/2});\,\alpha < \frac{1}{2} - b; \, b \in \bigg(\frac{1}{2+\eta_0}, \frac{1}{2}\bigg) \text{ and } \eta_0 \text{ is defined in \textbf{T2}.}$$
> > With such a specification for $K_n$, the impact of the cross-terms can certainly be controlled. However, there are many other places where theoretical issues creep up. Even if we assume simultaneous diagonalizability, the corresponding $H$ and $G$ would be infinite dimensional limiting joint spectral distributions. This requires a careful specification of the notion of convergence of the joint empirical spectral distributions ($H_n$ and $G_n$) to their limiting counterparts through the Kolmogorov Consistency Theorem. The limiting objects corresponding to the vector Stieltjes transforms $(\textbf{h}_n, \textbf{g}_n)$ are now elements of $\ell^\infty$. This means that we have to establish a form of uniform convergence with respect to the elements of the sequences as $n,p,K \to \infty$. The proof, in its current form breaks down at many critical places a few of which are highlighted below.
> >
> > $\textbf{1.}$ The proof of Lemma B.5 (pages 31/32), requires (i) $||\textbf{h}_n(z) - \mathbb{E}\textbf{h}_n(z)||_1 \xrightarrow{a.s.} 0$ -- which is no longer valid (see page 28) -- and also  (ii) Lemma B.4 -- which explicitly utilizes the finiteness of $K$ (on page 31). On page 34, we show how the impact of cross terms can be controlled, as indicated in our comment on the contribution of the cross-terms. In the same page, there is a switching of the order of summation which follows through when $K$ is finite.
> >
> > $\textbf{2.}$ The key to establishing a recursive relationship between \textbf{h} and \textbf{g} (equation (12)) is a contraction argument which breaks down when $K \to \infty$. See page 37.
> >
> > $\textbf{3.}$ The necessary and sufficient condition for the limit of a sequence of Stieltjes transforms to be a Stieltjes transform of a probability measure is established in equations (67)/(68) on page 38. Here, finiteness of $K$ is explicitly used.
> >
> > $\textbf{4.}$ In the $\textit{Lindeberg Implementation}$ on page 42, finiteness of $K$ is used.
> >
> > $\textbf{5.}$ While proving the uniqueness of vector fixed point solutions, we require finite K on pages 48 and 49.

---

### Review · Reviewer_1MKj · 2026-06-01

**Summary Of Contributions:**

The paper studies the limiting spectral distribution of sample covariance matrices
\(S_n = n^{-1}XX^*\), where the data matrix is a finite superposition
\[
X = \sum_{r=1}^K A_r^{1/2} Z_r B_r^{1/2}
\]
of matrices with separable covariance structure. Under high-dimensional asymptotics
\(p/n \to c \in (0,\infty)\), moment assumptions on the innovations, convergence of
the joint empirical spectral distributions of the commuting families \(\{A_r\}\) and
\(\{B_r\}\), and bounded average traces, the main theorem establishes almost-sure
weak convergence of the empirical spectral distribution of \(S_n\). The limiting
Stieltjes transform is characterized by a finite system of nonlinear integral
equations involving the limiting joint spectral laws \(H\) and \(G\).

The paper further provides alternative characterizations via the dual covariance
matrix, discusses the atom at zero, proves a continuity property of the fixed-point
solution, derives several special cases including a scaled Marchenko--Pastur case,
proposes relaxations of exact commutativity under low-rank or nuclear
perturbation-type conditions, and includes numerical simulations.

**Audience:**

Yes

**Broader Impact Concerns:**

Nil

**Claims And Evidence:**

Yes

**Requested Changes:**

1. Abstract: ``digaonalizable'' should be ``diagonalizable.''

2.  Page 3: $n^1XX^*$ shall be $n^{-1}XX^*$.


3. Definition 3.6: replace \(I_p\) by \(I_n\) in the definition of
\(\widetilde Q(z)\).

4.  Remark 3.2, the last sentence: the mass is placed
at an eigenvalue of \(S_n\), not at \(\lambda_j^{(r)}\).

5.  Section 4.2: ``which make them distribution-invariant'' should be ``which makes
them distribution-invariant.''

6. Section 4.2: ``Replacing \(Z_n^{(r)}\) with \(W_n^{(r)}\) throughout in (2))''
has an extra closing parenthesis.

7.  Corollary 6.3: specify the branch of the square root used in the Stieltjes
transform formula.

8. Simulation section: ``We repeated these simulations at for \(K=1,2,3\)''
should be ``We repeated these simulations for \(K=1,2,3\).''

9.  Simulation section: ``\(c=0.5,1,1,2.5\)'' appears to contain a duplicated
\(1\).

10.  Simulation section: ``numerical approximate'' should be ``numerical
approximation.''

11.  Figure captions use inconsistent punctuation, e.g. ``\(K=1:c=0.5\)'' versus
``\(K=1,c=2.5\)''; please standardize.

**Strengths And Weaknesses:**

Overall, the problem is interesting and the finite-\(K\) reduction to a system
involving joint spectral distributions is potentially useful. However, several
technical and presentation issues should be addressed before publication.

There are two major comments:

1. Since you assume that the $A$ and $B$ matrices are simultaneously diagonalizable, in the case when $Z$
matrices are Gaussian, it seems that your model is simply a random matrix with general variance profile.
There have been many papers in this direction; see arXiv:1506.05098 and arXiv:1612.04428, for instance.
Could you make a connection with your result and those result, especially the vector Dyson Equation in those papers.

2.  One main assumption is the simultaneous diagonalizability of the $A$ and $B$ matrices. There is some discussion on the case when it is not satisfied in Section 7, but I hope the authors can provide more heuristic comments on why such a constraint is largely needed, how it might be completely removed, and what the difficulty is. Also, is there any comment on whether such a constraint is acceptable from an application point of view?

---

> ### Author Response · Authors · 2026-06-25
> **All points under Requested Changes have been addressed in the revised manuscript. Here is our response to the two major comments.**
>
> $\textbf{Dyson Equations:}$ The self-consistent equation (9), where $\textbf{h}(z) \in C_+^K$ is a fixed point solution, and equation (11), where $\textbf{g}(z) \in C_+^K$ is a fixed point solution, are exactly quadratic vector equations or \textit{Vector Dyson Equations} as introduced in these papers arXiv:1506.05098 and arXiv:1612.04428.
>
> $\textbf{Assumption of Simultaneous Diagonalizability:}$ We address this point from different perspectives. First, under the assumption of simultaneous diagonalizability condition, the average trace of the deterministic equivalent (established in Theorem 4.4) of the resolvent $Q(z) = (S_n-zI_p)^{-1}$ reduces to an integral equation as given in Remark 4.3. Recall the deterministic equivalent object:
>
> $$
> \bar{Q}(z) = \bigg(-zI_p + \sum_{r=1}^K A_{n}^{(r)} {\bigg(\frac{1}{n}\operatorname{trace}\{B_{n}^{(r)}[I_n + c_n\sum_{s=1}^K\mathbb{E}h_{sn}(z)B_{n}^{(s)}]^{-1}\}\bigg)} \bigg)^{-1}.
> $$
> The tracial expression called out in the above equation is $t_r(z) = t_r(z, c_n, B_1, \ldots, B_K)$. Since the $B_k$'s commute, we have
>
> $$
>     t_r(z) = \int_{\mathbb{R}_+^n} \frac{ \theta_r dG_n(\boldsymbol{\theta})}{1 + c_n\boldsymbol{\theta}^T\mathbf{E}\textbf{h}_n(z)} = -zO_r(z, c_n\mathbb{E}\textbf{h}_n(z), G_n).
> $$
>
> Since the $A_k$'s commute as well, the average trace of $\bar{Q}(z)$ thus simplifies to:
> \begin{align*}
>     \frac{1}{p}\operatorname{trace}\bar{Q}(z) = \frac{1}{p}\operatorname{trace}(-zI_p + \sum_{r=1}^K t_r(z)A_n^{(r)})^{-1} = \int_{\mathbb{R}_+^p} \frac{ dH_n(\boldsymbol{\lambda})}{-z(1 + \boldsymbol{\lambda}^T\textbf{O}(z, c_n\mathbb{E}\textbf{h}_n(z), G_n))}.
> \end{align*}
> This vastly helps us since the average trace of the deterministic equivalent approaches the limiting Stieltjes transform (of the LSD) of the sample covariance matrix of interest. Note that Section 7 establishes the validity of the expressions under a few notions of approximate commutativity.
>
>
> Based on the existing literature on free probability, we expect that, in order for an LSD to hold, we still need certain regularity conditions on the joint behavior of the component covariance matrices $A_k$'s and $B_k$'s. One possible relaxation is to assume the joint convergence of the mixed tracial moments of the covariance matrices of the components to some limiting quantities satisfying an appropriate relationship. In order to prove the existence of LSD under such significantly less restrictive settings, we
> may have to resort to more involved combinatorial techniques. Such techniques will be needed if one were to apply the method of moments for establishing existence of LSD, as is a common practice in the free probability literature. The current technique based on Stieltjes transform is not suitable for such situations.
>
>
> From an application point of view, Joint Approximate Diagonalization might be more realistic as opposed to exact simultaneous diagonalizability, i.e. there exists unitary matrices $P \in \mathbb{C}^{p \times p}$ and $Q \in \mathbb{C}^{n \times n}$ and small perturbations $E_k, F_k$ such that
>     $$A_k = P\Lambda_kP^* + E_k; B_k = Q\Gamma_kQ^* + F_k.$$
> Glashoff and Bronstein (2003), Loring, Glashoff and Bronstein (2013) and Li, Lu and Yu (2023) have worked using the idea of almost commuting matrices with applications in quantum mechanics, computer graphics, signal processing, blind source separation,
> multimedia documents, audio and video, images with different lighting conditions, medical imaging modalities and 3-D shape analysis.
>
> Beyond the above examples, in spatio-temporal data analysis, which provides a key statistical motivation for the assumed model, if we have a signal-plus-noise type representation, where the signal is of low rank and the noise component is stationary in space and time, then a model like ours with approximate simultaneous diagonalizability of the component covariances would be applicable.
> Here, the temporal variation across the $K$ components of the noise can be represented by a common circulant matrix (Fourier basis),
> while spatial variation can be represented by a common orthogonal basis, which, when the spatial domain is a sphere, would be the spherical harmonics basis (or, more generally, eigenfunctions of the Laplacian operator on the spatial domain).

---

### Author Response · Authors · 2026-06-25
**Miscellaneous information**

A detailed point‑by‑point response (including all equations and references) has been uploaded as a supplementary file. Please refer to that PDF for the full response.

---

### Decision · Action_Editor_9p9B · 2026-06-28

**Recommendation:** Accept as is

**Comment:**

I am pleased to recommend acceptance of this submission. The paper makes a clean and substantial contribution to high-dimensional random matrix theory, both reviewers recommended acceptance, and the authors' revision together with a detailed response letter have satisfactorily resolved every point that was raised.

**Audience:**

Yes

**Claims And Evidence:**

Yes